# Optimization with Access to Auxiliary Information

**El Mahdi Chayti**  *el-mahdi.chayti@epfl.ch*
*EPFL*

**Sai Praneeth Karimireddy**  *sp.karimireddy@berkeley.edu*
*UC Berkeley*

Reviewed on OpenReview: *https://openreview.net/forum?id=kxYqgSkH8I*

## Abstract

We investigate the fundamental optimization question of minimizing a *target* function $f(\boldsymbol{x})$, whose gradients are expensive to compute or have limited availability, given access to some *auxiliary* side function $h(\boldsymbol{x})$ whose gradients are cheap or more available. This formulation captures many settings of practical relevance, such as i) re-using batches in SGD, ii) transfer learning, iii) federated learning, iv) training with compressed models/dropout, Et cetera. We propose two generic new algorithms that apply in all these settings; we also prove that we can benefit from this framework under the Hessian similarity assumption between the target and side information. A benefit is obtained when this similarity measure is small; we also show a potential benefit from stochasticity when the auxiliary noise is correlated with that of the target function.

## 1 Introduction

**Motivation.** Stochastic optimization methods such as SGD (Robbins & Monro, 1951b) or Adam (Kingma & Ba, 2014) are arguably at the core of the success of large-scale machine learning (LeCun et al., 2015; Schmidhuber, 2015). This success has led to significant (perhaps even excessive) research efforts dedicated to designing new variants of these methods (Schmidt et al., 2020). In all these methods, massive datasets are collected centrally on a server, and immense parallel computational resources of a data center are leveraged to perform training (Goyal et al., 2017; Brown et al., 2020). Meanwhile, modern machine learning is moving away from this centralized training setup with new paradigms emerging, such as i) distributed/federated learning, ii) semi-supervised learning, iii) personalized/multi-task learning, iv) model compression, Et cetera. Relatively little attention has been devoted to these more practical settings from the optimization community. In this work, we focus on extending the framework and tools of stochastic optimization to bear on these novel problems.

At the heart of these newly emergent training paradigms lies the following fundamental optimization question: We want to minimize a target loss function $f(\boldsymbol{x})$, but computing its stochastic gradients is either very expensive or unreliable due to limited data. However, we assume having access to some auxiliary loss function $h(\boldsymbol{x})$ whose stochastic gradient computation is relatively cheaper or more available. For example, in transfer learning, $f(\boldsymbol{x})$ would represent the downstream task we care about and for which we have very little data available, whereas $h(\boldsymbol{x})$ would be the pretraining task for which we have plenty of data (Yosinski et al., 2014). Similarly, in semi-supervised learning, $f(\boldsymbol{x})$ would represent the loss over our clean labeled data, whereas $h(\boldsymbol{x})$ represents the loss over unlabeled or noisily labeled data (Chapelle et al., 2009). Our challenge is the following question:

How can we leverage an auxiliary $h(\boldsymbol{x})$ to speed up the optimization of our target loss function $f(\boldsymbol{x})$?

Of course, if $f(\boldsymbol{x})$ and $h(\boldsymbol{x})$ are entirely unrelated, our task is impossible, and we cannot hope for any speedup over simply running standard stochastic optimization methods (e.g., SGD) on $f(\boldsymbol{x})$. Thus, the additional

question before us is to define and take advantage of useful similarity measures between $f(\boldsymbol{x})$ and $h(\boldsymbol{x})$. In this work, we will mainly consider Hessian similarity (defined later in Assumption 3.3) and leave devising more practical similarity measures as a future direction.

**Contributions.** The main results in this work are

- We formulate the following as stochastic optimization with auxiliary information: i)Re-using batches in SGD, ii) Semi-supervised learning, iii) transfer learning, iv) Federated Learning, v) personalized learning, and vi) training with sparse models.
- We show a useful and simple trick (Eq3) to construct biased gradients using gradients from an auxiliary function.
- Based on the above trick, we design a biased gradient estimator of $f(\boldsymbol{x})$, which reuses stochastic gradients of $f(\boldsymbol{x})$ and combines it with gradients of $h(\boldsymbol{x})$.
- We then use this estimator to develop algorithms for minimizing smooth non-convex functions. Our methods improve upon known optimal rates that don't use any side information.

**Related work.** Optimizing one function $f$ while accessing another function $h$ (or its gradients) is an important idea that has only been considered in specific cases in machine learning and optimization communities. To the best of our knowledge, this problem has never been regarded in all of its generality before this time. For this reason, we are limited to only citing works that used this idea in particular cases. Lately, masked training of neural networks was considered, for example, in (Alexandra et al., 2019; Amirkeivan et al., 2021); this approach is a special case of our framework, where the auxiliary information is given by the sub-network (or mask). In distributed optimization, (Shamir et al., 2013) define sub-problems based on available local information; the main problem with this approach is that the defined sub-problems need to be solved precisely in theory and to high precision in practice. In Federated Learning (Konecny et al., 2016; McMahan et al., 2017a; Mohri et al., 2019), the local functions (constructed using local datasets) can be seen as side information. Applying our framework recuperates an algorithm close to MiMe (Karimireddy et al., 2020a). In personalization, Chayti et al. (2021) study the collaborative personalization problem where one user optimizes its loss by using gradients from other available users (that are willing to collaborate); again, these collaborators can be seen as side information, one drawback of the approach in (Chayti et al., 2021) is that they need the same amount of work from the main function and the helpers, in our case we alleviate this by using the helpers more.

There is also auxiliary learning (Baifeng et al.; Aviv et al.; Xingyu et al.) that is very similar to what we are proposing in this work. Auxiliary learning also has the goal of learning one given task using helper tasks, however, all these works come without any theoretical convergence guarantees, furthermore, our approach is more general.

The proposed framework is general enough to include all the above problems and more. More importantly, we don't explicitly make assumptions on how the target function $f$ is related to the auxiliary side information $h$ (potentially a set of functions) like in Distributed optimization or Federated learning where we assume $f$ is the average of the side-information $h$. Also, it is not needed to solve the local problems precisely as expected by DANE (Shamir et al., 2013).

## 2 General Framework

Our main goal is to solve the following optimization problem:

$$\min_{\boldsymbol{x} \in \mathbb{R}^d} f(\boldsymbol{x}), \tag{1}$$

and we suppose that we have access to an auxiliary function $h(\boldsymbol{x})$ that is related to $f$ in a sense that we don't specify at this level.

Specifically, we are interested in the stochastic optimization framework. We assume that the target function is of the form $f(\boldsymbol{x}) := \mathbb{E}_{\xi_f}\left[f(\boldsymbol{x}; \xi_f)\right]$ over $\boldsymbol{x} \in \mathbb{R}^d$ while the auxiliary function has the form $h(\boldsymbol{x}) := \mathbb{E}_{\xi_h}\left[h(\boldsymbol{x}; \xi_h)\right]$ defined over the same parameter space. We will refer to these two functions simply as $f$ and $h$ and stress that they should not be confused with $f(\cdot; \zeta_f)$ and $h(\cdot; \zeta_h)$. It is evident that if both functions $f$ and $h$

are unrelated, we can't hope to benefit from the auxiliary information $h$. Hence, we need to assume some similarity between $f$ and $h$. In our case, we propose to use the hessian similarity (defined in assumption 3.3).

Many optimization algorithms can be framed as sequential schemes that, starting from an estimate $\boldsymbol{x}$ of the minimizer, compute the next (hopefully better estimate) $\boldsymbol{x}^+$ by solving a potentially simpler problem

$$\boldsymbol{x}^+ \in arg \min_{\boldsymbol{z} \in \mathbb{R}^d} \{\hat{f}(\boldsymbol{z}; \boldsymbol{x}) + \lambda R(\boldsymbol{z}; \boldsymbol{x})\}, \tag{2}$$

where $\hat{f}(\cdot; \boldsymbol{x})$ is an approximation of $f$ around the current state $\boldsymbol{x}$, $R(\boldsymbol{z}; \boldsymbol{x})$ is a regularization function that measures the quality of the approximation and $\lambda$ is a parameter that trades off the two terms.

For example, the gradient descent algorithm is obtained by choosing $\hat{f}(\boldsymbol{z}; \boldsymbol{x}) = f(\boldsymbol{x}) + \nabla f(\boldsymbol{x})^\top (\boldsymbol{z} - \boldsymbol{x})$, $R(\boldsymbol{z}; \boldsymbol{x}) = \frac{1}{2}\|\boldsymbol{z} - \boldsymbol{x}\|_2^2$ and $\lambda = \frac{1}{\eta}$ where $\eta$ is the stepsize. Mirror descent uses the same approximation $\hat{f}$ but a different regularizer $R(\boldsymbol{z}; \boldsymbol{x}) = \mathcal{D}_\phi(\boldsymbol{z}; \boldsymbol{x})$ where $\mathcal{D}_\phi$ is the Bregman divergence of a certain strongly-convex function $\phi$.

We take inspiration from this approach in this work. However, we would like to take advantage of the existence of the auxiliary function $h$. We will mainly focus on first-order approximations of $f$ throughout this work; we will also fix the regularization function $R(\boldsymbol{z}; \boldsymbol{x}) = \frac{1}{2}\|\boldsymbol{z} - \boldsymbol{x}\|_2^2$, we note that our ideas can be easily adapted to other choices of $R$ and more involved approximations of $f$.

For any function $h$, we can always write $f$ as

$$f(\boldsymbol{z}) := \underbrace{h(\boldsymbol{z})}_{\text{cheap}} + \underbrace{f(\boldsymbol{z}) - h(\boldsymbol{z})}_{\text{expensive}},$$

we call the first term "cheap", but this should not be understood strictly; it can also, for example, mean more available.

A very straightforward approach is to use $f$ whenever it is available and use $h$ as its proxy whenever it is not; we call this approach **the naive approach**. This approach is equivalent to simply ignoring (in other words, using a zeroth-order approximation of) the "expensive" part.

A more involved strategy is to approximate the "expensive" part $f(\boldsymbol{z}) - h(\boldsymbol{z})$ as well but not as much as the "cheap" part $h(\boldsymbol{z})$. We can do this by approximating $f(\boldsymbol{z}) - h(\boldsymbol{z})$ around $\boldsymbol{x}$ ( a global state, or a snapshot, the idea is that it is the state of $f$) and approximating $h(\boldsymbol{z})$ around $\boldsymbol{y}$ (a local state in the sense that it is updated by $h$). Doing this, we get the following update rule:

$$\boldsymbol{y}^+ \in arg \min_{\boldsymbol{z} \in \mathbb{R}^d} \{\hat{f}(\boldsymbol{z}; \boldsymbol{y}, \boldsymbol{x}) + \frac{1}{2\eta}\|\boldsymbol{z} - \boldsymbol{y}\|_2^2\},$$

where $\hat{f}(\boldsymbol{z}; \boldsymbol{y}, \boldsymbol{x}) := h(\boldsymbol{y}) + f(\boldsymbol{x}) - h(\boldsymbol{x}) + \nabla h(\boldsymbol{y})^\top (\boldsymbol{x} - \boldsymbol{y}) + (\nabla h(\boldsymbol{y}) - \nabla h(\boldsymbol{x}) + \nabla f(\boldsymbol{x}))^\top (\boldsymbol{z} - \boldsymbol{x})$.

This is equivalent to

$$\boldsymbol{y}^+ = \boldsymbol{y} - \eta(\nabla h(\boldsymbol{y}) - \nabla h(\boldsymbol{x}) + \nabla f(\boldsymbol{x})) \tag{3}$$

We will refer to (3) as a local step because it uses a new gradient of $h$ to update the state. The idea is that for each state $\boldsymbol{x}$ of $f$, we perform a number of local steps, then update the state $\boldsymbol{x}$ based on the last "local" steps.

**Control variates and SVRG.** (3) can be understood as a generalization of the control variate idea used in SVRG (Johnson & Zhang, 2013). To optimize a function $f(\boldsymbol{x}) := \frac{1}{n}\sum_{i=1}^n f_i(\boldsymbol{x})$, SVRG uses the modified gradient $\boldsymbol{g}_{SVRG} = \nabla f_i(\boldsymbol{y}) - \nabla f_i(\boldsymbol{x}) + \nabla f(\boldsymbol{x})$ where $\boldsymbol{x}$ is a snapshot that is updated less frequently and $i$ is sampled randomly so that this new gradient is still unbiased. The convergence of SVRG is guaranteed by the fact that the "error" of this new gradient is $\mathbb{E}\|\boldsymbol{g}_{SVRG} - \nabla f(\boldsymbol{x})\|_2^2 = \mathcal{O}(\|\boldsymbol{y} - \boldsymbol{x}\|_2^2)$ so that if $\boldsymbol{y} - \boldsymbol{x} \to 0$, then convergence is guaranteed without needing to take small step-sizes. The main idea of our work is to use instead of $f_i$ another function $h$ that is related to $f$; this means using a gradient $\boldsymbol{g} = \nabla h(\boldsymbol{y}) - \nabla h(\boldsymbol{x}) + \nabla f(\boldsymbol{x})$, then if we can still guarantee that the error is $\mathcal{O}(\|\boldsymbol{y} - \boldsymbol{x}\|_2^2)$ everything should still work fine.

**Other Variance reduction techniques.** given the form of (3) that is very similar to the SVRG gradient, it is natural to ask what would happen when using a form similar to other variance reduction techniques such as SARAH (Nguyen et al., 2017) (this will amount to choosing a gradient $\boldsymbol{g}^t = \nabla h(\boldsymbol{y}^t) - \nabla h(\boldsymbol{y}^{t-1}) + \boldsymbol{g}^{t-1}$ with $\boldsymbol{g}^0 = \nabla f(\boldsymbol{y}^0 := \boldsymbol{x})$). Unfortunately, this choice leads to the same theoretical rate obtained by the SVRG-like choice, the main reason being that on top of the biasedness of SARAH, the fact that $h$ is potentially different from $f$ (even in average) introduces another biasedness, which limits the potential gain; moreover, it is not evident how to treat the case where we only have access to stochastic gradients of $f$.

**What if we can't access the true gradient of $f$?** When $f$ is not a finite average, we can't access its true gradient; in this case, we propose replacing the "correction" $\nabla f(\boldsymbol{x}) - \nabla h(\boldsymbol{x})$ by a quantity $\boldsymbol{m}_{f-h}$ that is a form of momentum (takes into account past observed gradients of $f - h$), the idea of using momentum is used to stabilize the estimate of the quantity $\nabla f(\boldsymbol{x}) - \nabla h(\boldsymbol{x})$ as momentum can be used to reduce the variance. Specifically, we use $\boldsymbol{g} = \nabla h(\boldsymbol{y}) + \boldsymbol{m}_{f-h}$. We note that for this estimate we have $\mathbb{E}\|\boldsymbol{g} - \nabla f(\boldsymbol{y})\|_2^2 = \mathcal{O}(\|\boldsymbol{y} - \boldsymbol{x}\|_2^2 + \|\boldsymbol{m}_{f-h} - \nabla f(\boldsymbol{x}) + \nabla h(\boldsymbol{x})\|_2^2)$ which means that $\boldsymbol{g}$ approximates $\nabla f(\boldsymbol{y})$ as long as $\boldsymbol{y}$ is not far from $\boldsymbol{x}$ and $\boldsymbol{m}_{f-h}$ is a good estimate of the quantity $\nabla f(\boldsymbol{x}) - \nabla h(\boldsymbol{x})$.

**Local steps or a subproblem?** we note that another possible approach is, instead of defining local steps based on $h$, to define a subproblem that gives the next estimate of $f$ directly by solving

$$\boldsymbol{x}^+ \in arg \min_{\boldsymbol{y} \in \mathbb{R}^d} \{h(\boldsymbol{y}) + \boldsymbol{m}_{f-h}^\top(\boldsymbol{y} - \boldsymbol{x}) + \frac{1}{2\eta}\|\boldsymbol{y} - \boldsymbol{x}\|_2^2\} . \tag{4}$$

Our results can be understood as approximating a solution to this sub-optimization problem (4).

**Notation.** For a given function $J$, we denote $\boldsymbol{g}_J(\cdot, \xi_J)$ an unbiased estimate of the gradient of $J$ with randomness $\xi_J$.

**General meta-algorithm.** Based on the discussion above, we propose the (meta)-Algorithm 1: at the beginning of each round $t$, we have an estimate $\boldsymbol{x}^{t-1}$ of the minimizer. We sample a new $\zeta_{f-h}$, and compute $\boldsymbol{g}_{f-h}(\boldsymbol{x}_{t-1}, \zeta_{f-h})$, a noisy unbiased estimate of the gradient of $f - h$; we then update $\boldsymbol{m}_{f-h}^t$ a momentum of $f - h$. We transfer both $\boldsymbol{x}^{t-1}$ and $\boldsymbol{m}_{f-h}^t$ to the helper $h$ which uses both to construct a set of biased gradients $\boldsymbol{d}_k^t$ of $\nabla f(\boldsymbol{y}_{k-1}^t)$ that are updated by $h$. These biased gradients are then used to update the "local" states $\boldsymbol{y}_k^t$, then $f$ updates its own state $\boldsymbol{x}^t$ based on the last local states $\boldsymbol{y}_{0 \leq k \leq K}^t$; throughout this work, we simply take $\boldsymbol{x}_t = \boldsymbol{y}_K^t$.

---

**Algorithm 1** stochastic optimization of $f$ with access to the auxiliary $h$

---

**Require:** $\boldsymbol{x}_0$, $\eta$, $T$, $K$
  **for** $t = 1$ to $T$ **do**
    sample $\boldsymbol{g}_{f-h}(\boldsymbol{x}^{t-1}, \xi_{f-h}^t) \approx \nabla f(\boldsymbol{x}^{t-1}) - \nabla h(\boldsymbol{x}^{t-1})$
    update $\boldsymbol{m}_{f-h}^t \approx \nabla f(\boldsymbol{x}^{t-1}) - \nabla h(\boldsymbol{x}^{t-1})$ § momentum
    define $\boldsymbol{y}_0^t = \boldsymbol{x}^{t-1}$
    **for** $k = 1$ to $K$ **do**
      sample $\boldsymbol{g}_h(\boldsymbol{y}_{k-1}^t, \xi_h^{t,k}) \approx \nabla h(\boldsymbol{y}_{k-1}^t)$
      use it and $\boldsymbol{m}^t$ to form $\boldsymbol{d}_k^t \approx \nabla f(\boldsymbol{y}_{k-1}^t)$
      $\boldsymbol{y}_k^t = \boldsymbol{y}_{k-1}^t - \eta \boldsymbol{d}_k^t$
    **end for**
    update $\boldsymbol{x}^t$
  **end for**

---

For our purposes we can take $\boldsymbol{d}_k^t = \boldsymbol{g}_h(\boldsymbol{y}_{k-1}^t, \xi_h^{t,k}) + \boldsymbol{m}_{f-h}^t$ which should be a good approximation of $\nabla f(\boldsymbol{y}_{k-1}^t)$ and $\boldsymbol{m}_{f-h}^t$ is a momentum of $f - h$, we will consider two options: **classical momentum** or **MVR** (for momentum based variance reduction (Cutkosky & Orabona, 2019)).

**Decentralized auxiliary information.** More generally, we can assume having access to $N$ auxiliary functions $h_i(\boldsymbol{x}) := \mathbb{E}_{\xi_{h_i}}[h_i(\boldsymbol{x}; \xi_{h_i})]$. While we can treat this case by taking $h = (1/N)\sum_{i=1}^N h_i$, we propose a more interesting solution that also works if the helpers $h_i$ are decentralized and cannot live in the same server.

In this case, we can sample a set $S^t$ of helpers, each $h_i \in S^t$ will do the updates exactly as in Algorithm1, but this time $\boldsymbol{x}^t$ will be constructed using all $\{\boldsymbol{y}_{i,K}^t, i \in S^t\}$. In our case, we propose to use the average $\boldsymbol{x}^t = (1/S) \sum_{i \in S^t} \boldsymbol{y}_{i,K}^t$ .

## 3 Algorithms and Results

We discuss here some particular cases of Algorithm 1 based on choices of $\boldsymbol{m}_{f-h}^t$ and $\boldsymbol{d}_k^t$, which we kept a little bit vague purposefully.

We will consider mainly two approaches. We call the first one the **Naive approach** and the second one we refer to as **Bias correction**.

We remind the reader that we can take $\boldsymbol{d}_k^t = \boldsymbol{g}_h(\boldsymbol{y}_{k-1}^t, \xi_h^{t,k}) + \boldsymbol{m}_{f-h}^t$.

**Naive approach.** This approach is exactly as suggested by its name, naive; it simply ignores the part $f - h$ or, in other words, sets $\boldsymbol{m}_{f-h}^t = 0$. The main idea is to use gradients (or gradient estimates) of $f$ whenever they are available and use gradients of $h$ when gradients of $f$ are not available. In our case, we alternate between one step using a gradient of $f$ and $(K-1)$-steps using gradients of $h$ without any correction. We will show that this approach suffers heavily from the bias between the gradients of $f$ and $h$. It is worth noting that in federated learning, this approach corresponds to Federated averaging (McMahan et al., 2017a).We remind the reader that we can take $\boldsymbol{d}_k^t = \boldsymbol{g}_h(\boldsymbol{y}_{k-1}^t, \xi_h^{t,k}) + \boldsymbol{m}_{f-h}^t$.

**Bias correction.** In the absence of noise, this approach simply implements (3). Specifically, the inner loop in Algorithm 1 does the following:

$$\boldsymbol{y}_k^t = \boldsymbol{y}_{k-1}^t - \eta(\underbrace{\nabla h(\boldsymbol{y}_{k-1}^t) - \nabla h(\boldsymbol{x}^{t-1}) + \nabla f(\boldsymbol{x}^{t-1})}_{:=\boldsymbol{d}_k^t}) .$$

In the noisy case (when we can only have access to noisy gradients of $f$), we approximate the above step in the following way:

$$\boldsymbol{y}_k^t = \boldsymbol{y}_{k-1}^t - \eta(\underbrace{\boldsymbol{g}_h(\boldsymbol{y}_{k-1}^t, \xi_{k-1}^t) + \boldsymbol{m}_{f-h}^t}_{:=\boldsymbol{d}_k^t}) ,$$

where $\boldsymbol{m}_{f-h}^t$ is a momentum of $f - h$. We consider two methods for defining this momentum:

$$\boldsymbol{m}_{f-h}^t = (1-a)\boldsymbol{m}_{f-h}^{t-1} + a\boldsymbol{g}_{f-h}(\boldsymbol{x}^{t-1}, \xi_{f-h}^{t-1}) \qquad (\textbf{AuxMOM}), \tag{5}$$

$$\begin{aligned} \boldsymbol{m}_{f-h}^t = {}&(1-a)\boldsymbol{m}_{f-h}^{t-1} + a\boldsymbol{g}_{f-h}(\boldsymbol{x}^{t-1}, \xi_{f-h}^{t-1}) \\ &+ (1-a)\big(\boldsymbol{g}_{f-h}(\boldsymbol{x}^{t-1}, \xi_{f-h}^{t-1}) - \boldsymbol{g}_{f-h}(\boldsymbol{x}^{t-2}, \xi_{f-h}^{t-1})\big) \qquad (\textbf{AuxMVR}). \end{aligned} \tag{6}$$

**AuxMOM** simply uses the classical momentum, whereas **AuxMVR** uses the momentum-based variance reduction technique introduced in (Cutkosky & Orabona, 2019).

**Assumptions.** To analyze our algorithms, we will make the following assumptions on the target $f$ and the helper $h$.

**Assumption 3.1. (Smoothness.)** We assume that $f$ has $L$-Lipschitz gradients and satisfy

$$\|\nabla f(\boldsymbol{x}) - \nabla f(\boldsymbol{y})\|_2 \le L\|\boldsymbol{x} - \boldsymbol{y}\|_2 .$$

**Assumption 3.2. (Variance.)** The stochastic gradients $\boldsymbol{g}_f(\boldsymbol{x}; \zeta_f)$, $\boldsymbol{g}_h(\boldsymbol{x}; \zeta_h)$ and $\boldsymbol{g}_{f-h}(\boldsymbol{x}; \zeta_{f-h})$ are unbiased, and satisfy

$$\mathbb{E}_{\zeta_J}\|\boldsymbol{g}_J(\boldsymbol{x}; \zeta_J) - \nabla J(\boldsymbol{x})\|_2^2 \le \sigma_J^2 , J \in \{f, h, f-h\} .$$

In Assumption 3.2, we assume that we directly have access to unbiased gradient estimates of $f - h$; this does not restrict in any way our work since $\boldsymbol{g}_f - \boldsymbol{g}_h$ is such an estimate; however, this last estimate has a variance of $\sigma_{f-h}^2 = \sigma_f^2 + \sigma_h^2$, in general, it is possible to have a correlated estimate such that $\sigma_{f-h}^2 < \sigma_f^2 + \sigma_h^2$. If the batches $\xi_f$ and $\xi_h$ are drawn such that $\boldsymbol{g}_f$ and $\boldsymbol{g}_h$ are positively correlated, then it is even possible to have $\sigma_{f-h}^2 < \sigma_f^2$.

**Assumption 3.3. Hessian similarity.** Finally, we will assume that for some $\delta \in [0, 2L]$ we have

$$\|\nabla^2 f(\boldsymbol{x}) - \nabla^2 h(\boldsymbol{x})\|_2 \le \delta \,.$$

Note that if $h(\cdot)$ is also smooth (satisfies Assumption 3.1), then we would have Hessian similarity with $\delta \le 2L$ since

$$\|\nabla^2 f(\boldsymbol{x}) - \nabla^2 h(\boldsymbol{x})\|_2 \le \|\nabla^2 f(\boldsymbol{x})\|_2 + \|\nabla^2 h(\boldsymbol{x})\|_2 \le 2L \,.$$

As sanity checks, $h = 0$ corresponds to $\delta = L$ (this case should not lead to any benefit), and $h = f$ gives $\delta = 0$; we will consider these two cases to verify our convergence rates.

**Discussion of the assumptions.** Assumptions 3.1 and 3.2 are very common assumptions in the optimization literature. Under these two assumptions, it is well-known that SGD (Robbins & Monro, 1951a; Kiefer & Wolfowitz, 1952) has an optimal convergence rate. Assumption 3.3 was used extensively in Federated Learning (Karimireddy et al., 2020a;b) and was considered for personalization (Chayti et al., 2021).

**Relaxing the Hessian similarity.** Jingzhao et al. propose a generalized smoothness assumption where the norm of the Hessian can grow with the norm of the gradient. We believe it possible to extend our theory to accommodate such an assumption. In the same spirit, we can let $\|\nabla^2 f(\boldsymbol{x}) - \nabla^2 h(\boldsymbol{x})\|_2$ grow with $\|\nabla f(\boldsymbol{x})\|_2$. More important than this is finding similarity measures that apply to some of the potential applications we cite in Section 5.

## 4 Results

We start by showing that the convergence rate of the naive approach is dominated by the gradient bias. We then show the convergence rate of our momentum variant **AuxMOM** that will be compared to SGD/GD. We will also state the convergence rate of our **AuxMVR** variant and compare it to MVR/GD.

**Notation.** We assume $f$ is bounded from below and denote $f^\star = \inf_{\boldsymbol{x} \in \mathbb{R}^d} f(\boldsymbol{x})$.

**Remark.** We would like to note that while we state our results for the case of one helper $h$, they also apply without needing any additional assumption when we have $N$ decentralized helpers $h_1, \ldots, h_N$ from which we sample $S = 1$ helper at random. In the case where $S > 1$, we need an additional weak-convexity assumption to deal with the averaging performed at the end of each step; this general case is treated in Appendix C.4.

### 4.1 Naive approach

In this section, we show the convergence results using the naive approach that uses a gradient of $f$ followed by $K - 1$ gradients of $h$. For the analysis of this case (and only of this case), we need to make an assumption on the gradient bias between $f$ and $h$.

**Assumption 4.1.** The gradient bias between $f$ and $h$ is $(m, \zeta^2)-$bounded: $\forall \boldsymbol{x} \in \mathbb{R}^d \;:\; \|\nabla f(\boldsymbol{x}) - \nabla h(\boldsymbol{x})\|_2^2 \le m\|\nabla f(\boldsymbol{x})\|_2^2 + \zeta^2$.

> **Theorem 4.2.** *There exists $f$ and $h$ satisfying assumptions 3.1,3.2,3.3,4.1 with $\delta = 0$, $\sigma_f = \sigma_h = 0$ such that $\frac{1}{KT} \sum_{t=1}^{T} \|\nabla f(\boldsymbol{x}^{t-1})\|_2^2 = \Omega(\zeta^2)$.*

Using the biased SGD analysis in (Ajalloeian & Stich, 2020), it is easy to prove an upper bound, but we only need a lower bound for our purposes. Theorem 4.2 shows that this naive approach cannot guarantee convergence to less than $\zeta^2$ (up to some constant).

**Note.** Theorem 4.2 is very loose for Federated Averaging as, in this case, the function $f$ is directly tied to the helper entities (it is the average). Nevertheless, it's worth noting that heterogeneity and client drift,

which are akin to gradient bias, are recognized as factors that can constrain the performance of FedAVG. Consequently, there have been efforts to mitigate these issues through approaches such as those discussed in (Karimireddy et al., 2020b;a).

In what follows, we will show that our two proposed algorithms solve this bias problem.

### 4.2 Momentum based approach

We consider the instance of Algorithm 1 with the momentum choice in (5). For clarity, a detailed Algorithm can be found in the Appendix Algorithm 3.

**Convergence rate.** We prove the following theorem that gives the convergence rate of this algorithm in the non-convex case.

---

**Theorem 4.3.** *Under assumptions A3.1, 3.2,3.3. For $\boldsymbol{m}^0$ such that $E^0 \leq \sigma_{f-h}^2/T$, $a = \max(\frac{1}{T}, 36\delta K\eta)$ and $\eta = \min(\frac{1}{L}, \frac{1}{192\delta K}, \sqrt{\frac{F^0}{144L\beta K^2 T\sigma_f^2}})$, we get the following :*

$$\frac{1}{KT}\sum_{t=1}^{T}\sum_{k=0}^{K-1}\mathbb{E}\big[\|\nabla f(\boldsymbol{y}_k^t)\|_2^2\big] \leq \mathcal{O}\Big(\sqrt{\frac{L\beta F^0\sigma_f^2}{T}} + \frac{(L+\delta K)F^0}{KT} + \frac{\sigma_{f-h}^2}{T} + \frac{\sigma_h^2}{K}\Big).$$

*Where $F^0 = f(\boldsymbol{x}^0) - f^\star$, $E^0 = \mathbb{E}[\|\boldsymbol{m}^0 - \nabla f(\boldsymbol{x}^0) + \nabla h(\boldsymbol{x}^0)\|_2^2]$ and $\beta = \mathcal{O}\big(\frac{\delta}{L}\frac{\sigma_{f-h}^2}{\sigma_f^2} + \frac{1}{K}(1+\frac{\delta}{L})\frac{\sigma_h^2}{\sigma_f^2}\big)$ .*

---

**Note.** the condition $E^0 \leq \sigma_{f-h}^2/T$ can be ensured by using a batch size $T$ times larger to estimate $\boldsymbol{m}^0$, this will at most result in doubling the number of steps $T$. In practice, we did not need to ensure this condition. We also note the term $\sigma_h^2/K$, which corresponds to the error of solving the inner problem 4. Remarkably, although we are using SGD steps of the helper $h$, we get a $1/K$ error instead of $1/\sqrt{K}$; this can be explained by the fact that we initialize using the (approximate) solution of the last inner problem which accelerates convergence; we can potentially get faster rates by using variance reduction methods on $h$.

We will compare this rate to that of SGD under the same amount of work asked from $f$: $\mathcal{O}\Big(\sqrt{\frac{LF^0\sigma_f^2}{T}} + \frac{LF^0}{T}\Big)$

which corresponds to $\mathcal{O}\Big(\frac{LF^0\sigma_f^2}{\varepsilon^2} + \frac{LF^0}{\varepsilon}\Big)$ stochastic gradient calls of $f$ necessary to get an $\varepsilon$-stationary point $\hat{\boldsymbol{x}}$ in expectation i.e. a point $\hat{\boldsymbol{x}}$ such that $\mathbb{E}[\|\nabla f(\hat{\boldsymbol{x}})\|_2^2] \leq \varepsilon$ (the expectation is taken over the algorithm that generated $\hat{\boldsymbol{x}}$).

In comparison, based on Theorem 4.3, we can show the following corollary:

---

**Corollary 4.4** (Iteration complexity of AuxMOM)**.** *Let $\hat{x}$ be chosen uniformly at random from the iterates generated by AuxMOM. To guarantee $\mathbb{E}[\|\nabla f(\hat{\boldsymbol{x}})\|_2^2] \leq \varepsilon$, AuxMOM needs at most*

$$\mathcal{O}\Big(\frac{\delta F^0\sigma_{f-h}^2}{\varepsilon^2} + \frac{\delta F^0}{\varepsilon} + \frac{\sigma_{f-h}^2}{\varepsilon} + \frac{\sigma_h^2}{LF^0} + 1_{\sigma_h\neq 0}\frac{LF^0}{\varepsilon}\Big)$$

*(stochastic) gradient calls of $f$.*

---

In particular, we see that in the dominating order of $1/\varepsilon$, when access to gradients of $f$ is stochastic (**noisy case**), we replaced $L\sigma_f^2$ by $\delta\sigma_{f-h}^2$ which might be very small, either because $\delta \ll L$ or $\sigma_{f-h}^2 \ll \sigma_f^2$. In the **noiseless case** (when we have full access to gradients of $f$), we replaced $L$ by $\delta$.

Of course, the gain that we obtain is not for free; it comes at the cost of using $K = \mathcal{O}\Big(\frac{\sigma_h^2}{\varepsilon} + 1_{\delta\neq 0}\frac{L}{\delta} + 1\Big)$ inner steps of the helper $h$.

**Sanity checks.** For $h = f$, we have $\delta = 0$, we get the iteration complexity $\mathcal{O}\Big(\frac{\sigma^2}{\varepsilon} + \frac{LF^0}{\varepsilon}\Big)$ which corresponds to the rate of $KT$-steps of SGD. For $h = 0$, we have $\delta = L$; in this case, we don't gain anything as should be the case.

**SVRG in the non-convex setting.** In particular, because Theorem 4.3 also applies for the case where we have multiple helpers, and we sample each time $S = 1$ helpers, we get that SVRG converges in this case as $\mathcal{O}\big(\frac{(L+\delta K)F^0}{KT}\big)$, which matches the known SVRG rate (Reddi et al., 2016) (up to $\delta$ being small). More interestingly, we obtain the same convergence rate by using only one batch (no need to sample) if the batch is representative enough of the data (i.e., satisfies our Hessian similarity Assumption 3.3).

**Local steps help in federated learning.** By employing the decentralized version of this theorem (as detailed in Appendix C.4), we can ascertain that local steps (denoted as $K$ in our context) indeed provide a beneficial contribution to Federated Learning. This finding aligns with the results presented in (Karimireddy et al., 2020a).

### 4.3 MVR based approach

We consider now the instance of Algorithm 1, which uses the MVR momentum in in (6). The detailed algorithm can be found in the Appendix Algorithm 5.

**Stronger assumptions.** For the analysis of this variant, we need a stronger similarity assumption

**Assumption 4.5. Stronger Hessian similarity.**

$$\exists \delta \in [0, L] \; \forall \zeta_{f-h} \;\; : \;\; \boldsymbol{g}_{f-h}(\cdot, \zeta_{f-h}) \text{ is } \delta\text{-Lipschitz.}$$

Assumption 4.5 is stronger than its counterpart Assumption 3.3 used for AuxMOM. It is reasonable to need a stronger assumption since, already, when using the MVR momentum, a stronger smoothness assumption has to hold.

**Convergence of this variant.** We prove the following theorem that gives the convergence rate of this algorithm in the non-convex case.

---

**Theorem 4.6.** *Under assumptions A3.1, 3.2,4.5. For $\boldsymbol{m}^0$ such that $E^0 \leq \sigma_{f-h}^2/T$, for $a = \max(\frac{1}{T}, 1156\delta^2 K^2 \eta^2)$ and $\eta = \min\left(\frac{1}{L}, \frac{1}{192\delta K}, \frac{1}{K}\left(\frac{F^0}{18432\delta^2 T \sigma_{f-h}^2}\right)^{1/3}, \sqrt{\frac{F^0}{KT(L/2+8\delta K)}}\right)$. This choice gives us the rate :*

$$\frac{1}{KT}\sum_{t=1}^{T}\sum_{k=1}^{K-1}\mathbb{E}\big[\|\nabla f(\boldsymbol{y}_k^t)\|_2^2\big] = \mathcal{O}\Big(\big(\frac{\delta F^0 \sigma_{f-h}}{T}\big)^{2/3} + \sqrt{\frac{(L+\delta)F^0 \sigma_h^2}{KT}} + \frac{(L+\delta K)F^0}{KT} + \frac{\sigma_{f-h}^2}{T} + \frac{\sigma_h^2}{K}\Big).$$

---

**Baseline.** Under the same assumptions and for the same amount of work, MVR or STORM (Cutkosky & Orabona, 2019) has the rate: $\mathcal{O}\left(\big(\frac{LF^0 \sigma_f}{T}\big)^{2/3} + \frac{LF^0}{T}\right)$, this rate corresponds to needing at most $\mathcal{O}\left(\frac{LF^0 \sigma_f}{\varepsilon^{3/2}} + \frac{LF^0}{\varepsilon}\right)$ (stochastic) gradient calls of $f$ to reach a point $\hat{\boldsymbol{x}}$ such $\mathbb{E}[\|\nabla f(\hat{\boldsymbol{x}})\|_2^2] \leq \varepsilon$ or $\varepsilon$-stationary point. This rate/iteration complexity is known to be optimal under the strong smoothness assumption: $f(\cdot, \xi)$ is $L$-smooth for all $\xi$ almost surely.

Using Theorem 4.6, it is easy to show the following corollary:

---

**Corollary 4.7** (Iteration complexity of AuxMVR)**.** *Let $\hat{x}$ be chosen uniformly at random from the iterates generated by AuxMVR. To guarantee $\mathbb{E}[\|\nabla f(\hat{\boldsymbol{x}})\|_2^2] \leq \varepsilon$, AuxMVR needs at most*

$$\mathcal{O}\Big(\frac{\delta F^0 \sigma_{f-h}}{\varepsilon^{3/2}} + \frac{\delta F^0}{\varepsilon} + \frac{\sigma_{f-h}^2}{\varepsilon} + \frac{\sigma_h^2}{LF^0} + 1_{\sigma_h \neq 0}\frac{LF^0}{\varepsilon}\Big)$$

*(stochastic) gradient calls of $f$.*

---

The same conclusions as for AuxMOM are valid here. In the **noisy case**, we replaced $L\sigma_f^2$ by $\delta\sigma_{f-h}^2$ which might be very small, either because $\delta \ll L$ or $\sigma_{f-h}^2 \ll \sigma_f^2$. In the **noiseless case** (when we have full access to gradients of $f$), we replaced $L$ by $\delta$.

Again, this potential gain is obtained at the cost of using $K = \mathcal{O}\left(\frac{\sigma_h^2}{\varepsilon} + 1_{\delta \neq 0}\frac{L}{\delta} + 1\right)$ inner steps of the helper $h$.

# 5 Potential applications

The optimization with access to auxiliary information proposed is general enough that we can use it in many applications where we either have access to auxiliary information explicitly, such as in auxiliary learning or transfer learning, or implicitly, such as in semi-supervised learning. We present here a non-exhaustive list of potential applications.

**Reusing batches in SGD training and core-sets.** In Machine Learning, the empirical risk minimization consisting of minimizing a function of the form $f(\boldsymbol{x}) = \frac{1}{N}\sum_{i=1}^{N} L(\boldsymbol{x}, \xi_i)$ is ubiquitous. In many applications, we want to summarize the data-set $\{\xi_i\}_{i=1}^{N}$ by a smaller potentially weighted subset $CS = \{(\xi_{i_j}, w_j)\}_{j=1}^{M}$, for positive weights $(w_j)_{j=1}^{M}$ that add up to one, this is referred to as a core-set (Bachem et al., 2017). In this case we can set $h(\boldsymbol{x}) = \sum_{(w,\xi)\in CS} wL(\boldsymbol{x},\xi)$. An even sampler problem is when we sample a batch $B \subset \{\xi_i\}_{i=1}^{N}$ of size $b \leq N$, one question we can ask is how can we reuse this same batch to optimize $f$? In this case we set $h(\boldsymbol{x}) = (1/b)\sum_{\xi\in B} L(\boldsymbol{x},\xi)$. In the case where we have many batches $\{B_i\}_{i\in I}$, we can set $h_i(\boldsymbol{x}) = (1/|B_i|)\sum_{\xi\in B_i} L(\boldsymbol{x},\xi)$ for each $i \in I$ and use our decentralized framework to sample each time a helper $h$.

**Note.** In case $h$ is obtained using a subset $B$ of the dataset defining $f$, there is a-priori a trade-off between the similarity between $f$ and $h$ measured by the hessian similarity parameter $\delta(B)$ and the cheapness of the gradients of $h$. A-priori, the bigger the size of $B$ is, the easier it is to obtain a small $\delta(B)$, but the more expensive it is to compute the gradients of $h$.

**Semi-supervised learning.** In Semi-supervised Learning (Zhu, 2005), We have a small set of carefully cleaned data $\mathcal{Z}$ directly related to our target task and a large set of unlabeled data $\tilde{\mathcal{Z}}$. Let us also assume there is an auxiliary pre-training task defined over the source data, e.g., this can be the popular learning with contrastive loss (Chen et al., 2020). In this setting, we have a set of transformations $\mathcal{T}$ which preserves the semantics of the data, two unlabeled data samples $\tilde{\zeta}_1, \tilde{\zeta}_2 \in \tilde{\mathcal{Z}}$, and a feature extractor $\phi_{\boldsymbol{x}}(\cdot) : \tilde{\mathcal{Z}} \to \mathbb{R}^k$ parameterized by $\boldsymbol{x}$. Then, the contrastive loss is of the form: $\tilde{\ell}(\phi_{\boldsymbol{x}}(\tilde{\zeta}_1), \phi_{\boldsymbol{x}}(\mathcal{T}(\tilde{\zeta}_1)), \phi_{\boldsymbol{x}}(\tilde{\zeta}_2))$ where the loss tries to minimize the distance between the representations $\phi_{\boldsymbol{x}}(\tilde{\zeta}_1)$ and $\phi_{\boldsymbol{x}}(\mathcal{T}(\tilde{\zeta}_1))$, while simultaneously maximizing distance to $\phi_{\boldsymbol{x}}(\tilde{\zeta}_2)$. Similarly, we also have a target loss $\ell : \mathcal{Z} \to \mathbb{R}$, which we care about; then, we can define

$$f(\boldsymbol{x}) = \mathbb{E}_{\zeta\in\mathcal{Z}}\big[\ell(\boldsymbol{x};\zeta)\big]$$

and

$$h(\boldsymbol{x}) = \mathbb{E}_{\tilde{\zeta}_1,\tilde{\zeta}_2,\mathcal{T}}\big[\tilde{\ell}(\phi_{\boldsymbol{x}}(\tilde{\zeta}_1), \phi_{\boldsymbol{x}}(\mathcal{T}(\tilde{\zeta}_1)), \phi_{\boldsymbol{x}}(\tilde{\zeta}_2))\big].$$

Another way our framework can be used for semi-supervised learning is by endowing the unlabeled data with synthetic labels (either random labels as we will see for logistic regression or via Pseudo-labeling (Lee, 2013)) and defining the helper $h$ as the loss over the unlabeled data with its assigned labels.

**Transfer learning.** For a survey, see (Zhuang et al., 2020). In this case, in addition to a cleaned data set $\mathcal{Z}$ we have access to $\tilde{\mathcal{Z}}$ a pre-training source $\tilde{\mathcal{Z}}$. Given a fixed mask $M$ (for masking deep layers in a neural network) and a loss $\tilde{\ell}$, we set

$$f(\boldsymbol{x}) = \mathbb{E}_{\zeta\in\mathcal{Z}}\big[\ell(\boldsymbol{x};\zeta)\big] \quad \text{and} \quad h(\boldsymbol{x}) = \mathbb{E}_{\tilde{\zeta}\in\tilde{\mathcal{Z}}}\big[\tilde{\ell}(M \odot \boldsymbol{x}, \tilde{\zeta})\big].$$

**Federated learning.** Consider the problem of distributed/federated learning where data is decentralized over $N$ workers (McMahan et al., 2017b). Let $\{F_1(\boldsymbol{x}), \ldots, F_N(\boldsymbol{x})\}$ represent the $N$ loss functions defined over their respective local datasets. In such settings, communication is much more expensive (say $M$ times more expensive) than the minibatch gradient computation time (Karimireddy et al., 2018). In this case, we set

$$f(\boldsymbol{x}) = \frac{1}{N}\sum_{i=1}^{N} F_i(\boldsymbol{x}) \quad \text{and} \quad h_i(\boldsymbol{x}) = F_i(\boldsymbol{x}).$$

Thus, we want to minimize the target function $f(\boldsymbol{x})$ defined over all the workers' data but only have access to the cheap loss functions defined over local data. The main goal, in this case, is to limit the number of communications and use as many local steps as possible. Our proposed two variants are very close to MiMeSGDm and MiMeMVR from (Karimireddy et al., 2020a).

**Personalized learning.** This problem is a combination of the federated learning and transfer learning

problems described above and is closely related to multi-task learning (Ruder, 2017). Here, there are $N$ workers, each with a task $\{F_1(\boldsymbol{x}), \ldots, F_N(\boldsymbol{x})\}$, and without loss of generality, we describe the problem from the perspective of worker 1. In contrast to the delayed communication setting above, in this scenario, we only care about the *local loss* $F_1(\boldsymbol{x})$, whereas all the other worker training losses $\{F_2(\boldsymbol{x}), \ldots, F_N(\boldsymbol{x})\}$ constitute auxiliary data:

$$f(\boldsymbol{x}) = \big[ F_1(\boldsymbol{x}) = E_{\zeta_1}[F_1(\boldsymbol{x}; \zeta_1)] \big]$$

and for $i > 2$

$$h_i(\boldsymbol{x}) = \big[ F_i(\boldsymbol{x}) = \mathbb{E}_{\zeta_i}[F_i(\boldsymbol{x}; \zeta_i)] \big] .$$

In this setting, our main concern is the limited amount of training data available on any particular worker—if this was not an issue, we could have simply directly minimized the local loss $F_1(\boldsymbol{x})$.

**Training with compressed models.** Here, we want to train a large model parameterized by $\boldsymbol{x} \in \mathbb{R}^d$. To decrease the cost (both time and memory) of computing the backprop, we instead mask (delete) a large part of the parameters and perform backprop only on the remaining small subset of parameters (Sun et al., 2017; Yu & Huang, 2019). Suppose that our loss function $\ell$ is defined over sampled minibatches $\xi$ and parameters $\boldsymbol{x}$. Also, let us suppose we have access to some sparse/low-rank masks $\{\boldsymbol{1}_{\mathcal{M}_1}, \ldots, \boldsymbol{1}_{\mathcal{M}_k}\}$ from which we can choose. Then, we can define the problem as

$$f(\boldsymbol{x}) = \mathbb{E}_\xi\big[\ell(\boldsymbol{x}\,;\,\xi)\big] \quad \text{and} \quad h(\boldsymbol{x}) = \mathbb{E}_{\xi, \mathcal{M}}\big[\ell(\boldsymbol{1}_{\mathcal{M}} \odot \boldsymbol{x}\,;\,\xi)\big] .$$

Thus, to compute a minibatch stochastic gradient of $h(\boldsymbol{x})$ requires only storing and computing a significantly smaller model $\boldsymbol{1}_{\mathcal{M}} \odot \boldsymbol{x}$ where most of the parameters are masked out. Let $\boldsymbol{D}_{\mathcal{M}} = diag(\boldsymbol{1}_{\mathcal{M}})$ be a diagonal matrix with the same mask as $\boldsymbol{1}_{\mathcal{M}}$. The similarity condition then becomes

$$\|\nabla^2 f(\boldsymbol{x}) - \mathbb{E}_{\mathcal{M}}\big[\boldsymbol{D}_{\mathcal{M}} \nabla^2 f(\boldsymbol{1}_{\mathcal{M}} \odot \boldsymbol{x})\boldsymbol{D}_{\mathcal{M}}\big]\|_2 \leq \delta .$$

The quantity above first computes the Hessian on the masked parameters $\boldsymbol{1}_{\mathcal{M}} \odot \boldsymbol{x}$ and then is averaged over the various possible masks to compute $\mathbb{E}_{\mathcal{M}}\big[\boldsymbol{D}_{\mathcal{M}} \nabla^2 f(\boldsymbol{1}_{\mathcal{M}} \odot \boldsymbol{x})\boldsymbol{D}_{\mathcal{M}}\big]$. Thus, the parameter $\delta$ here measures the decomposability of the full Hessian $\nabla^2 f(\boldsymbol{x})$ along the different masked components. Again, we do not need the functions $f$ and $h$ to be related to each other in any other way beyond this condition on the Hessians—they may have completely different gradients and optimal parameters.

**Does the Hessian similarity hold in these examples?** In general, the answer is a no since already smoothness does not necessarily hold. Also, this will depend on the models we have and should be treated on a case-by-case basis. However, we can see from the experiment section that our algorithms perform well even for deep learning models. This work is a first attempt to unify these frameworks; we do not pretend that our answers or algorithms are the final attempt. We hope to spark further research in this direction, both theoretically and practically.

**Examples where it holds.** There are two special and simple examples where this similarity assumption holds. In both semi-supervised linear and logistic regressions, we note that the hessian does not depend on the label distribution; for this reason, we can endow the unlabeled data with any label distribution and construct the helper $h$ (based on unlabeled data) with $\delta = 0$ (under the assumption that unlabeled data come from the same distribution as that of labeled data).

## 6 Experiments

**Baselines.** We will consider fine-tuning and the naive approach as baselines. Fine-tuning is equivalent to using the gradients of the helper all at the beginning and then only using the gradients of the main objective $f$. We note that in our experiments, $K = 1$ corresponds to SGD with momentum; this means we are also comparing with SGDm.

### 6.1 Toy example

We consider a simple problem that consists in optimizing a function $f(x) = \frac{1}{2}x^2$ by enlisting the help of the function $h(x) = \frac{1}{2}(1 + \delta)(x - \zeta/(1 + \delta))^2$ for $x \in \mathbb{R}$.

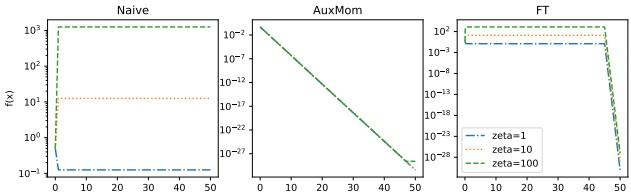

Figure 1: Effect of the bias $\zeta$ (zeta in the figure) on the naive approach (Naive), AuxMOM and Fine Tuning (FT) for $K = 10, \delta = 1$ and $\eta = \min(1/2, 1/(\delta K))$. We can see that the naive approach fails to converge for large bias values, whereas AuxMOM converges all the time, no matter the value of the bias. Fine Tuning converges much slower for small values of $\delta$, but beats AuxMOM for $\delta = 10$.

**Effect of the bias $\zeta$.** Figure 1 shows that indeed our algorithm **AuxMOM** does correct for the bias. We note that in this simple example, having a big value of $\zeta$ means that the gradients of $h$ point opposite to those of $f$, and hence, it's better to not use them in a naive way. However, our approach can correct for this automatically and hence does not suffer from increasing values of $\zeta$. In real-world data, it is very difficult to quantify $\zeta$, thus why we can still benefit a little bit (in non-extreme cases) using the naive way.

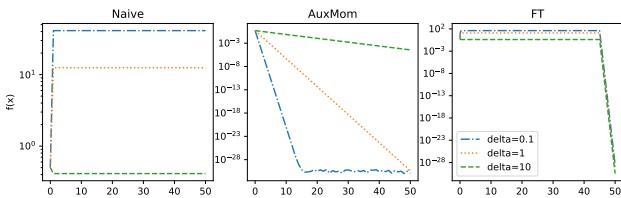

Figure 2: Effect of the similarity $\delta$ (delta in the figure) on both the naive approach (Naive), AuxMOM and Fine Tuning for $K = 10, \zeta = 10$ and $\eta = 0.5/(1 + \delta)$ for Naive and AuxMOM and $\eta = 0.5/(1 + \delta)$ then $eta = 0.5$ for Fine Tuning. We can see that the naive approach fails to benefit from small values of $\delta$; AuxMOM does not suffer from the same problem, whereas Fine Tuning is slower than AuxMOM.

**Effect of the similarity $\delta$.** Figure 2 shows how the three approaches compare when changing $\delta$. We note that $\delta = 0.1$ corresponds to $L/\delta = K = 10$ the value used in our experiment; our theory predicts that for such values of $\delta$, the convergence is as if we were using only gradients of $f$.

## 6.2 Leveraging noisy or mislabeled data

We show here how our approach can be used to leverage data with questionable quality, like the case where some of the inputs might be noisy or, in general, transformed in a way that does not preserve the labels. A second example is when part of the data is either unlabeled or has wrong labels.

**Rotated features.** We consider a simple feed-forward neural network ($Linear(28 * 28, 512) \rightarrow ReLU \rightarrow Linear(512, 512) \rightarrow ReLU \rightarrow Linear(512, 10)$) to classify the MNIST dataset (LeCun & Cortes, 2010) , which is the main task $f$. As a helper function $h$, we rotate MNIST images by a certain angle $\in \{0, 45, 90, 180\}$; the rotation plays the role of heterogeneity; it is worth noting that this is not simple data augmentation as numbers "meanings" are not conserved under such a transformation. We used a 256 batch size for $f$ and a 64 batch size for $h$; in our experiments changing the batch size of $h$ to 256 or 512 led to similar results. We plot the test accuracy that is obtained using both the naive approach and **AuxMOM**.

First of all, Figure 3 shows that we indeed benefit from using larger values of $K$ up to a certain level (this is predicted by our theory), this suggests that we have somehow succeeded in making a new gradient of $f$ out of each gradient of $h$ we had.

Next, Figure 4 shows how much the rotation angle affects the performances of the naive approach and AuxMOM. Astonishingly, AuxMOM seems to not suffer from increasing the value of the angle (which we should increase the bias).

Figure 5 shows a comparison with the fine-tuning approach as well.

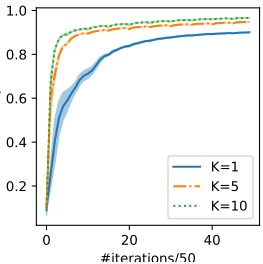

Figure 3: Effect of $K$ ($K-1$ is the number of times we use the helper h) on the test accuracy of the main task (for an angle = 45). We can see that our approach, as our theory predicts, benefits from bigger values of $K$.

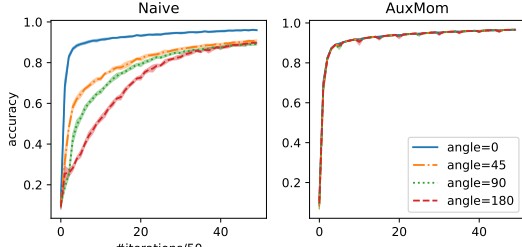

Figure 4: Test accuracy obtained using different angles as helpers, for $K = 10$, step size $\eta = 0.01$ and momentum parameter $a = 0.1$. We see that, astonishingly, **AuxMOM** does not suffer much from the change in the angle, whereas, as expected, the bigger the angle, the worse the accuracy on the main task for the naive approach.

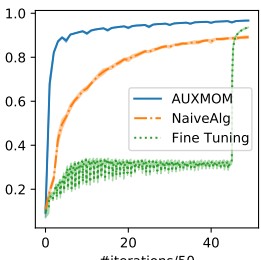

Figure 5: comparison of The Naive approach, AuxMOM, and Fine Tuning for an angle = 90. Again, we see that while not suffering from the added bias, Fine Tuning is slower than AuxMOM.

**Mislabeled data.** In a similar experiment, the helper $h$ is given again by MNIST images, but this time, we choose a wrong label for each image with a probability $p$. Figure 6 shows the results.

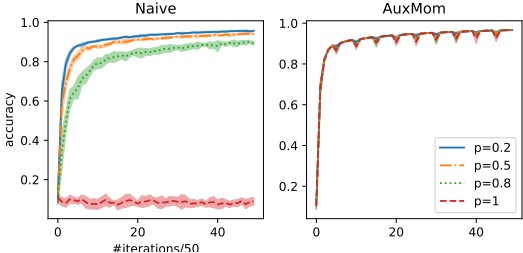

Figure 6: Test accuracy obtained using different probabilities $p$ as helpers, for $K = 10$, step size $\eta = 0.01$ and momentum parameter $a = 0.1$. Again, astonishingly, **AuxMOM** does not suffer much from the change in the angle, whereas, as expected, the bigger the angle, the worse the accuracy on the main task.

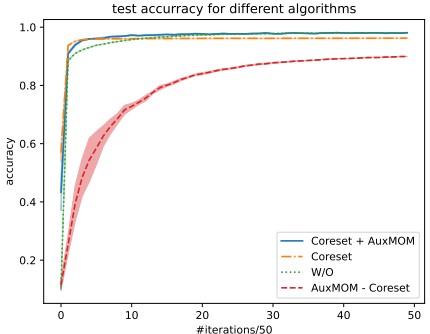

Figure 7: Comparing Coresets with and without AuxMOM. We see that with AuxMOM, we reach a higher accuracy, similar to using the true dataset (W/O).

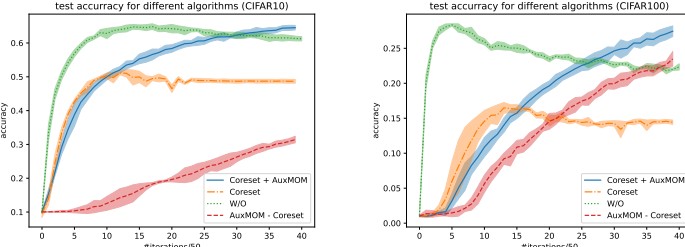

Figure 8: CIFAR10 and CIFAR100 experiments. Again, using AuxMOM leads to higher accuracy, similar to using the true dataset (W/O), whereas using the coreset alone leads to a loss in accuracy.

## 6.3  Training with Coresets

**MNIST experiment.** We consider again the MNIST dataset. This time, we randomly sample a subset that has one-fifth the size of the original training set; this subset will play the role of the coreset.

We compare running SGDm solely on the subset (referred to as Coreset) to training using AuxMOM with the coreset as a helper (referred to as Coreset + AuxMOM). We also consider as baselines running SGDm on the same total amount of work performed by AuxMOM and SGDm on the coreset (referred to as W/O) and running on the same amount of work done by the main function on AuxMOM (referred to as AuxMOM - Coreset).

The results of this experiment are shown in Figure 7. We can see that Coreset loses around 2% accuracy compared to W/O, whereas Coreset + AuxMOM reaches the same accuracy as W/O (it even beats it a little). Again, we see that AuxMOM bests training alone (AuxMOM - Coreset here).

**CIFAR10/100 experiments.** We performed the same experiments on the CIFAR10 and CIFAR100 datasets; again, we considered a randomly sampled coreset one-fifth the size of the original training set. We note that in these experiments, we used simple convolutional neural networks (two convolutions, followed each by a max-pooling layer, followed by three linear layers, to which a ReLu activation is applied, except for the last layer) and a constant step-size strategy, which shouldn't lead to state-of-the-art performance.

The results depicted in Figure 8 show that, similar to the experiment with the MNIST dataset, we observe that by using AuxMOM, we rival the performance of W/O; on the other hand, Coreset loses nearly 10% accuracy.

**Varying the size of the coreset.** We consider the CIFAR10 dataset and assess the performance of AuxMOM and training on the coreset alone for different coreset sizes (5%, 10%, and 25%). Figure 9 shows how AuxMOM is only influenced in that it needs more iterations to reach the same accuracy for a smaller

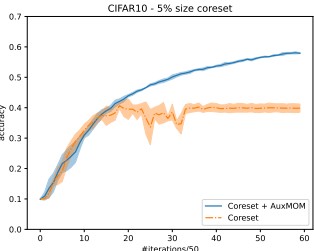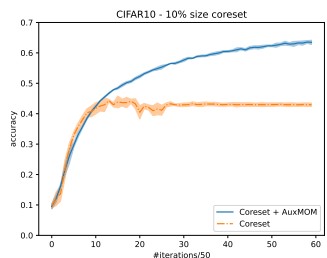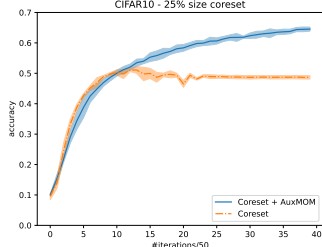

Figure 9: Effect of varying the coreset size on AuxMOM and pure coreset training. We see that the smaller the size of the coreset is, the worse the performance of training on the coreset alone; however, AuxMOM is only affected in terms of the number of iterations it needs to reach a given accuracy, which increases as the size of the coreset decreases.

coreset size, whereas training on the coreset alone is highly affected by the size of the corset as we see clearly that the smaller this size is, the lower the reached accuracy is.

**Overhead of AuxMOM over training alone.** The only additional time AuxMOM needs is the time necessary for sampling a new batch from the original dataset, computing its gradient, and updating the momentum used by AuxMOM; it should be negligible in practice.

## 6.4 Semi-supervised logistic regression

We consider a semi-supervised logistic regression task on the "Mushrooms" dataset from the libsvmtools repository (Chang & Lin, 2011), which has 8124 samples, each with 112 features. We divide this dataset into three equal parts: one for training and, one for testing, and the third one is unlabeled. In this context, the helper task $h$ is constituted of the unlabeled data to which we assigned random labels. Figure 10 shows indeed **AuxMOM** accelerates convergence on the training set. More importantly, it also leads to a smaller loss on the test set, which suggests a generalization benefit coming from using the unlabeled set.

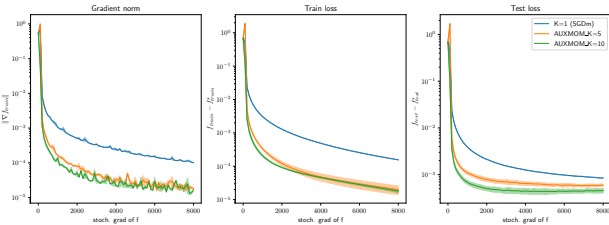

Figure 10: Gradient norm, train loss, and test loss. We set the parameter $a = 0.1$, and the stepsizes were optimized using grid-search for best training loss.

## 7 Discussion

**Fine Tuning.** Fine Tuning uses the helper gradients first and then uses the gradients of $f$; in this sense, fine-tuning has the advantage that it can be used for different functions $f$ without having to go through the first phase each time. AuxMOM seems to beat it (especially for small values of the similarity $\delta$); however, it does not enjoy the same advantage. Can we reconcile both worlds?
**Better measures of similarity?** Quantifying similarity between functions of the previous stochastic form is an open problem in Machine Learning that touches many domains such as Federated learning, transfer learning, curriculum learning, and continual learning. Knowing what measure is most appropriate for each case is an interesting problem. In this work, we don't pretend to solve the latter problem.
**Higher order strategies.** We believe our work can be "easily" extended to higher order strategies by using proxies of $f - h$ based on higher order derivatives. We expect that, in general, if we use a given Taylor approximation, we will need to make assumptions about its error. For example, if we use a 2nd order Taylor

approximation, we expect we will need to bound the difference of the third derivatives of $f$ and $h$; this will be similar to the analysis of the Newton algorithm with cubic regularization(Nesterov & Polyak, 2006).

**Dealing with the noise of the snapshot.** In this work, we proposed to deal with the noise of the snapshot gradient of $f - h$ using momentum, other approaches such as using batch sizes of varying sizes with training (typically a batch size that increases as convergence is near) are possible, such an approach was used in SCSG(Lihua et al.).

**Positively correlated noise.** We showed in this work that we might benefit if we could sample gradients of $h$ that are positively correlated with gradients of $f$, but we did not mention how this can be done. This is an interesting question that we intend to follow in the future.

## 8 Conclusion

We studied the general problem of optimizing a target function with access to a set of potentially decentralized helpers. Our framework is broad enough to recover many machine learning and optimization settings (ignoring the applicability of the similarity assumptions). While there are different ways of solving this problem in general, we proposed two variants **AuxMOM** and **AuxMVR** that we showed improve on known optimal convergence rates. We also showed how we could go beyond the bias correction that we have proposed; this can be potentially accomplished by using higher-order approximations of the difference between target and helper functions. Furthermore, we only considered the hessian similarity assumption in this work, but we think it is possible to use other similarity measures depending on the solved problem; finding such measures is outside of the scope of this work, but it might be a good future direction.

## 9 Acknowledgements

The authors would like to express their gratitude to Martin Jaggi and the MLO team at EPFL for their valuable discussions. Additionally, we appreciate the helpful feedback provided by the TMLR reviewers.

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

## A  Code

The code for our experiments is available at https://github.com/elmahdichayti/OptAuxInf.

## B  Basic lemmas

**Lemma B.1.** $\forall \boldsymbol{a}, \boldsymbol{b} \in \mathbb{R}^d, c > 0 \ : \ \|\boldsymbol{a} + \boldsymbol{b}\|_2^2 \leq (1 + c)\|\boldsymbol{a}\|_2^2 + (1 + \frac{1}{c})\|\boldsymbol{b}\|_2^2$ .

*Proof.* The difference of the two quantities above is exactly $\|\sqrt{c}\boldsymbol{a} - \frac{1}{\sqrt{c}}\boldsymbol{b}\|_2^2 \geq 0$ . □

**Lemma B.2.** $\forall N \in \mathrm{N}, \forall \boldsymbol{a}_1, \ldots, \boldsymbol{a}_N \in \mathbb{R}^d \ : \ \|\sum_{i=1}^N \boldsymbol{a}_i\|_2^2 \leq N \sum_{i=1}^N \|\boldsymbol{a}_i\|_2^2$ .

**Lemma B.3.** *It is common knowledge that if $f$ is $L$-smooth, .i.e. satisfies A3.1 then :*

$$\forall \boldsymbol{x}, \boldsymbol{y} \in \mathbb{R}^d \ : f(\boldsymbol{y}) - f(\boldsymbol{x}) \leq \nabla f(\boldsymbol{x})^\top (\boldsymbol{y} - \boldsymbol{x}) + \frac{L}{2}\|\boldsymbol{y} - \boldsymbol{x}\|_2^2 \,.$$

## C  Missing proofs

### C.1  Naive approach

As a reminder, the naive approach uses one unbiased gradient of $f$ at the beginning of each cycle followed by $K - 1$ unbiased gradients of $h$.

We prove the following theorem :

**Theorem C.1.** *Under Assumptions 3.1,3.2,4.1. Starting from $\boldsymbol{x}^0$ and using a step size $\eta$ we have :*

$$\frac{1}{2KT}\sum_{t=1}^T \mathbb{E}[\|\nabla f(\boldsymbol{x}^{t-1})\|_2^2] + \frac{1-m}{2KT}\sum_{t=1}^T \sum_{k=2}^K \mathbb{E}[\|\nabla f(\boldsymbol{y}_{k-1}^t)\|_2^2] \leq \frac{F^0}{KT\eta} + \frac{L\sigma^2}{2}\eta + \frac{K-1}{K}\zeta^2 \,.$$

*For $\sigma^2 = \frac{\sigma_f^2 + (K-1)\sigma_h^2}{K}$ the average variance, $F^0 = \mathbb{E}[f(\boldsymbol{x}^0)] - f^\star$.*

---

**Algorithm 2** Naive($f, h$)

---

**Require:** $\boldsymbol{x}_0, \boldsymbol{m}^0, \eta, T, K$
  **for** $t = 1$ to $T$ **do**
    Sample $\xi_f^t$; compute $\boldsymbol{g}_f(\boldsymbol{x}^{t-1}, \xi_f^t)$
    $\boldsymbol{m}^t = \boldsymbol{g}_f(\boldsymbol{x}^{t-1}, \xi_f^t)$
    $\boldsymbol{y}_0^t = \boldsymbol{x}^{t-1}$
    **for** $k = 0$ to $K - 1$ **do**
      **if** k=0 **then**
        $\boldsymbol{d}_k^t = \boldsymbol{m}^t$
      **else**
        Sample $\xi_h^{t,k}$; Compute $\boldsymbol{g}_h(\boldsymbol{y}_k^t, \xi_h^{t,k})$
        $\boldsymbol{d}_k^t = \boldsymbol{g}_h(\boldsymbol{y}_k^t, \xi_h^{t,k})$
      **end if**
      $\boldsymbol{y}_{k+1}^t = \boldsymbol{y}_k^t - \eta \boldsymbol{d}_k^t$
    **end for**
    Update $\boldsymbol{x}^t = \boldsymbol{y}_K^t$
  **end for**

---

*Proof.* The proof is based on biased SGD theory. From (Ajalloeian & Stich, 2020) we have that if we the gradients $\boldsymbol{g}(\boldsymbol{x})$ that we are using are such that :

$$\mathbb{E}[\boldsymbol{g}(\boldsymbol{x})] = \nabla f(\boldsymbol{x}) + \boldsymbol{b}(\boldsymbol{x})$$

For a target function $f$ and a gradient bias $\boldsymbol{b}(\boldsymbol{x})$. Then we have for $\eta \leq \frac{1}{L}$:

$$f(\boldsymbol{y}_{k+1}^t) - f(\boldsymbol{y}_k^t) \leq \frac{\eta}{2}\big(-\|\nabla f(\boldsymbol{y}_k^t)\|_2^2 + \|\boldsymbol{b}(\boldsymbol{y}_k^t)\|_2^2\big) + \frac{L\eta^2}{2}\sigma_k^2$$

In our case, by Assumption4.1 : $\|\boldsymbol{b}(\boldsymbol{y}_k^t)\|_2^2 \leq \big(\zeta^2 + m\|\nabla f(\boldsymbol{y}_k^t)\|_2^2\big)1_{k>0}$ and $\sigma_k^2 = 1_{k=0}\sigma_f^2 + 1_{k>0}\sigma_h^2$. Summing the above inequality from $t = 1, k = 0$ to $t = T, k = K - 1$ and then dividing by $KT$ we get the statement of the theorem. $\square$

**Lower bound.** It is not difficult to prove that we cannot do better using the naive strategy. We can for example pick in $1d$ a target function $f(x) = \frac{1}{2}x^2$ and an auxiliary function $h(x) = \frac{1}{2}(x - \zeta)^2$. Both functions are 1-smooth and satisfy hessian similarity with $\delta = 0$, however, $\zeta$ is not (necessarily zero), in particular, this shows that hessian similarity (Assumption3.3) and gradient dissimilarity (Assumption4.1) are orthogonal. To show the lower bound we can consider the perfect case where we have access to full gradients of $f$ and $h$.

The dynamics of the naive approach can be written in the form:

$$\boldsymbol{x}^{t+1} = (1 - \eta)\boldsymbol{x}^t + \eta\zeta 1_{t \neq 0 mod(K)}$$

Which implies

$$\boldsymbol{x}^t = (1 - \eta)^t \boldsymbol{x}^0 + \zeta \sum_{i=0}^{t-1} \eta(1 - \eta)^{t-i-1} 1_{i \neq 0 mod(K)}$$

$$= (1 - \eta)^t \boldsymbol{x}^0 + \Omega(\zeta)$$

This sequence does not converge to zero no matter the choice of $\eta < 1$.

### C.2 Momentum with auxiliary information

#### C.2.1 Algorithm description

The algorithm that we proposed proceeds in cycles, at the beginning of each cycle we have $\boldsymbol{x}^{t-1}$ and take $K$ iterations of the form $\boldsymbol{y}_k^t = \boldsymbol{y}_{k-1}^t - \eta \boldsymbol{d}_k^t$ where $\boldsymbol{y}_0^t = \boldsymbol{x}^{t-1}$, $\boldsymbol{d}_k^t = \boldsymbol{g}_h(\boldsymbol{y}_{k-1}^t, \xi_{h,k}^t) + \boldsymbol{m}^t$ and $\boldsymbol{m}^t = (1-a)\boldsymbol{m}^{t-1} + a\boldsymbol{g}_{f-h}(\boldsymbol{x}^{t-1}, \xi_{f-h}^{t-1})$. Then we set $\boldsymbol{x}^t = \boldsymbol{y}_K^t$.

---

**Algorithm 3** AuxMOM($f, h$)

---

**Require:** $\boldsymbol{x}_0, \boldsymbol{m}^0, \eta, a, T, K$
  **for** $t = 1$ to $T$ **do**
    Sample $\xi_{f-h}^t$; compute $\boldsymbol{g}_{f-h}(\boldsymbol{x}^{t-1}, \xi_{f-h}^{t-1})$
    $\boldsymbol{m}^t = (1-a)\boldsymbol{m}^{t-1} + a\boldsymbol{g}_{f-h}(\boldsymbol{x}^{t-1}, \xi_{f-h}^{t-1})$
    $\boldsymbol{y}_0^t = \boldsymbol{x}^{t-1}$
    **for** $k = 0$ to $K - 1$ **do**
      Sample $\xi_h^{t,k}$; Compute $\boldsymbol{g}_h(\boldsymbol{y}_k^t, \xi_h^{t,k})$
      $\boldsymbol{d}_k^t = \boldsymbol{g}_h(\boldsymbol{y}_k^t, \xi_h^{t,k}) + \boldsymbol{m}^t$
      $\boldsymbol{y}_{k+1}^t = \boldsymbol{y}_k^t - \eta \boldsymbol{d}_k^t$
    **end for**
    Update $\boldsymbol{x}^t = \boldsymbol{y}_K^t$
  **end for**

---

We will introduce $\bar{\boldsymbol{d}}_k^t = \nabla h(\boldsymbol{y}_{k-1}^t) + \boldsymbol{m}^t$ and we note that $\mathbb{E}[\boldsymbol{d}_k^t | \boldsymbol{y}_{k-1}^t] = \bar{\boldsymbol{d}}_k^t$ and $\mathbb{E}[\|\boldsymbol{d}_k^t - \bar{\boldsymbol{d}}_k^t\|_2^2] \leq \sigma_h^2$. We will be using this fact and sometimes we will condition on $\boldsymbol{y}_{k-1}^t$ implicitly to replace $\boldsymbol{d}_k^t$ by $\bar{\boldsymbol{d}}_k^t$.

**Note.** Algorithm 4 is another version of AuxMOM that works very well in practice and seems more robust to the choice of the step size than Algorithm 3. The main difference with Algorithm 3 is that the momentum is applied on $f$ only. For this Algorithm, we could not prove any benefit from the similarity $\delta$.

---

**Algorithm 4** AuxMOM($f, h$) $- V0$

---

**Require:** $\boldsymbol{x}_0, \boldsymbol{m}^0, \eta, a, T, K$
  **for** $t = 1$ to $T$ **do**
    Sample $\xi_f^t$; compute $\boldsymbol{g}_f(\boldsymbol{x}^{t-1}, \xi_f^{t-1})$
    $\boldsymbol{m}^t = (1-a)\boldsymbol{m}^{t-1} + a\boldsymbol{g}_f(\boldsymbol{x}^{t-1}, \xi_f^{t-1})$
    $\boldsymbol{y}_0^t = \boldsymbol{x}^{t-1}$
    **for** $k = 0$ to $K - 1$ **do**
      Sample $\xi_h^{t,k}$; Compute $\boldsymbol{g}_h(\boldsymbol{y}_k^t, \xi_h^{t,k}), \boldsymbol{g}_h(\boldsymbol{x}^{t-1}, \xi_h^{t,k})$
      $\boldsymbol{d}_k^t = \boldsymbol{g}_h(\boldsymbol{y}_k^t, \xi_h^{t,k}) - \boldsymbol{g}_h(\boldsymbol{x}^{t-1}, \xi_h^{t,k}) + \boldsymbol{m}^t$
      $\boldsymbol{y}_{k+1}^t = \boldsymbol{y}_k^t - \eta \boldsymbol{d}_k^t$
    **end for**
    Update $\boldsymbol{x}^t = \boldsymbol{y}_K^t$
  **end for**

---

#### C.2.2 Convergence rate

We prove the following theorem that gives the convergence rate of this algorithm in the non-convex case.

**Theorem C.2.** *Under assumptions A3.1, 3.2, 3.3. For $a = 36\delta K\eta$ and $\eta = \min(\frac{1}{L}, \frac{1}{144\delta K}, \sqrt{\frac{\tilde{F}}{128L\beta KT\sigma_f^2}})$.*

*This choice gives us the rate :*

$$\frac{1}{8KT}\sum_{t=1}^{T}\sum_{k=0}^{K-1}\mathbb{E}\big[\|\nabla f(\boldsymbol{y}_k^t)\|_2^2\big] \leq 24\sqrt{\frac{L\beta\tilde{F}\sigma_f^2}{T}} + \frac{(L+192\delta K)\tilde{F}}{KT} + \frac{\sigma_h^2}{6K}.$$

*where $\tilde{F} = F^0 + \frac{E^0}{8\delta}$ and $\beta = \frac{\delta}{L}\left(\frac{\sigma_{f-h}^2}{\sigma_f^2} + \frac{1}{18K}\frac{\sigma_h^2}{\sigma_f^2}\right) + \frac{1}{288K}\frac{\sigma_h^2}{\sigma_f^2}$, $F^0 = f(\boldsymbol{x}^0) - f^\star$ and $E^0 = \mathbb{E}[\|\boldsymbol{m}^0 - \nabla f(\boldsymbol{x}^0) + \nabla h(\boldsymbol{x}^0)\|_2^2]$.*

*Furthermore, if we use a batch-size $T$ times bigger for computing an estimate $\boldsymbol{m}^0$ of $\nabla f(\boldsymbol{x}^0) - \nabla h(\boldsymbol{x}_0)$, then by taking $a = \max(\frac{1}{T}, 36\delta K\eta)$ and replacing $\tilde{F}$ by $F^0$ in the expression of $\eta$, we get :*

$$\eta = \min(\frac{1}{L}, \frac{1}{192\delta K}, \sqrt{\frac{F^0}{144L\beta K^2 T\sigma_f^2}}).$$

*we get the following :*

$$\frac{1}{8KT}\sum_{t=1}^{T}\sum_{k=0}^{K-1}\mathbb{E}\big[\|\nabla f(\boldsymbol{y}_k^t)\|_2^2\big] \leq 24\sqrt{\frac{L\beta F^0\sigma_f^2}{T}} + \frac{(L+192\delta K)F^0}{KT} + \frac{8\sigma_{f-h}^2}{T} + \frac{\sigma_h^2}{6K}.$$

### C.2.3 Bias in helpers updates and Notation

**Lemma C.3.** *Under the $\delta$-Bounded Hessian Dissimilarity assumption, we have :*
$$\forall \boldsymbol{x}, \boldsymbol{y} \in \mathbb{R}^d \quad : \|\nabla h(\boldsymbol{y}) - \nabla h(\boldsymbol{x}) - \nabla f(\boldsymbol{y}) + \nabla f(\boldsymbol{x})\|^2 \leq \delta^2\|\boldsymbol{y} - \boldsymbol{x}\|^2.$$

*Proof.* A simple application of the $\delta$-Bounded Hessian Dissimilarity assumption. $\qquad\square$

**Notation :** We will use the following notations : $\boldsymbol{e}^t = \boldsymbol{m}^t - \nabla f(\boldsymbol{x}^{t-1}) + \nabla h(\boldsymbol{x}^{t-1})$ to denote the momentum error and $E^t = \mathbb{E}[\|\boldsymbol{e}^t\|_2^2]$ to denote its expected squared norm. $\Delta_k^t = \mathbb{E}[\|\boldsymbol{y}_k^t - \boldsymbol{x}^{t-1}\|^2]$ will denote the progress made up to the k-th round of the t-th cycle, and for the progress in a whole cycle we will use $\Delta^t = \mathbb{E}[\|\boldsymbol{x}^t - \boldsymbol{x}^{t-1}\|^2]$. We also denote $\bar{\boldsymbol{d}}_k^t = \boldsymbol{d}_k^t - \boldsymbol{g}_h(\boldsymbol{y}_k^t, \xi_h^{t,k}) + \nabla h(\boldsymbol{y}_{k-1}^t) = \nabla h(\boldsymbol{y}_{k-1}^t) + \boldsymbol{m}^t$.

### C.2.4 Change during each cycle

**Variance of $\bar{\boldsymbol{d}}_k^t$.** We would like $\bar{\boldsymbol{d}}_k^t$ to be $\nabla f(\boldsymbol{y}_{k-1}^t)$, but it is not. In the next lemma, we control the error resulting from these two quantities being different.

**Lemma C.4.** *Under assumption A3.3, we have the following inequality :*
$$\mathbb{E}[\|\bar{\boldsymbol{d}}_k^t - \nabla f(\boldsymbol{y}_{k-1}^t)\|_2^2] \leq 2\delta^2\Delta_{k-1}^t + 2E^t$$

*Proof.* We have

$$\begin{aligned}
\bar{\boldsymbol{d}}_k^t - \nabla f(\boldsymbol{y}_{k-1}^t) &= \nabla h(\boldsymbol{y}_{k-1}^t) - \nabla f(\boldsymbol{y}_{k-1}^t) + \boldsymbol{m}^t \\
&= h(\boldsymbol{y}_{k-1}^t) - \nabla f(\boldsymbol{y}_{k-1}^t) - \nabla f(\boldsymbol{y}_{k-1}^t) + \nabla f(\boldsymbol{x}^{t-1}) + \boldsymbol{m}^t - \nabla f(\boldsymbol{x}^{t-1}) + \nabla h(\boldsymbol{x}^{t-1}) \\
&= h(\boldsymbol{y}_{k-1}^t) - \nabla f(\boldsymbol{y}_{k-1}^t) - \nabla f(\boldsymbol{y}_{k-1}^t) + \nabla f(\boldsymbol{x}^{t-1}) + \boldsymbol{e}^t
\end{aligned}$$

Using Lemma B.1 with $c = 1$, we get :

$$\begin{aligned}
\mathbb{E}[\|\bar{\boldsymbol{d}}_k^t - \nabla f(\boldsymbol{y}_{k-1}^t)\|_2^2] &\leq 2\mathbb{E}[\|h(\boldsymbol{y}_{k-1}^t) - \nabla f(\boldsymbol{y}_{k-1}^t) - \nabla f(\boldsymbol{y}_{k-1}^t) + \nabla f(\boldsymbol{x}^{t-1})\|_2^2] + 2\mathbb{E}[\|\boldsymbol{e}^t\|_2^2] \\
&\leq 2\delta^2\|\boldsymbol{y}_{k-1}^t - \boldsymbol{x}^{t-1}\|_2^2 + 2\mathbb{E}[\|\boldsymbol{e}^t\|_2^2] \\
&= 2\delta^2\Delta_{k-1}^t + 2E^t
\end{aligned}$$

$\square$

**Distance moved in each step.**

**Lemma C.5.** *for $\eta \leq \frac{1}{5\delta K}$ we have:*

$$\Delta_k^t \leq (1 + \frac{1}{K})\Delta_{k-1}^t + 12K\eta^2 E^t + 6K\eta^2 \mathbb{E}[\|\nabla f(\boldsymbol{y}_{k-1}^t)\|_2^2] + \eta^2 \sigma_h^2 \,.$$

*Proof.*

$$\begin{aligned}
\Delta_k^t &= \mathbb{E}[\|\boldsymbol{y}_k^t - \boldsymbol{x}^{t-1}\|_2^2] \\
&= \mathbb{E}[\|\boldsymbol{y}_{k-1}^t - \eta \boldsymbol{d}_k^t - \boldsymbol{x}^{t-1}\|_2^2] \\
&= \mathbb{E}[\|\boldsymbol{y}_{k-1}^t - \eta \bar{\boldsymbol{d}}_k^t - \boldsymbol{x}^{t-1}\|_2^2] + \eta^2 \mathbb{E}[\|\boldsymbol{d}_k^t - \bar{\boldsymbol{d}}_k^t\|_2^2] \\
&\leq (1 + \frac{1}{2K})\Delta_{k-1}^t + (2K+1)\eta^2 \mathbb{E}[\|\bar{\boldsymbol{d}}_k^t\|_2^2] + \eta^2 \sigma_h^2 \\
&= (1 + \frac{1}{2K})\Delta_{k-1}^t + 3K\eta^2 \mathbb{E}[\|\bar{\boldsymbol{d}}_k^t \pm \nabla f(\boldsymbol{y}_{k-1}^t)\|_2^2] + \eta^2 \sigma_h^2 \\
&\leq (1 + \frac{1}{2K})\Delta_{k-1}^t + 6K\eta^2 \mathbb{E}[\|\bar{\boldsymbol{d}}_k^t - \nabla f(\boldsymbol{y}_{k-1}^t)\|_2^2] + 6K\eta^2 \mathbb{E}[\|\nabla f(\boldsymbol{y}_{k-1}^t)\|_2^2] + \eta^2 \sigma_h^2 \\
&\leq (1 + \frac{1}{2K})\Delta_{k-1}^t + 6K\eta^2 (2\delta^2 \Delta_{k-1}^t + 2E^t) + 6K\eta^2 \mathbb{E}[\|\nabla f(\boldsymbol{y}_{k-1}^t)\|_2^2] + \eta^2 \sigma_h^2 \\
&= (1 + \frac{1}{2K} + 12\delta^2 K\eta^2)\Delta_{k-1}^t + 12K\eta^2 E^t + 6K\eta^2 \mathbb{E}[\|\nabla f(\boldsymbol{y}_{k-1}^t)\|_2^2] + \eta^2 \sigma_h^2
\end{aligned}$$

The condition $\eta \leq \frac{1}{5\delta K}$ ensures $12\delta^2 K\eta^2 \leq \frac{1}{2K}$ which finishes the proof.

$\square$

**Progress in one step.**

**Lemma C.6.** *For $\eta \leq \min(\frac{1}{L}, \frac{1}{192\delta K})$, under assumptions A3.1 and A3.3, the following inequality is true:*

$$\begin{aligned}
\mathbb{E}[f(\boldsymbol{y}_k^t) + \delta(1 + \frac{2}{K})^{K-k}\Delta_k^t] &\leq \mathbb{E}[f(\boldsymbol{y}_{k-1}^t) + \delta(1 + \frac{2}{K})^{K-(k-1)}\Delta_{k-1}^t] \\
&\quad - \frac{\eta}{4}\mathbb{E}[\|\nabla f(\boldsymbol{y}_{k-1}^t)\|_2^2] + 2\eta E^t + (\frac{L}{2} + 8\delta)\eta^2 \sigma_h^2 \,.
\end{aligned}$$

*Proof.* The $L$-smoothness of $f$ guarantees :

$$f(\boldsymbol{y}_k^t) - f(\boldsymbol{y}_{k-1}^t) \leq -\eta \nabla f(\boldsymbol{y}_{k-1}^t)^\top \boldsymbol{d}_k^t + \frac{L\eta^2}{2}\|\boldsymbol{d}_k^t\|_2^2 \,.$$

By taking expectation conditional to the knowledge of $\boldsymbol{y}_{k-1}^t$ we have:

$$\mathbb{E}[f(\boldsymbol{y}_k^t) - f(\boldsymbol{y}_{k-1}^t)] \leq -\eta \mathbb{E}[\nabla f(\boldsymbol{y}_{k-1}^t)^\top \bar{\boldsymbol{d}}_k^t] + \frac{L\eta^2}{2}\mathbb{E}[\|\bar{\boldsymbol{d}}_k^t\|_2^2] + \frac{L\eta^2}{2}\sigma_h^2 \,.$$

Using the identity $-2ab = (a-b)^2 - a^2 - b^2$, we have :

$$E[f(\boldsymbol{y}_k^t) - f(\boldsymbol{y}_{k-1}^t)] \leq -\frac{\eta}{2}E[\|\nabla f(\boldsymbol{y}_{k-1}^t)\|_2^2] + \frac{\eta}{2}E[\|\boldsymbol{d}_k^t - \nabla f(\boldsymbol{y}_{k-1}^t)\|_2^2] + \frac{L\eta^2 - \eta}{2}E[\|\boldsymbol{d}_k^t\|_2^2] + \frac{L\eta^2}{2}\sigma_h^2 \,.$$

Using $\eta \leq \frac{1}{L}$ we can get rid of the last term in the above inequality.

Using LemmaC.4, we get :

$$\mathbb{E}[f(\boldsymbol{y}_k^t) - f(\boldsymbol{y}_{k-1}^t)] \le -\frac{\eta}{2}\mathbb{E}\big[\|\nabla f(\boldsymbol{y}_{k-1}^t)\|_2^2\big] + \frac{\eta}{2}\mathbb{E}\big[\|\boldsymbol{d}_k^t - \nabla f(\boldsymbol{y}_{k-1}^t)\|_2^2\big] + \frac{L\eta^2}{2}\sigma_h^2$$

$$\le -\frac{\eta}{2}\mathbb{E}\big[\|\nabla f(\boldsymbol{y}_{k-1}^t)\|_2^2\big] + \frac{\eta}{2}(2\delta^2\Delta_{k-1}^t + 2E^t) + \frac{L\eta^2}{2}\sigma_h^2$$

$$= \delta^2\eta\Delta_{k-1}^t + \eta E^t - \frac{\eta}{2}\mathbb{E}\big[\|\nabla f(\boldsymbol{y}_{k-1}^t)\|_2^2\big] + \frac{L\eta^2}{2}\sigma_h^2 .$$

Now we multiply LemmaC.5 by $\delta(1+\frac{2}{K})^{K-k}$. Note that $1 \le (1+\frac{2}{K})^{K-k} \le 8$.

$$\delta(1+\frac{2}{K})^{K-k}\Delta_k^t \le \delta(1+\frac{2}{K})^{K-k}\big((1+\frac{1}{K})\Delta_{k-1}^t + 12K\eta^2 E^t + 6K\eta^2\mathbb{E}[\|\nabla f(\boldsymbol{y}_{k-1}^t)\|_2^2]\big)$$

$$+ \delta(1+\frac{2}{K})^{K-k}\eta^2\sigma_h^2$$

$$\le \delta(1+\frac{2}{K})^{K-(k-1)}\Delta_{k-1}^t - \frac{\delta}{K}(1+\frac{2}{K})^{K-k}\Delta_{k-1}^t + 96K\delta\eta^2 E^t$$

$$+ 48K\delta\eta^2\mathbb{E}[\|\nabla f(\boldsymbol{y}_{k-1}^t)\|_2^2] + 8\delta\eta^2\sigma_h^2$$

$$\le \delta(1+\frac{2}{K})^{K-(k-1)}\Delta_{k-1}^t - \frac{\delta}{K}\Delta_{k-1}^t + 96K\delta\eta^2 E^t$$

$$+ 48K\delta\eta^2\mathbb{E}[\|\nabla f(\boldsymbol{y}_{k-1}^t)\|_2^2] + 8\delta\eta^2\sigma_h^2$$

Adding the last two inequalities, we get :

$$\mathbb{E}[f(\boldsymbol{y}_k^t)] + \delta(1+\frac{2}{K})^{K-k}\Delta_k^t \le \mathbb{E}[f(\boldsymbol{y}_{k-1}^t)] + \delta(1+\frac{2}{K})^{K-(k-1)}\Delta_{k-1}^t$$

$$+ (\delta^2\eta - \frac{\delta}{K})\Delta_{k-1}^t$$

$$+ (\eta + 96K\delta\eta^2)E^t$$

$$+ (-\frac{\eta}{2} + 48K\delta\eta^2)\mathbb{E}[\|\nabla f(\boldsymbol{y}_{k-1}^t)\|_2^2]$$

$$+ (L/2 + 8\delta)\eta^2\sigma_h^2$$

For $\eta \le \frac{1}{192\delta K}$ we have $\delta^2\eta - \frac{\delta}{K} \le 0$, $\eta + 96K\delta\eta^2 \le 2\eta$ and $-\frac{\eta}{2} + 48K\delta\eta^2 \le -\frac{\eta}{4}$ which gives the lemma. $\quad\square$

**Distance moved in a cycle.**

**Lemma C.7.** *For $\eta \le \frac{1}{5K\delta}$ and under assumptions A3.1 and A3.3 with $G^t = \frac{1}{K}\sum_{k=0}^{K-1}\mathbb{E}\big[\|\nabla f(\boldsymbol{y}_k^t)\|_2^2\big]$, we have :*

$$\Delta^t\Big( := \mathbb{E}\big[\|\boldsymbol{x}^t - \boldsymbol{x}^{t-1}\|_2^2\big]\Big) \le 36K^2\eta^2 E^t + 18K^2\eta^2 G^t + 3K\eta^2\sigma_h^2 .$$

*Proof.* We use the fact $\boldsymbol{x}^t = \boldsymbol{y}_K^t$, which means $\Delta^t = \Delta_K^t$. The recurrence established in LemmaC.5 implies :

$$\Delta^t \le \sum_{k=1}^K (1+\frac{1}{K})^{K-k}\Big(12K\eta^2 E^t + 6K\eta^2\mathbb{E}[\|\nabla f(\boldsymbol{y}_{k-1}^t)\|_2^2] + \eta^2\sigma_h^2\Big)$$

$$\le 36K^2\eta^2 E^t + 18K^2\eta^2\frac{1}{K}\sum_k \mathbb{E}\big[\|\nabla f(\boldsymbol{y}_k^t)\|_2^2\big] + 3K\eta^2\sigma_h^2$$

$$= 36K^2\eta^2 E^t + 18K^2\eta^2 G^t + 3K\eta^2\sigma_h^2 .$$

Where we used the fact that $(1+\frac{1}{K})^{K-k} \le 3$. $\quad\square$

**Momentum variance.** Here we will bound the quantity $E^t$.

**Lemma C.8.** *Under assumptions A3.1,A3.2 and for $a \geq 12K\delta\eta$ , we have :*

$$E^t \leq (1 - \frac{a}{2})E^{t-1} + \frac{36\delta^2 K^2 \eta^2}{a}G^{t-1} + a^2 \sigma_{f-h}^2 + \frac{a}{24K}\sigma_h^2 .$$

*Proof.*

$$
\begin{aligned}
E^t &= \mathbb{E}\big[\|\boldsymbol{m}^t - \nabla f(\boldsymbol{x}^{t-1}) + \nabla h(\boldsymbol{x}^{t-1})\|_2^2\big] \\
&= \mathbb{E}\big[\|(1-a)(\boldsymbol{m}^{t-1} - \nabla f(\boldsymbol{x}^{t-1}) + \nabla h(\boldsymbol{x}^{t-1})) + a(\boldsymbol{g}_f(\boldsymbol{x}^{t-1}) + \boldsymbol{g}_h(\boldsymbol{x}^{t-1}) - \nabla f(\boldsymbol{x}^{t-1}) + \nabla h(\boldsymbol{x}^{t-1}))\|_2^2\big] \\
&\leq (1-a)^2 \mathbb{E}\big[\|\boldsymbol{m}^{t-1} \pm (\nabla f(\boldsymbol{x}^{t-2}) - \nabla h(\boldsymbol{x}^{t-2})) - (\nabla f(\boldsymbol{x}^{t-1}) - \nabla f(\boldsymbol{x}^{t-1}))\|_2^2\big] + a^2 \sigma_{f-h}^2 \\
&\leq (1-a)^2(1+a/2)E^{t-1} + (1-a)^2(1+2/a)\mathbb{E}\big[\|\nabla f(\boldsymbol{x}^{t-2}) - \nabla h(\boldsymbol{x}^{t-2}) - \nabla f(\boldsymbol{x}^{t-1}) + \nabla h(\boldsymbol{x}^{t-1})\|_2^2\big] \\
&\quad + a^2 \sigma_{f-h}^2 \\
&\leq (1-a)E^{t-1} + \frac{2\delta^2}{a}\Delta^{t-1} + a^2 \sigma_{f-h}^2
\end{aligned}
$$

Where we used the L-smoothness of $f$ to get the last inequality. We have also used the inequalities $(1-a)^2(1+a/2) \leq (1-a)$ and $(1-a)^2(1+2/a) \leq 2/a$ true for all $a \in (0,1]$.

We can now use LemmaC.7 to have :

$$E^t \leq (1 - a + \frac{72\delta^2 K^2 \eta^2}{a})E^{t-1} + \frac{36\delta^2 K^2 \eta^2}{a}G^{t-1} + a^2 \sigma_{f-h}^2 + \frac{6K\delta^2 \eta^2}{a}\sigma_h^2$$

By taking $a \geq 12\delta K\eta$ we ensure $\frac{72\delta^2 K^2 \eta^2}{a} \leq \frac{a}{2}$, So :

$$E^t \leq (1 - \frac{a}{2})E^{t-1} + \frac{36\delta^2 K^2 \eta^2}{a}G^{t-1} + a^2 \sigma_{f-h}^2 + \frac{a}{24K}\sigma_h^2$$

$\square$

**Progress in one round.**

**Lemma C.9.** *Under the same assumptions as in LemmaC.6, we have :*

$$\frac{\eta}{4}G^t \leq \frac{F^{t-1} - F^t}{K} - \frac{\delta}{K}\Delta^t + 2\eta E^t + (\frac{L}{2} + 8\delta)\eta^2 \sigma_h^2 .$$

*Where $F^t = \mathbb{E}[f(\boldsymbol{x}^t)] - f^\star$.*

*Proof.* We use the inequality established in LemmaC.6, which can be rearranged in the following way :

$$
\begin{aligned}
\frac{\eta}{4}\mathbb{E}[\|\nabla f(\boldsymbol{y}_{k-1}^t)\|_2^2] &\leq \mathbb{E}\big[f(\boldsymbol{y}_{k-1}^t) + \delta(1 + \frac{2}{K})^{K-(k-1)}\Delta_{k-1}^t\big] - \Big(\mathbb{E}\big[f(\boldsymbol{y}_k^t) + \delta(1 + \frac{2}{K})^{K-k}\Delta_k^t\big]\Big) \\
&\quad + 2\eta E^t + (\frac{L}{2} + 8\delta)\eta^2 \sigma_h^2 .
\end{aligned}
$$

We sum this inequality from $k = 1$ to $k = K$, this will give:

$$\frac{K\eta}{4}G^t \leq \mathbb{E}\big[f(\boldsymbol{y}_0^t) + \delta(1 + \frac{2}{K})^K \Delta_0^t\big] - \Big(\mathbb{E}\big[f(\boldsymbol{y}_K^t) + \delta\Delta_K^t\big]\Big) + 2\eta K E^t + (\frac{L}{2} + 8\delta)K\eta^2 \sigma_h^2 .$$

We note that $\boldsymbol{y}_0^t = \boldsymbol{x}^{t-1}$ and $\boldsymbol{y}_K^t = \boldsymbol{x}^t$, which means $\Delta_0^t = 0$ and $\Delta_K^t = \Delta^t$. So we have :

$$\frac{\eta}{4}G^t \leq \frac{F^{t-1} - F^t}{K} - \frac{\delta}{K}\Delta^t + 2\eta E^t + (\frac{L}{2} + 8\delta)\eta^2 \sigma_h^2 .$$

$\square$

Let's derive now the convergence rate.

We have :
$$\begin{cases} \frac{\eta}{4}G^t \le \frac{F^{t-1}-F^t}{K} + 2\eta E^t + (\frac{L}{2}+8\delta)\eta^2\sigma_h^2, \\ E^t \le (1-\frac{a}{2})E^{t-1} + \frac{36\delta^2 K^2\eta^2}{a}G^{t-1} + a^2\sigma_{f-h}^2 + \frac{a}{24K}\sigma_h^2. \end{cases}$$

We will add to both sides of the first inequality the quantity $\frac{4\eta}{a}E^t$.

So :
$$\frac{\eta}{4}G^t + \frac{4\eta}{a}E^t \le \frac{F^{t-1}-F^t}{K} + 2\eta E^t + (\frac{L}{2}+8\delta)\eta^2\sigma_h^2 + \frac{4\eta}{a}\left((1-\frac{a}{2})E^{t-1} + \frac{36\delta^2 K^2\eta^2}{a}G^{t-1} + a^2\sigma_{f-h}^2 + \frac{a}{24K}\sigma_h^2\right),$$

Which gives for $a \ge 36\delta K\eta$ :
$$\frac{\eta}{4}G^t - \frac{\eta}{8}G^{t-1} \le \Phi^{t-1} - \Phi^t + 4\eta a\sigma_{f-h}^2 + (\frac{L}{2}+8\delta)\eta^2\sigma_h^2 + \frac{\eta}{6K}\sigma_h^2, \tag{7}$$

For a potential $\Phi^t = \frac{F^t}{K} + (\frac{4\eta}{a}-2\eta)E^t \le \frac{F^t}{K} + \frac{4\eta}{a}E^t$.

Summing the inequality 7 over $t$, gives :
$$\frac{1}{8T}\sum_{t=1}^{T}G^t \le \frac{\Phi^0}{\eta T} + 4a\sigma_{f-h}^2 + (\frac{L}{2}+8\delta)\eta\sigma_h^2 + \frac{\sigma_h^2}{6K},$$
$$\le \frac{F^0}{\eta KT} + \frac{4}{aT}E^0 + 4a\sigma_{f-h}^2 + (\frac{L}{2}+8\delta)\eta\sigma_h^2 + \frac{\sigma_h^2}{6K},$$

Now taking $a = 36\delta K\eta \le 1$, we get :
$$\frac{1}{8T}\sum_{t=1}^{T}G^t \le \frac{F^0}{\eta KT} + \frac{1}{8\delta K\eta T}E^0 + 144\delta K\eta\sigma_{f-h}^2 + (\frac{L}{2}+8\delta)\eta\sigma_h^2 + \frac{\sigma_h^2}{6K},$$
$$= \frac{\tilde{F}}{\eta KT} + 144\delta K\eta\sigma_{f-h}^2 + (\frac{L}{2}+8\delta)\eta\sigma_h^2 + \frac{\sigma_h^2}{6K},$$
$$= \frac{\tilde{F}}{\eta KT} + 144L\beta K\eta\sigma_f^2 + \frac{\sigma_h^2}{6K},$$

For $\tilde{F} = F^0 + \frac{E^0}{8\delta}$ and $\beta = \frac{\delta}{L}\left(\frac{\sigma_{f-h}^2}{\sigma_f^2} + \frac{1}{18K}\frac{\sigma_h^2}{\sigma_f^2}\right) + \frac{1}{288K}\frac{\sigma_h^2}{\sigma_f^2}$.

Taking into account all the conditions on $\eta$ that were necessary, we can take :
$$\eta = \min(\frac{1}{L}, \frac{1}{192\delta K}, \sqrt{\frac{\tilde{F}}{144L\beta K^2 T\sigma_f^2}}).$$

This choice gives us the rate :
$$\frac{1}{8T}\sum_{t=1}^{T}G^t \le 24\sqrt{\frac{L\beta\tilde{F}\sigma_f^2}{T}} + \frac{(L+192\delta K)\tilde{F}}{KT} + \frac{\sigma_h^2}{6K}.$$

**Dealing with the term $E^0$.** If we use a batch-size $S$ times bigger for generating $\boldsymbol{m}^0$ than the other batch-sizes, then we have $E^0 \le \frac{\sigma_{f-h}^2}{S}$. In particular, for $S = T$, $E^0 \le \frac{\sigma_{f-h}^2}{T}$.

Now by taking $a = \max(\frac{1}{T}, 36\delta K\eta)$, we ensure $\frac{E^0}{aT} \le E^0 \le \frac{\sigma_{f-h}^2}{T}$, then

$$\frac{1}{8T} \sum_{t=1}^{T} G^t \le \frac{F^0}{\eta KT} + \frac{4}{aT} E^0 + 4a\sigma_{f-h}^2 + (\frac{L}{2} + 8\delta)\eta\sigma_h^2 + \frac{\sigma_h^2}{6K},$$

$$\le \frac{F^0}{\eta KT} + 4E^0 + \frac{4\sigma_{f-h}^2}{T} + 144K\delta\eta\sigma_{f-h}^2 + (\frac{L}{2} + 8\delta)\eta\sigma_h^2 + \frac{\sigma_h^2}{6K},$$

$$\le \frac{F^0}{\eta KT} + \frac{8\sigma_{f-h}^2}{T} + 144K\delta\eta\sigma_{f-h}^2 + (\frac{L}{2} + 8\delta)\eta\sigma_h^2 + \frac{\sigma_h^2}{6K},$$

$$= \frac{F^0}{\eta KT} + 144L\beta K\eta\sigma_f^2 + \frac{8\sigma_{f-h}^2}{T} + \frac{\sigma_h^2}{6K},$$

Taking

$$\eta = \min(\frac{1}{L}, \frac{1}{192\delta K}, \sqrt{\frac{F^0}{144L\beta K^2 T\sigma_f^2}}).$$

we get the following :

$$\frac{1}{8T} \sum_{t=1}^{T} G^t \le 24\sqrt{\frac{L\beta F^0 \sigma_f^2}{T}} + \frac{(L + 192\delta K)F^0}{KT} + \frac{8\sigma_{f-h}^2}{T} + \frac{\sigma_h^2}{6K}.$$

### C.3  MVR with auxiliary information

#### C.3.1  Algorithm description

The algorithm that we proposed proceeds in cycles, at the beginning of each cycle we have states $\boldsymbol{x}^{t-2}$ and $\boldsymbol{x}^{t-1}$. To update these states, we take $K+1$ iterations of the form $\boldsymbol{y}_k^t = \boldsymbol{y}_{k-1}^t - \eta\boldsymbol{d}_k^t$ where $\boldsymbol{y}_0^t = \boldsymbol{x}^{t-1}$, $\boldsymbol{d}_k^t = \boldsymbol{g}_h(\boldsymbol{y}_{k-1}^t, \xi_{h,k}^t) + \boldsymbol{m}^t$ and $\boldsymbol{m}^t = (1-a)\boldsymbol{m}^{t-1} + a\boldsymbol{g}_{f-h}(\boldsymbol{x}^{t-1}, \xi_{f-h}^{t-1}) + (1-a)\Big(\boldsymbol{g}_{f-h}(\boldsymbol{x}^{t-1}, \xi_{f-h}^{t-1}) - \boldsymbol{g}_{f-h}(\boldsymbol{x}^{t-2}, \xi_{f-h}^{t-1})\Big)$. Then we set $\boldsymbol{x}^t = \boldsymbol{y}_K^t$.

---

**Algorithm 5** AUXMVR($f, h$)

---

**Require:** $\boldsymbol{x}_0, \boldsymbol{m}^0, \eta, a, T, K$

  $\boldsymbol{x}_{-1} = \boldsymbol{x}_0$
  **for** $t = 1$ to $T$ **do**
    Sample $\xi_{f-h}^{t-1}$
    $\boldsymbol{g}_{prev}^{f-h} = \boldsymbol{g}_{f-h}(\boldsymbol{x}^{t-2}, \xi_{f-h}^{t-1})$
    $\boldsymbol{g}^{f-h} = \boldsymbol{g}_{f-h}(\boldsymbol{x}^{t-1}, \xi_{f-h}^{t-1})$
    $\boldsymbol{m}^t = (1-a)\boldsymbol{m}^{t-1} + a\boldsymbol{g}^{f-h} + (1-a)(\boldsymbol{g}^{f-h} - \boldsymbol{g}_{prev}^{f-h})$ §update momentum
    $\boldsymbol{y}_0^t = \boldsymbol{x}^{t-1}$
    **for** $k = 0$ to $K - 1$ **do**
      Sample $\xi_h^{t,k}$; Compute $\boldsymbol{g}_h(\boldsymbol{y}_k^t, \xi_h^{t,k})$
      $\boldsymbol{d}_k^t = \boldsymbol{g}_h(\boldsymbol{y}_k^t, \xi_h^{t,k}) + \boldsymbol{m}^t$
      $\boldsymbol{y}_{k+1}^t = \boldsymbol{y}_k^t - \eta\boldsymbol{d}_k^t$
    **end for**
    Update $\boldsymbol{x}^t = \boldsymbol{y}_K^t$
  **end for**

---

We will use the same notations as section C.2.

#### C.3.2  Convergence of MVR with auxiliary information

We prove the following theorem that gives the convergence rate of this algorithm in the non-convex case.

**Theorem C.10.** *Under assumptions A3.1, 3.2,4.5. For $\boldsymbol{m}^0$ such that $E^0 \le \sigma^2_{f-h}/T$, for $a = \max(\frac{1}{T}, 1156\delta^2 K^2\eta^2)$ and $\eta = \min\left(\frac{1}{L}, \frac{1}{192\delta K}, \frac{1}{K}\left(\frac{F^0}{18432\delta^2 T\sigma^2_{f-h}}\right)^{1/3}, \sqrt{\frac{F^0}{KT(L/2+8\delta K)}}\right)$. This choice gives us the rate :*

$$\frac{1}{8KT}\sum_{t=1}^{T}\sum_{k=1}^{K-1}\mathbb{E}\big[\|\nabla f(\boldsymbol{y}_k^t)\|_2^2\big] \le 40\Big(\frac{\delta F^0\sigma_{f-h}}{T}\Big)^{2/3} + 2\sqrt{\frac{(L/2+8\delta)F^0\sigma_h^2}{KT}} + \frac{(L+192\delta K)F^0}{KT} + \frac{8\sigma^2_{f-h}}{T} + \frac{\sigma_h^2}{48K}.$$

*Where $F^0 = f(\boldsymbol{x}^0) - f^\star$ and $E^0 = \mathbb{E}\|\boldsymbol{m}^0 - (\nabla f(\boldsymbol{x}^0) - \nabla h(\boldsymbol{x}^0))\|^2$.*

### C.3.3 Proof

Given that the form of $\boldsymbol{d}_k^t$ is the same as in AuxMOM (Algorithm 3), the same Lemmas still hold, except for Lemma C.8 which changes to the following Lemma.

**Momentum variance of AUXMVR.** Here we will bound the quantity $E^t$.

**Lemma C.11.** *Under assumptions A4.5,A3.2 , we have :*

$$E^t \le (1-a)E^{t-1} + 2\delta^2\Delta^{t-1} + 2a^2\sigma^2_{f-h}.$$

*Combining this inequality with Lemma C.7, we get for $\eta \le \frac{1}{5K\delta}$ and $a \ge 144K^2\delta^2\eta^2$:*

$$E^t \le (1-\frac{a}{2})E^{t-1} + 36K^2\delta^2\eta^2 G^{t-1} + 2a^2\sigma^2_{f-h} + 6K\delta^2\eta^2\sigma_h^2.$$

*Proof.* First, we notice that

$$\begin{aligned}
\boldsymbol{e}^t &= \boldsymbol{m}^t - \nabla f(\boldsymbol{x}^{t-1}) + \nabla h(\boldsymbol{x}^{t-1}) \\
&= (1-a)\boldsymbol{m}^{t-1} + a\boldsymbol{g}_{f-h}(\boldsymbol{x}^{t-1}, \xi_f^{t-1}) \\
&\quad + (1-a)\Big(\boldsymbol{g}_{f-h}(\boldsymbol{x}^{t-1}, \xi_{f-h}^{t-1}) - \boldsymbol{g}_{f-h}(\boldsymbol{x}^{t-2}, \xi_{f-h}^{t-1})\Big) - \nabla f(\boldsymbol{x}^{t-1}) + \nabla h(\boldsymbol{x}^{t-1}) \\
&= (1-a)\boldsymbol{e}^{t-1} + a(\boldsymbol{g}_{f-h}(\boldsymbol{x}^{t-1}, \xi_{f-h}^{t-1}) - \nabla f(\boldsymbol{x}^{t-1}) + \nabla h(\boldsymbol{x}^{t-1})) \\
&\quad + (1-a)\big(\boldsymbol{g}_{f-h}(\boldsymbol{x}^{t-1}, \xi_{f-h}^{t-1}) - \nabla f(\boldsymbol{x}^{t-1}) + \nabla h(\boldsymbol{x}^{t-1}) - \boldsymbol{g}_{f-h}(\boldsymbol{x}^{t-2}, \xi_{f-h}^{t-1}) \\
&\quad + \nabla f(\boldsymbol{x}^{t-2}) - \nabla h(\boldsymbol{x}^{t-2}))
\end{aligned}$$

Notice that $\boldsymbol{e}^{t-1}$ is independent of the rest of the formulae which is itself centered (has a mean equal to zero), so :

$$E^t \le (1-a)^2 E^{t-1} + 2a^2\sigma^2_{f-h} + 2\delta^2\Delta^{t-1}.$$

Now for $\eta \le \frac{1}{5K\delta}$, we can use Lemma C.7 and we get

$$\begin{aligned}
E^t &\le (1-a)E^{t-1} + 2a^2\sigma^2_{f-h} + 2\delta^2\big(36K^2\eta^2 E^t + 18K^2\eta^2 G^t + 3K\eta^2\sigma_h^2\big) \\
&\le (1-a+74K^2\delta^2\eta^2)E^{t-1} + 2a^2\sigma^2_{f-h} + 36K^2\delta^2\eta^2 G^{t-1} + 6K\delta^2\eta^2\sigma_h^2
\end{aligned}$$

It is easy to verify that for $a \ge 144K^2\delta^2\eta^2$, we have $74K^2\delta^2\eta^2 \le a/2$ which finishes the proof. $\qquad\square$

Let's now derive the convergence rate of AUXMVR.

We have :

$$\begin{cases} \frac{\eta}{4}G^t \le \frac{F^{t-1}-F^t}{K} + 2\eta E^t + (\frac{L}{2}+8\delta)\eta^2\sigma_h^2, \\ E^t \le (1-\frac{a}{2})E^{t-1} + 36K^2\delta^2\eta^2 G^{t-1} + 2a^2\sigma^2_{f-h} + 6K\delta^2\eta^2\sigma_h^2. \end{cases}$$

We will add to both sides of the first inequality the quantity $\frac{4\eta}{a}E^t$.

So :

$$\frac{\eta}{4}G^t + \frac{4\eta}{a}E^t \le \frac{F^{t-1}-F^t}{K} + 2\eta E^t + (\frac{L}{2}+8\delta)\eta^2\sigma_h^2 + \frac{4\eta}{a}\Big((1-\frac{a}{2})E^{t-1} + 36K^2\delta^2\eta^2 G^{t-1} + 2a^2\sigma^2_{f-h} + 6K\delta^2\eta^2\sigma_h^2\Big),$$

Let's define the potential $\Phi^t = \frac{F^t}{K} + (\frac{4\eta}{a} - 2\eta)E^t \leq \frac{F^t}{K} + \frac{4\eta}{a}E^t$.

Then

$$\frac{\eta}{4}G^t \leq \Phi^{t-1} - \Phi^t + (\frac{L}{2} + 8\delta)\eta^2\sigma_h^2 + \frac{144K^2\delta^2\eta^3}{a}G^{t-1} + 8a\eta\sigma_{f-h}^2 + \frac{24K\delta^2\eta^3}{a}\sigma_h^2 \,,$$

for $a \geq 1152K^2\delta^2\eta^2$, we have $\frac{144K^2\delta^2\eta^3}{a} \leq \eta/8$, which means

$$\frac{\eta}{4}G^t - \frac{\eta}{8}G^{t-1} \leq \Phi^{t-1} - \Phi^t + (\frac{L}{2} + 8\delta)\eta^2\sigma_h^2 + 8a\eta\sigma_{f-h}^2 + \frac{24K\delta^2\eta^3}{a}\sigma_h^2 \,,$$

Summing the last inequality over $t$ gives :

$$\frac{1}{8T}\sum_{t=1}^{T}G^t \leq \frac{\Phi^0}{\eta T} + 8a\sigma_{f-h}^2 + (\frac{L}{2} + 8\delta)\eta\sigma_h^2 + \frac{24K\delta^2\eta^2}{a}\sigma_h^2 \,,$$

$$\leq \frac{F^0}{\eta KT} + \frac{4}{aT}E^0 + 8a\sigma_{f-h}^2 + (\frac{L}{2} + 8\delta)\eta\sigma_h^2 + \frac{24K\delta^2\eta^2}{a}\sigma_h^2 \,,$$

If we use a batch $T$ times larger at the beginning, we can ensure $E^0 \leq \frac{\sigma_{f-h}^2}{T}$, so:

$$\frac{1}{8T}\sum_{t=1}^{T}G^t \leq \frac{F^0}{\eta KT} + \frac{4\sigma_{f-h}^2}{aT^2} + 8a\sigma_{f-h}^2 + (\frac{L}{2} + 8\delta)\eta\sigma_h^2 + \frac{24K\delta^2\eta^2}{a}\sigma_h^2 \,,$$

Now taking $a = \max(\frac{1}{T}, 1152\delta^2K^2\eta^2)$, we get :

$$\frac{1}{8T}\sum_{t=1}^{T}G^t \leq \frac{F^0}{\eta KT} + \frac{4\sigma_{f-h}^2}{T} + \frac{8\sigma_{f-h}^2}{T} + 9216\delta^2\eta^2K^2\sigma_{f-h}^2 + (\frac{L}{2} + 8\delta)\eta\sigma_h^2 + \frac{\sigma_h^2}{48K} \,,$$

Taking into account all the conditions on $\eta$ that were necessary, we can take :

$$\eta = \min\left(\frac{1}{L}, \frac{1}{192\delta K}, \frac{1}{K}\left(\frac{F^0}{18432\delta^2 T\sigma_{f-h}^2}\right)^{1/3}, \sqrt{\frac{F^0}{KT(L/2 + 8\delta K)}}\right).$$

This choice gives us the rate :

$$\frac{1}{8T}\sum_{t=1}^{T}G^t \leq 40\left(\frac{\delta F^0\sigma_{f-h}}{T}\right)^{2/3} + 2\sqrt{\frac{(L/2 + 8\delta)F^0}{KT}} + \frac{(L + 192\delta K)F^0}{KT} + \frac{8\sigma_{f-h}^2}{T} + \frac{\sigma_h^2}{48K} \,.$$

## C.4 Generalization to multiple decentralized helpers.

We consider now the case where we have $N$ helpers: $h_1, \ldots, h_N$. This case can be easily solved by merging all the helpers into one helper $h = \frac{1}{N}\sum_{i=1}^{N} h_i$ for example (it is easy to see that if each $h_i$ is $\delta_i$-BHD from $f$ that their average $h$ would be $\frac{1}{N}\sum_{i=1}^{N}\delta_i$-BHD from f ). However, this is not possible if the helpers are decentralized (are not in the same place and cannot be made to be for privacy reasons for example). For this reason, we consider a Federated version of our optimization problem in the presence of auxiliary decentralized information.

In this case, we consider that all functions $h_i$ are such that :

$$\forall \boldsymbol{x}, \xi \: : \: \|\nabla^2 f(\boldsymbol{x}) - \nabla^2 h_i(\boldsymbol{x})\|_2 \leq \delta \,.$$

We will also need an additional assumption on $f$ :

**Assumption C.12. (Weak convexity.)** $f$ is $\delta$- weakly convex i.e. $\boldsymbol{x} \mapsto f(\boldsymbol{x}) + \delta\|\boldsymbol{x}\|_2^2$ is convex.

**Lemma C.13.** *Under Assumption C.12 the following is true :*

$$\forall N \forall \boldsymbol{x}, \boldsymbol{x}_1, \dots, \boldsymbol{x}_N \forall \alpha \geq \delta \ : \ f(\frac{1}{N}\sum_{i=1}^{N} \boldsymbol{x}_i) + \alpha \|\frac{1}{N}\sum_{i=1}^{N} \boldsymbol{x}_i - \boldsymbol{x}\|_2^2 \leq \frac{1}{N}\sum_{i=1}^{N}\left(f(\boldsymbol{x}_i) + \alpha\|\boldsymbol{x}_i - \boldsymbol{x}\|_2^2\right).$$

**About the need for Assumption C.12.** Assumption C.12 becomes strong for $\delta$ small which is not good, as this should be the easiest case. however, we would like to point out that we only need Assumption C.12 to deal with the averaging that we perform to construct the new state $\boldsymbol{x}^t$. In the case where we sample each time one and only one helper (i.e. $S = 1$), we don't need such an assumption. This assumption was made in the context of Federated Learning for example in (Karimireddy et al., 2020a).

### C.4.1 Decentralized momentum version

As in C.2.1 we start from $\boldsymbol{x}^{t-1}$, we sample (randomly) a set $S^t$ of $S$ helpers, we sample $\xi_f^t$ , compute $\boldsymbol{g}_f(\boldsymbol{x}^{t-1}, \xi_f^t)$, share it with all of the helpers for $h_i, i \in S^t$, which each sample $\xi_h^t$ (potentially correlated with $\xi_f^t$) and then update $\boldsymbol{m}_i^t$, then perform $K$ steps of $\boldsymbol{y}_{i,k}^t = \boldsymbol{y}_{i,k-1}^t - \eta \boldsymbol{d}_{i,k}^t$, once this finishes $\boldsymbol{y}_{i,K}^t$ is sent back to $f$ which then does $\boldsymbol{x}^t = \frac{1}{S}\sum_{i \in S^t} \boldsymbol{y}_{i,K}^t$.

For each $i$ we will denote $E_i^t = \mathbb{E}[\|\boldsymbol{m}_i^t - \nabla f(\boldsymbol{x}^{t-1}) + \nabla h_i(\boldsymbol{x}^{t-1})\|_2^2]$ and $E^t = \frac{1}{S}\sum_{i \in S^t} E_i^t$.

---

**Theorem C.14.** *Under assumptions A3.1, 3.2,3.3 (and assumption C.12 if $S > 1$). For $a = 36\delta K\eta$ and $\eta = \min(\frac{1}{L}, \frac{1}{144\delta K}, \sqrt{\frac{\tilde{F}}{128L\beta KT\sigma_f^2}})$. This choice gives us the rate :*

$$\frac{1}{8KST}\sum_{t=1}^{T}\sum_{k=1}^{K-1}\sum_{i \in S^t}\mathbb{E}\big[\|\nabla f(\boldsymbol{y}_{i,k}^t)\|_2^2\big] \leq 24\sqrt{\frac{L\beta\tilde{F}\sigma_f^2}{T}} + \frac{(L+192\delta K)\tilde{F}}{KT} + \frac{\sigma_h^2}{6K}.$$

*where $\tilde{F} = F^0 + \frac{E^0}{8\delta}$ and $\beta = \frac{\delta}{L}\left(\frac{\sigma_{f-h}^2}{\sigma_f^2} + \frac{1}{18K}\frac{\sigma_h^2}{\sigma_f^2}\right) + \frac{1}{288K}\frac{\sigma_h^2}{\sigma_f^2}$, $F^0 = f(\boldsymbol{x}^0) - f^\star$ and $E^0 = \mathbb{E}[\|\boldsymbol{m}^0 - \nabla f(\boldsymbol{x}^0) + \nabla h(\boldsymbol{x}^0)\|_2^2]$.*

*Furthermore, if we use a batch-size $T$ times bigger for computing an estimate $\boldsymbol{m}_i^0$ of $\nabla f(\boldsymbol{x}^0) - \nabla h_i(\boldsymbol{x}_0)$, then by taking $a = \max(\frac{1}{T}, 36\delta K\eta)$ and replacing $\tilde{F}$ by $F^0$ in the expression of $\eta$, we get :*

$$\eta = \min(\frac{1}{L}, \frac{1}{192\delta K}, \sqrt{\frac{F^0}{144L\beta K^2 T\sigma_f^2}}).$$

*we get the following :*

$$\frac{1}{8KST}\sum_{t=1}^{T}\sum_{k=1}^{K-1}\sum_{i \in S^t}\mathbb{E}\big[\|\nabla f(\boldsymbol{y}_{i,k}^t)\|_2^2\big] \leq 24\sqrt{\frac{L\beta F^0\sigma_f^2}{T}} + \frac{(L+192\delta K)F^0}{KT} + \frac{8\sigma_{f-h}^2}{T} + \frac{\sigma_h^2}{6K}.$$

---

*Proof.* The proof follows the same lines as the proof of in C.2.1.
Two changes should be made to the proof: $G^t$ needs to be updated to $G^t = \frac{1}{SK}\sum_{i \in S^t, k}\mathbb{E}\big[\|\nabla f(\boldsymbol{y}_k^t)\|_2^2\big]$ in both Lemmas C.7 and C.9

In fact, in this case $\boldsymbol{x}^t = \frac{1}{S}\sum_{i\in S^t} \boldsymbol{y}^t_{i,K}$, which means using convexity of the squared norm :

$$\Delta^t \leq \frac{1}{S}\sum_{i\in S^t} \Delta^t_{i,K}(:= \mathbb{E}\big[\|\boldsymbol{y}^t_{i,K} - \boldsymbol{x}^{t-1}\|^2_2\big])$$

$$\leq \frac{1}{S}\sum_{i\in S^t}\sum_k (1 + \frac{1}{K})^{K-k}\Big(12K\eta^2 E^t_i + 6K\eta^2\mathbb{E}[\|\nabla f(\boldsymbol{y}^t_{i,k-1})\|^2_2] + \eta^2\sigma^2_h\Big)$$

$$\leq 36K^2\eta^2 E^t + 18K^2\eta^2\frac{1}{KS}\sum_{i\in S^t}\sum_k \mathbb{E}\big[\|\nabla f(\boldsymbol{y}^t_{i,k})\|^2_2\big] + 3K\eta^2\sigma^2_h$$

$$= 36K^2\eta^2 E^t + 18K^2\eta^2 G^t + 3K\eta^2\sigma^2_h.$$

And in the descent lemma (That modifies LemmaC.9) we will have :

$$\forall i : \frac{K\eta}{4}\frac{1}{K}\sum_k \mathbb{E}\big[\|\nabla f(\boldsymbol{y}^t_{i,k})\|^2_2\big] \leq \mathbb{E}\big[f(\boldsymbol{y}^t_{i,0}) + \delta(1 + \frac{2}{K})^K\Delta^t_{i,0}\big] - \Big(\mathbb{E}\big[f(\boldsymbol{y}^t_{i,K}) + \delta\Delta^t_{i,K}\big]\Big)$$

$$+ 2\eta KE^t + (\frac{L}{2} + 8\delta)\eta^2\sigma^2_h.$$

Where $\Delta^t_{i,0} = 0$ and using LemmaC.13 we have :

$$F^t + \delta\Delta^t \leq \frac{1}{S}\sum_{i\in S^t}\Big(\mathbb{E}\big[f(\boldsymbol{y}^t_{i,K}) + \delta\Delta^t_{i,K}\big]\Big).$$

Using the last inequality it is easy to get :

$$\frac{K\eta}{4}G^t \leq F^{t-1} - F^t - \delta\Delta^t + 2\eta KE^t + (\frac{L}{2} + 8\delta)\eta^2\sigma^2_h.$$

All the rest is the same. $\qquad\square$

### C.4.2 Decentralized MVR version

As in C.3.1 we start from $\boldsymbol{x}^{t-1}$, we sample (randomly) a set $S^t$ of $S$ helpers, we sample $\xi^t_f$ , compute $\boldsymbol{g}_f(\boldsymbol{x}^{t-1}, \xi^t_f)$, share it with all of the helpers for $h_i, i\in S^t$, which each sample $\xi^t_h$ (potentially correlated with $\xi^t_f$) and then update $\boldsymbol{m}^t_i$, then perform $K$ steps of $\boldsymbol{y}^t_{i,k} = \boldsymbol{y}^t_{i,k-1} - \eta\boldsymbol{d}^t_{i,k}$, once this finishes $\boldsymbol{y}^t_{i,K}$ is sent back to $f$ which then does $\boldsymbol{x}^t = \frac{1}{S}\sum_{i\in S^t}\boldsymbol{y}^t_{i,K}$.

We can prove the following theorem under the same changes as in the momentum case.

---

**Theorem C.15.** *Under assumptions A3.1, 3.2,3.3 (and assumption C.12 if $S > 1$). For momentum initialization $\boldsymbol{m}^0_i$ such that $E^0_i \leq \sigma^2_{f-h}/T$, for $a = \max(\frac{1}{T}, 1152\delta^2 K^2\eta^2)$, and $\eta = \min\left(\frac{1}{L}, \frac{1}{192\delta K}, \frac{1}{K}\left(\frac{F^0}{18432\delta^2 T\sigma^2_{f-h}}\right)^{1/3}, \sqrt{\frac{F^0}{KT(L/2+8\delta K)}}\right)$, we get:*

$$\frac{1}{8KST}\sum_{t=1}^T\sum_{i\in S^t}\sum_{k=1}^{K-1}\mathbb{E}\big[\|\nabla f(\boldsymbol{y}^t_{i,k})\|^2_2\big] \leq 40\Big(\frac{\delta F^0\sigma_{f-h}}{T}\Big)^{2/3} + 2\sqrt{\frac{(L/2+8\delta)F^0}{KT}} + \frac{(L+192\delta K)F^0}{KT}$$

$$+ \frac{8\sigma^2_{f-h}}{T} + \frac{\sigma^2_h}{48K}.$$

*Where $F^0 = f(\boldsymbol{x}^0) - f^\star$ and $E^0_i = \|\boldsymbol{m}^0_i - (\nabla f(\boldsymbol{x}^0) - \nabla h_i(\boldsymbol{x}^0))\|^2_2$.*

---

