# Optimization with Access to Auxiliary Information

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

.** ~~We note the term $E^0$ in~~ the ~~definition of $\tilde{F}$, we show in the appendix how to get rid of this term $E^0$ by making it $\mathcal{O}(1/T)$, for this reason, we may consider $\tilde{F} = F^0$~~ condition $E^0 \leq \sigma_{f-h}^2/T$ can be ensured by using a batch size $T$ times larger to estimate $\boldsymbol{m}^0$, this will at most result in doubling the number of steps $T$. In practice, we did not need to ensure this condition.
~~For simplicity, we take $\sigma_h = \sigma_f$ (it is reasonable to assume the two quantities of~~ We also note the term $\sigma_h^2/K$, which corresponds to ~~the same order).~~

**Sanity checks.** ~~For $h = f$,~~ error of solving the inner problem 4. Remarkably, although we are using SGD steps of the helper $h$, we ~~have $\delta = 0$, and it is easy to see that we get the rate of $KT$-steps of SGD. For $h = 0$ we have $\delta = L$, in this case, we don't gain anything as should be the case~~ get a $1/K$ error instead of $1/\sqrt{K}$; this can be explained by the fact that we initialize using the (approximate) solution of the last inner problem which accelerates convergence; we can potentially get faster rates by using variance reduction methods on $h$.
We will compare this rate to that of SGD under the same amount of work asked from $f$: $\mathcal{O}\Big(\sqrt{\frac{LF^0\sigma_f^2}{T}} + \frac{LF^0}{T}\Big)$. ~~**When do we gain from the helper? (Noiseless case.)**~~ for $\sigma_f = \sigma_h = 0$, we have the rate ~~$\mathcal{O}(LF^0/(KT) + \delta F^0/T)$, we compare this to $\mathcal{O}(LF^0/T)$ the expected rate had we only used gradients~~ which corresponds to $\mathcal{O}\big(\frac{LF^0\sigma_f^2}{\varepsilon^2} + \frac{LF^0}{\varepsilon}\big)$ stochastic gradient calls of $f$ necessary to get an $\varepsilon$-stationary point $\hat{\boldsymbol{x}}$ in expectation i.e. a point $\hat{\boldsymbol{x}}$ such that $\mathbb{E}[\|\nabla f(\hat{\boldsymbol{x}})\|_2^2] \leq \varepsilon$ (the expectation is taken over the algorithm that generated $\hat{\boldsymbol{x}}$). ~~This means that we gain a multiplicative factor of $1/K + \delta/L$, which means we need $\delta/L \ll 1$ (very small)~~ to neglect it. ~~however, there is another consequence: for $K \leq L/\delta$ the rate is $\mathcal{O}(LF^0/(KT))$, in other words, it is as if we were using $K$ gradients~~

In comparison, based on Theorem 4.3, we can show the following corollary:

---

**Corollary 4.4** (Iteration complexity of AuxMOM). *Let $\hat{x}$ be chosen uniformly at random from the iterates generated by AuxMOM. To guarantee $\mathbb{E}[\|\nabla f(\hat{\boldsymbol{x}})\|_2^2] \leq \varepsilon$, AuxMOM needs at most*

$$\mathcal{O}\Big(\frac{\delta F^0\sigma_{f-h}^2}{\varepsilon^2} + \frac{\delta F^0}{\varepsilon} + \frac{\sigma_{f-h}^2}{\varepsilon} + \frac{\sigma_h^2}{LF^0} + 1_{\sigma_h \neq 0}\frac{LF^0}{\varepsilon}\Big)$$

---

*(stochastic) gradient calls* of $f$ ~~instead of only 1 and $K-1$ gradients of $h$. **(Noisy case.)** We~~

In particular, we see that in the ~~dominant term we have the factor $\beta$ that multiplies the variance term $\sigma_f^2$ that we have in SGD~~ dominating order of $1/\varepsilon$, when access to gradients of $f$ is stochastic (**noisy case**), we replaced $L\sigma_f^2$ by $\delta\sigma_{f-h}^2$ which might be very small, either because $\delta \ll L$ or $\sigma_{f-h}^2 \ll \sigma_f^2$. In the **noiseless case** (when we ~~don't use the helper $h$), the smaller $\beta$ the better, this is the case in general if $\delta/L$, $\sigma_h^2/\sigma_f^2$ are small and $\sigma_{f-h}^2/\sigma_f^2$. We also notice a small gain from $K$.~~ have full access to gradients of $f$), we replaced $L$ by $\delta$.

~~Specifically, $\beta = \mathcal{O}\left(\frac{\delta}{L}\frac{\sigma_{f-h}^2}{\sigma_f^2} + \frac{1}{K}(1+\frac{\delta}{L})\frac{\sigma_h^2}{\sigma_f^2}\right)$ is constituted of two terms, the first one $\frac{\delta}{L}\frac{\sigma_{f-h}^2}{\sigma_f^2}$ suggests that we may benefit if $\delta \ll L$ or if $\sigma_{f-h}^2 \ll \sigma_f^2$, we get the latter if our gradient estimates of $f$ are positively correlated with the gradients of~~ Of course, the gain that we obtain is not for free; it comes at the cost of using $K = \mathcal{O}\left(\frac{\sigma_h^2}{\varepsilon} + \mathbb{1}_{\delta \neq 0}\frac{L}{\delta} + 1\right)$ inner steps of the helper $h$. ~~The second term $\frac{1}{K}(1+\frac{\delta}{L})\frac{\sigma_h^2}{\sigma_f^2}$ decreases with $K$ and we can neglect it if we take $K \geq K_0 = \max(L\sigma_f^2/\delta\sigma_{f-h}^2, \sigma_h^2/\sigma_{f-h}^2)$.~~

~~**Overall,we have two regimes:(I) if $K \leq K_0$ then our rate becomes $\mathcal{O}\left(\sqrt{\frac{LF^0\sigma_h^2}{KT}} + \frac{LF^0}{KT}\right)$ i.e. we replaced $L$ by $L/K$, we gain from increased values of $K$**~~ **Sanity checks.** ~~**(II) If $K \geq K_0$ our rate becomes $\mathcal{O}\left(\sqrt{\frac{\delta F^0\sigma_{f-h}^2}{T}} + \frac{\delta F^0}{T}\right)$, which means we replaced $L$ by $\delta$ which might be very small. This is equivalent to solving the sub-problem (4), but instead of needing $K \to \infty$ we only need $K \sim L/\delta$. We also replaced $\sigma_f^2$ by $\sigma_{f-h}^2$ which might also be small if we have positively correlated noise between gradients of $f$ and $h$.**~~ For $h = f$, we have $\delta = 0$, we get the iteration complexity $\mathcal{O}\left(\frac{\sigma^2}{\varepsilon} + \frac{LF^0}{\varepsilon}\right)$ which corresponds to the rate of $KT$-steps of SGD. For $h = 0$, we have $\delta = L$; in this case, we don't gain anything as should be the case.

**SVRG in the non-convex setting.** In particular, because Theorem 4.3 also applies for the case where we have multiple helpers, and we sample each time $S = 1$ helpers, we get that SVRG converges in this case as $\mathcal{O}\left(\frac{(L+\delta K)F^0}{KT}\right)$, which matches the known SVRG rate (Reddi et al., 2016) (up to $\delta$ being small). More interestingly, we obtain the same convergence rate by using only one batch (no need to sample) if the batch is representative enough of the data (i.e., satisfies our Hessian similarity Assumption 3.3).

**Local steps help in federated learning.** By employing the decentralized version of this theorem (as detailed in Appendix C.4), we can ascertain that local steps (denoted as $K$ in our context) indeed provide a beneficial contribution to Federated Learning. This finding aligns with the results presented in (Karimireddy et al., 2020a).

## 4.3 MVR based approach

We consider now the instance of Algorithm 1, which uses the MVR momentum in in (6). The detailed algorithm can be found in the Appendix Algorithm 5.

**Stronger assumptions.** For the analysis of this variant, we need a stronger similarity assumption

**Assumption 4.5. Stronger Hessian similarity.**

$$\exists \delta \in [0, L] \ \forall \zeta_{f-h} \ : \ \boldsymbol{g}_{f-h}(\cdot, \zeta_{f-h}) \text{ is } \delta\text{-Lipschitz.}$$

Assumption 4.5 is stronger than its counterpart Assumption 3.3 ~~, it~~ used for AuxMOM. It is reasonable to need ~~it since~~ already in a stronger assumption since, already, when using the MVR momentum ~~we need~~, a stronger smoothness assumption has to hold.

**Convergence of this variant.** We prove the following theorem that gives the convergence rate of this algorithm in the non-convex case.

**Theorem 4.6.** *Under assumptions A3.1, 3.2,4.5. ~~Assuming $\sigma_h = 0$, for $a = \max(\frac{1}{T}, 1296\delta^2 K^2\eta^2)$, for~~*

~~$\eta = \min\left(\frac{1}{L}, \frac{1}{1926\delta K}, \frac{1}{K}\left(\frac{F^0}{7776\delta^2 T\sigma_f^2}\right)^{1/3}\right)$, ensuring $E^0 \leq \sigma_f^2/T$ we have :~~

$$\frac{1}{KT}\sum_{t=1}^{T}\sum_{k=1}^{K-1}\mathbb{E}\big[\|\nabla f(\boldsymbol{y}_k^t)\|_2^2\big] = \mathcal{O}\left(\left(\frac{\delta F^0\sigma_{f-h}}{T}\right)^{2/3} + \frac{(L+\delta K)F^0}{KT} + \frac{\sigma_{f-h}^2}{T}\right).$$

*For $\boldsymbol{m}^0$ such that $E^0 \leq \sigma_{f-h}^2/T$, for $a = \max(\frac{1}{T}, 1156\delta^2 K^2\eta^2)$ and*

*$\eta = \min\left(\frac{1}{L}, \frac{1}{192\delta K}, \frac{1}{K}\left(\frac{F^0}{18432\delta^2 T\sigma_{f-h}^2}\right)^{1/3}, \sqrt{\frac{F^0}{KT(L/2+8\delta K)}}\right)$. This choice gives us the rate :*

$$\frac{1}{KT}\sum_{t=1}^{T}\sum_{k=1}^{K-1}\mathbb{E}\big[\|\nabla f(\boldsymbol{y}_k^t)\|_2^2\big] = \mathcal{O}\left(\left(\frac{\delta F^0\sigma_{f-h}}{T}\right)^{2/3} + \sqrt{\frac{(L+\delta)F^0\sigma_h^2}{KT}} + \frac{(L+\delta K)F^0}{KT} + \frac{\sigma_{f-h}^2}{T} + \frac{\sigma_h^2}{K}\right).$$

~~We provide the general theorem including $\sigma_h \neq 0$ in the Appendix.~~

**Baseline.** Under the same assumptions and for the same amount of work, MVR or STORM (Cutkosky & Orabona, 2019) has the rate: $\mathcal{O}\left(\left(\frac{LF^0\sigma_f}{T}\right)^{2/3} + \frac{LF^0}{T}\right)$. ~~**Gain.** (**Noiseless case**) If $\sigma_f = \sigma_{f-h} = 0$ then we~~

~~see that we replaced~~, this rate corresponds to needing at most $\mathcal{O}\left(\frac{LF^0\sigma_f}{\varepsilon^{3/2}} + \frac{LF^0}{\varepsilon}\right)$ (stochastic) gradient calls of $f$ to reach a point $\hat{\boldsymbol{x}}$ such $\mathbb{E}[\|\nabla f(\hat{\boldsymbol{x}})\|_2^2] \leq \varepsilon$ or $\varepsilon$-stationary point. This rate/iteration complexity is known to be optimal under the strong smoothness assumption: $f(\cdot, \xi)$ is $L$~~with $\frac{L}{K} + \delta$ which is smaller if $\delta \ll L$, in fact, exactly as for~~-smooth for all $\xi$ almost surely.
Using Theorem 4.6, it is easy to show the following corollary:

**Corollary 4.7** (Iteration complexity of AuxMVR)**.** *Let $\hat{x}$ be chosen uniformly at random from the iterates generated by AuxMVR. To guarantee $\mathbb{E}[\|\nabla f(\hat{\boldsymbol{x}})\|_2^2] \leq \varepsilon$, AuxMVR needs at most*

$$\mathcal{O}\left(\frac{\delta F^0\sigma_{f-h}}{\varepsilon^{3/2}} + \frac{\delta F^0}{\varepsilon} + \frac{\sigma_{f-h}^2}{\varepsilon} + \frac{\sigma_h^2}{LF^0} + 1_{\sigma_h \neq 0}\frac{LF^0}{\varepsilon}\right)$$

*(stochastic) gradient calls of $f$.*

The same conclusions as for AuxMOM are valid here. In the ~~**AUXMOM**~~**noisy case**, we ~~have two regimes, if $K \leq L/\delta$, then $\frac{L+\delta K}{K} = \mathcal{O}(L/K)$ this means that we improve with $K$, and we have a saturation regime when $K \geq L/\delta$ for which $\frac{L+\delta K}{K} = \mathcal{O}(\delta)$, which means we replace~~replaced $L\sigma_f^2$ by $\delta\sigma_{f-h}^2$ which might be very small, either because $\delta \ll L$ or $\sigma_{f-h}^2 \ll \sigma_f^2$. In the **noiseless case** (when we have full access to gradients of $f$), we replaced $L$ by $\delta$. ~~Overall, we gain from using $f$ the moment we have $\delta \leq L$.~~

~~**(Noisy case)** In this case We see that our noise term has $\delta\sigma_{f-h}^2$ instead of the $L\sigma_f^2$ in MVR's rate. Hence, if there is noise ($\sigma_{f-h} \neq 0$) then we have a better rate if $\delta \leq L$ or if $\sigma_{f-h}^2 \leq \sigma_f^2$. **Overall,** we gain when the similarity $\delta$ is small compared to $L$ or when we have a positive correlation between the noise of $f$ and~~ Again, this potential gain is obtained at the cost of using $K = \mathcal{O}\left(\frac{\sigma_h^2}{\varepsilon} + 1_{\delta \neq 0}\frac{L}{\delta} + 1\right)$ inner steps of the helper $h$. ~~–~~

## 5 Potential applications

The optimization with access to auxiliary information proposed is general enough that we can use it in many applications where ~~, we either ,~~ we either have access to auxiliary information explicitly, such as in auxiliary learning or transfer learning, or implicitly, such as in semi-supervised learning. We present here a non-exhaustive list of potential applications.

**Reusing batches in SGD training and core-sets.** In Machine Learning, the empirical risk minimization consisting of minimizing a function of the form $f(\boldsymbol{x}) = \frac{1}{N}\sum_{i=1}^{N} L(\boldsymbol{x}, \xi_i)$ is ubiquitous. In many applications, we want to summarize the data-set $\{\xi_i\}_{i=1}^{N}$ by a smaller potentially weighted subset $CS = \{(\xi_{i_j}, w_j)\}_{j=1}^{M}$, for positive weights $(w_j)_{j=1}^{M}$ that add up to one, this is referred to as a core-set (Bachem et al., 2017). In this case we can set $h(\boldsymbol{x}) = \sum_{(w,\xi)\in CS} wL(\boldsymbol{x}, \xi)$. An even ~~more~~ sampler problem is when we sample a batch $B \subset \{\xi_i\}_{i=1}^{N}$ of size $b \leq N$, one question we can ask is how can we reuse this same batch to optimize $f$? In this case we set $h(\boldsymbol{x}) = (1/b)\sum_{\xi\in B} L(\boldsymbol{x}, \xi)$. In the case where we have many batches $\{B_i\}_{i\in I}$, we can set $h_i(\boldsymbol{x}) = (1/|B_i|)\sum_{\xi\in B_i} L(\boldsymbol{x}, \xi)$ for each $i \in I$ and use our decentralized framework to sample each time a helper $h$.

**Note.** In case $h$ is obtained using a subset $B$ of the dataset defining $f$, there is a-priori a trade-off between the similarity between $f$ and $h$ measured by the hessian similarity parameter $\delta(B)$ and the cheapness of the gradients of $h$. A-priori, the bigger the size of $B$ is, the easier it is to obtain a small $\delta(B)$, but the more expensive it is to compute the gradients of $h$.

**Semi-supervised learning.** In Semi-supervised Learning (Zhu, 2005), We have a small set of carefully cleaned data $\mathcal{Z}$ directly related to our target task and a large set of unlabeled data $\tilde{\mathcal{Z}}$. Let us also assume ~~that~~ there is an auxiliary pre-training task defined over the source data, ~~for~~ e.g., this can be the popular learning with contrastive loss (Chen et al., 2020). In this setting, we have a set of transformations $\mathcal{T}$ which preserves the semantics of the data, two unlabeled data samples $\tilde{\zeta}_1, \tilde{\zeta}_2 \in \tilde{\mathcal{Z}}$, and a feature extractor $\phi_{\boldsymbol{x}}(\cdot) : \tilde{\mathcal{Z}} \to \mathbb{R}^k$ parameterized by $\boldsymbol{x}$. Then, the contrastive loss is of the form: $\tilde{\ell}(\phi_{\boldsymbol{x}}(\tilde{\zeta}_1), \phi_{\boldsymbol{x}}(\mathcal{T}(\tilde{\zeta}_1)), \phi_{\boldsymbol{x}}(\tilde{\zeta}_2))$ where the loss tries to minimize the distance between the representations $\phi_{\boldsymbol{x}}(\tilde{\zeta}_1)$ and $\phi_{\boldsymbol{x}}(\mathcal{T}(\tilde{\zeta}_1))$, while simultaneously maximizing distance to $\phi_{\boldsymbol{x}}(\tilde{\zeta}_2)$. Similarly, we also have a target loss $\ell : \mathcal{Z} \to \mathbb{R}$, which we care about. ~~Then;~~ then, we can define

$$f(\boldsymbol{x}) = \mathbb{E}_{\zeta\in\mathcal{Z}}\big[\ell(\boldsymbol{x}; \underline{\xi\zeta})\big]$$

and

$$h(\boldsymbol{x}) = \mathbb{E}_{\tilde{\zeta}_1, \tilde{\zeta}_2, \mathcal{T}}\big[\tilde{\ell}(\phi_{\boldsymbol{x}}(\tilde{\zeta}_1), \phi_{\boldsymbol{x}}(\mathcal{T}(\tilde{\zeta}_1)), \phi_{\boldsymbol{x}}(\tilde{\zeta}_2))\big].$$

~~The quality of the auxiliary unsupervised task can then be quantified as the Hessian distance $\|\nabla^2 f(\boldsymbol{x}) - \nabla^2 h(\boldsymbol{x})\|_2 \leq \delta$. Perhaps unintuitively, this is the *only* measure of similarity we need between the target task and the auxiliary source task. In particular, we do not need the optimum parameters to be related in any other manner. We hope this relaxation of the similarity requirements would allow for a more flexible design of the unsupervised source tasks~~

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

^0 \leq \frac{\sigma_{f-h}^2}{S}$, this way we get $\tilde{F} = F^0 + \frac{\sigma_{f-h}^2}{16\delta T} E^0 \leq \frac{\sigma_{f-h}^2}{T}$.

We can also take $a = \max(\frac{1}{T}, 32\delta K\eta)$, we get : Now by taking $a = \max(\frac{1}{T}, 36\delta K\eta)$, we ensure $\frac{E^0}{aT} \leq E^0 \leq \frac{\sigma_{f-h}^2}{T}$, then

$$\frac{1}{8T}\sum_{t=1}^{T} G^t \leq \frac{F^0}{\eta KT} + \frac{4}{aT}E^0 + 4a\sigma_{f-h}^2 + (\frac{L}{2} + 8\delta)\eta\sigma_h^2 + \frac{\sigma_h^2}{6K} \,,$$

$$\leq \frac{F^0}{\eta KT} + 4E^0 + \frac{4\sigma_{f-h}^2}{T} + 144K\delta\eta\sigma_{f-h}^2 + (\frac{L}{2} + 8\delta)\eta\sigma_h^2 + \frac{\sigma_h^2}{6K} \,,$$

$$\leq \frac{F^0}{\eta KT} + \frac{8\sigma_{f-h}^2}{T} + 144K\delta\eta\sigma_{f-h}^2 + (\frac{L}{2} + 8\delta)\eta\sigma_h^2 + \frac{\sigma_h^2}{6K} \,,$$

$$= \frac{F^0}{\eta KT} + 144L\beta K\eta\sigma_f^2 + \frac{8\sigma_{f-h}^2}{T} + \frac{\sigma_h^2}{6K} \,,$$

Taking

$$\eta = \min(\frac{1}{L}, \frac{1}{192\delta K}, \sqrt{\frac{F^0}{144L\beta K^2 T \sigma_f^2}}).$$

we get the following :

$$\frac{1}{8T}\sum_{t=1}^{T} G^t \leq 24\sqrt{\frac{L\beta F^0 \sigma_f^2}{T}} + \frac{(L + 192\delta K)\tilde{F}}{KT} \frac{(L + 192\delta K)F^0}{KT} + \frac{2\sigma_{f-h}^2}{T} \frac{8\sigma_{f-h}^2}{T} + \frac{\sigma_h^2}{6K} \,.$$

### C.3 MVR with auxiliary information

#### C.3.1 Algorithm description

The algorithm that we proposed proceeds in cycles, at the beginning of each cycle we have states $\boldsymbol{x}^{t-2}$ and $\boldsymbol{x}^{t-1}$. To update these states, we take $K + 1$ iterations of the form $\boldsymbol{y}_k^t = \boldsymbol{y}_{k-1}^t - \eta\boldsymbol{d}_k^t$ where $\boldsymbol{y}_0^t = \boldsymbol{x}^{t-1}$, $\boldsymbol{d}_k^t = \boldsymbol{g}_h(\boldsymbol{y}_{k-1}^t, \xi_{h,k}^t) + (1-a)\boldsymbol{m}^{t-1} + a\boldsymbol{g}_{f-h}(\boldsymbol{x}^{t-2}, \xi_{f-h}^{t-1})$ $\boldsymbol{d}_k^t = \boldsymbol{g}_h(\boldsymbol{y}_{k-1}^t, \xi_{h,k}^t) + \boldsymbol{m}^t$ and $\boldsymbol{m}^t = (1-a)\boldsymbol{m}^{t-1} + a\boldsymbol{g}_{f-h}(\boldsymbol{x}^{t-1}, \xi_{f-h}^{t-1}) + (1-a)\Big(\boldsymbol{g}_{f-h}(\boldsymbol{x}^{t-1}, \xi_{f-h}^{t-1}) - \boldsymbol{g}_{f-h}(\boldsymbol{x}^{t-2}, \xi_{f-h}^{t-1})\Big)$. Then we set $\boldsymbol{x}^t = \boldsymbol{y}_K^t$.

**Notation change.** From now on, we will denote $\Delta_k^t = \mathbb{E}\big[\|\boldsymbol{y}_k^t - \boldsymbol{x}^{t-2}\|_2^2\big]$, we will keep other quantities same as before. This time also we will denote $\bar{\boldsymbol{d}}_k^t = \nabla h(\boldsymbol{y}_{k-1}^t) + (1-a)\boldsymbol{m}^{t-1} + a\boldsymbol{g}_{f-h}(\boldsymbol{x}^{t-2}, \xi_f^{t-1})$ and we note that by fixing $\boldsymbol{y}_{k-1}^t$ we have $\mathbb{E}[\boldsymbol{d}_k^t] = \bar{\boldsymbol{d}}_k^t$ and $\mathbb{E}[\|\boldsymbol{d}_k^t - \bar{\boldsymbol{d}}_k^t\|_2^2] \leq \sigma_h^2$. We will use the same notations as section C.2.

#### C.3.2 Convergence of MVR with auxiliary information

We prove the following theorem that gives the convergence rate of this algorithm in the non-convex case.

**Theorem C.10.** *Under assumptions A3.1, 3.2,4.5.* *For* $a = \max(\frac{1}{T}, 1296\delta^2 K^2 \eta^2)$ *if* $\sigma_h = 0$, $a = \max(\frac{1}{T}, 36\delta K\eta)$ *otherwise, and* $\eta = \min\left(\frac{1}{L}, \frac{1}{1926\delta K}, \frac{1}{K}\left(\frac{F^0}{7776\delta^2 T \sigma_{f-h}^2}\right)^{1/3}, \frac{1}{K}\sqrt{\frac{2F^0}{\gamma LT \sigma_h^2}}\right).$ $\boldsymbol{m}^0$ *such*

---

**Algorithm 5** AUXMVR$(f, h)$

---

**Require:** $\boldsymbol{x}_0, \boldsymbol{m}^0, \eta, a, T, K$

  $\boldsymbol{x}_{-1} = \boldsymbol{x}_0$

  **for** $t = 1$ to $T$ **do**

    Sample ~~$\xi^t_{f-h}$~~ $\xi^{t-1}_{f \sim h} \sim$

    ~~$\boldsymbol{g}^{f-h}_{prev} = \boldsymbol{g}_{f-h}(\boldsymbol{x}^{t-2}, \xi^t_{f-h})$~~ $\boldsymbol{g}^{f-h}_{prev} = \boldsymbol{g}_{f \sim h}(\boldsymbol{x}^{t-2}, \xi^{t-1}_{f \sim h})$

    ~~$\boldsymbol{g}^{f-h} = \boldsymbol{g}_{f-h}(\boldsymbol{x}^{t-1}, \xi^t_{f-h})$~~ $\boldsymbol{g}^{f-h} = \boldsymbol{g}_{f \sim h}(\boldsymbol{x}^{t-1}, \xi^{t-1}_{f \sim h})$

    $\boldsymbol{m}^t = (1-a)\boldsymbol{m}^{t-1} + a\boldsymbol{g}^{f-h} + (1-a)(\boldsymbol{g}^{f-h} - \boldsymbol{g}^{f-h}_{prev})$ §update momentum

    $\boldsymbol{y}^t_0 = \boldsymbol{x}^{t-1}$

    **for** $k = 0$ to $K-1$ **do**

      Sample $\xi^{t,k}_h$; Compute $\boldsymbol{g}_h(\boldsymbol{y}^t_k, \xi^{t,k}_h)$

      ~~$\boldsymbol{d}^t_k = \boldsymbol{g}_h(\boldsymbol{y}^t_k, \xi^{t,k}_h) + (1-a)\boldsymbol{m}^{t-1} + a\boldsymbol{g}^{f-h}_{prev}$~~ $\boldsymbol{d}^t_k = \boldsymbol{g}_h(\boldsymbol{y}^t_k, \xi^{t,k}_h) + \boldsymbol{m}^t$

      $\boldsymbol{y}^t_{k+1} = \boldsymbol{y}^t_k - \eta\boldsymbol{d}^t_k$

    **end for**

    ~~$\boldsymbol{m}^t = (1-a)\boldsymbol{m}^{t-1} + a\boldsymbol{g}^{f-h} + (1-a)(\boldsymbol{g}^{f-h} - \boldsymbol{g}^{f-h}_{prev})$ §update momentum~~ Update $\boldsymbol{x}^t = \boldsymbol{y}^t_K$

  **end for**

---

~~*that $E^0 \leq \sigma^2_{f \sim h}/T$, for $a = \max(\frac{1}{T}, 1156\delta^2 K^2\eta^2)$ and $\eta = \min\left(\frac{1}{L}, \frac{1}{192\delta K}, \frac{1}{K}\left(\frac{F^0}{18432\delta^2 T\sigma^2_{f \sim h}}\right)^{1/3}, \sqrt{\frac{F^0}{KT(L/2 + 8\delta K)}}\right)$.*~~

*This choice gives us the rate :*

$$\frac{1}{8KT}\sum_{t=1}^T\sum_{k=1}^{K-1}\mathbb{E}\big[\|\nabla f(\boldsymbol{y}^t_k)\|^2_2\big] \leq 2\sqrt{\frac{L\gamma F^0\sigma^2_h}{KT}} + 30\!\!\!40\Big(\frac{\delta F^0\sigma_{f-h}}{T}\Big)^{2/3} + \frac{(L+1926\delta K)F^0}{T}\,2\sqrt{\frac{(L/2+8\delta)F^0\sigma^2_h}{KT}} + \frac{3\sigma^2_{f-h}}{T}\,\frac{(L+192}{K}$$

*Where $F^0 = f(\boldsymbol{x}^0) - f^\star$ and* ~~$\gamma = \frac{1}{2K} + \frac{126\delta}{L}$.~~ $E^0 = \mathbb{E}\|\boldsymbol{m}^0 - (\nabla f(\boldsymbol{x}^0) - \nabla h(\boldsymbol{x}^0))\|^2$.

### C.3.3 ~~Change during each cycle~~ Proof

~~**Variance of $\boldsymbol{d}^t_k$.** Again~~

Given that the form of $\boldsymbol{d}^t_k$ is ~~not perfectly equal to $\nabla f(\boldsymbol{y}^t_{k-1})$ due to the use of $h$ instead of $f$ the function we are actually optimizing. In the following lemma, we quantify the error resulting from this.~~

~~Under assumption A3.3, we have the following inequality :~~

$$\mathbb{E}[\|\bar{\boldsymbol{d}}^t_k - \nabla f(\boldsymbol{y}^t_{k-1})\|^2_2] \leq 3\delta^2\Delta^t_{k-1} + 3E^{t-1} + 3a^2\sigma^2_{f-h}.$$

~~We have~~

$$\bar{\boldsymbol{d}}^t_k - \nabla f(\boldsymbol{y}^t_{k-1}) = \nabla h(\boldsymbol{y}^t_{k-1}) + (1-a)\boldsymbol{m}^{t-1} + a(\boldsymbol{g}_f(\boldsymbol{x}^{t-2}, \xi^{t-1}_f) - \boldsymbol{g}_h(\boldsymbol{x}^{t-2}, \xi^{t-1}_h)) - \nabla f(\boldsymbol{y}^t_{k-1})$$

$$= (1-a)\boldsymbol{e}^{t-1} + (1-a)(\nabla f(\boldsymbol{x}^{t-2}) - \nabla h(\boldsymbol{x}^{t-2}))$$

$$+ \nabla h(\boldsymbol{y}^t_{k-1}) + a(\boldsymbol{g}_f(\boldsymbol{x}^{t-2}, \xi^{t-1}_f) - \boldsymbol{g}_h(\boldsymbol{x}^{t-2}, \xi^{t-1}_h)) - \nabla f(\boldsymbol{y}^t_{k-1})$$

$$= (1-a)\boldsymbol{e}^{t-1} + a\big(\boldsymbol{g}_f(\boldsymbol{x}^{t-2}, \xi^{t-1}_f) - \boldsymbol{g}_h(\boldsymbol{x}^{t-2}, \xi^{t-1}_h) - \nabla f(\boldsymbol{x}^{t-2}) + \nabla h(\boldsymbol{x}^{t-2})\big)$$

$$+ \nabla h(\boldsymbol{y}^t_{k-1}) - \nabla h(\boldsymbol{x}^{t-2}) - \nabla f(\boldsymbol{y}^t_{k-1}) + \nabla f(\boldsymbol{x}^{t-2})$$

Using Lemma B.2 with $N = 3$, we get :

$$\mathbb{E}[\|\boldsymbol{d}_k^t - \nabla f(\boldsymbol{y}_{k-1}^t)\|_2^2] \leq 3\mathbb{E}[\|\nabla h(\boldsymbol{y}_{k-1}^t) - \nabla h(\boldsymbol{x}^{t-2}) - \nabla f(\boldsymbol{y}_{k-1}^t) + \nabla f(\boldsymbol{x}^{t-2})\|_2^2]$$

$$+3(1-a)^2\mathbb{E}[\|\boldsymbol{e}^{t-1}\|_2^2]$$

$$+3a^2\mathbb{E}[\|\boldsymbol{g}_f(\boldsymbol{x}^{t-2}, \xi_f^{t-1}) - \boldsymbol{g}_h(\boldsymbol{x}^{t-2}, \xi_h^{t-1}) - \nabla f(\boldsymbol{x}^{t-2}) + \nabla h(\boldsymbol{x}^{t-2})\|_2^2]$$

$$\leq 3\delta^2\Delta_{k-1}^t + 3E^{t-1} + 3a^2\sigma_{f-h}^2.$$

**Distance moved in each step.**

for $\eta \leq \frac{1}{6\delta K}$ we have:

$$\Delta_k^t \leq (1 + \frac{1}{K})\Delta_{k-1}^t + 18K\eta^2 E^{t-1} + 18K\eta^2 a^2\sigma_{f-h}^2 + 6K\eta^2\mathbb{E}[\|\nabla f(\boldsymbol{y}_{k-1}^t)\|_2^2] + \eta^2\sigma_h^2$$

Like for the proof of LemmaC.5, we get :

$$\Delta_k^t = \mathbb{E}[\|\boldsymbol{y}_k^t - \boldsymbol{x}^{t-2}\|_2^2]$$

$$= \mathbb{E}[\|\boldsymbol{y}_{k-1}^t - \eta\boldsymbol{d}_k^t - \boldsymbol{x}^{t-2}\|_2^2]$$

$$= \mathbb{E}[\|\boldsymbol{y}_{k-1}^t - \eta\bar{\boldsymbol{d}}_k^t - \boldsymbol{x}^{t-2}\|_2^2] + \eta^2\mathbb{E}[\|\boldsymbol{d}_k^t - \bar{\boldsymbol{d}}_k^t\|_2^2]$$

$$\leq (1 + \frac{1}{2K})\Delta_{k-1}^t + (2K+1)\eta^2\mathbb{E}[\|\bar{\boldsymbol{d}}_k^t\|_2^2] + \eta^2\sigma_h^2$$

$$= (1 + \frac{1}{2K})\Delta_{k-1}^t + 3K\eta^2\mathbb{E}[\|\boldsymbol{d}_k^t \pm \nabla f(\boldsymbol{y}_{k-1}^t)\|_2^2] + \eta^2\sigma_h^2$$

$$\leq (1 + \frac{1}{2K})\Delta_{k-1}^t + 6K\eta^2\mathbb{E}[\|\bar{\boldsymbol{d}}_k^t - \nabla f(\boldsymbol{y}_{k-1}^t)\|_2^2] + 6K\eta^2\mathbb{E}[\|\nabla f(\boldsymbol{y}_{k-1}^t)\|_2^2] + \eta^2\sigma_h^2$$

$$\leq (1 + \frac{1}{2K})\Delta_{k-1}^t + 6K\eta^2(3\delta^2\Delta_{k-1}^t + 3E^{t-1} + 3a^2\sigma_{f-h}^2) + 6K\eta^2\mathbb{E}[\|\nabla f(\boldsymbol{y}_{k-1}^t)\|_2^2] + \eta^2\sigma_h^2$$

$$= (1 + \frac{1}{2K} + 18\delta^2K\eta^2)\Delta_{k-1}^t + 18K\eta^2 E^t + 18K\eta^2 a^2\sigma_{f-h}^2 + 6K\eta^2\mathbb{E}[\|\nabla f(\boldsymbol{y}_{k-1}^t)\|_2^2] + \eta^2\sigma_h^2$$

The condition $\eta \leq \frac{1}{6\delta K}$ ensures $18\delta^2K\eta^2 \leq \frac{1}{2K}$ which finishes the proof.

**Progress in one step.** For $\eta \leq \min(\frac{1}{L}, \frac{1}{288\delta K})$, under assumptions A3.1 and A3.3, the following inequality is true:

$$\mathbb{E}\big[f(\boldsymbol{y}_k^t) + \delta(1 + \frac{2}{K})^{K-k}\Delta_k^t\big] \leq \mathbb{E}\big[f(\boldsymbol{y}_{k-1}^t) + \delta(1 + \frac{2}{K})^{K-(k-1)}\Delta_{k-1}^t\big] - \frac{\eta}{4}\mathbb{E}[\|\nabla f(\boldsymbol{y}_{k-1}^t)\|_2^2]$$

$$+2\eta E^{t-1} + 2\eta a^2\sigma_{f-h}^2 + (\frac{L}{2} + 8\delta)\eta^2\sigma_h^2.$$

Like Lemma C.6, the $L$-smoothness of $f$ gives us

$$E[f(\boldsymbol{y}_k^t) - f(\boldsymbol{y}_{k-1}^t)] \leq -\frac{\eta}{2}E[\|\nabla f(\boldsymbol{y}_{k-1}^t)\|_2^2] + \frac{\eta}{2}E[\|\boldsymbol{d}_k^t - \nabla f(\boldsymbol{y}_{k-1}^t)\|_2^2] + \frac{L\eta^2 - \eta}{2}E[\|\boldsymbol{d}_k^t\|_2^2] + \frac{L\eta^2}{2}\sigma_h^2.$$

Using $\eta \leq \frac{1}{L}$ we can get rid of the third term in the right-hand side of the above inequality the same as in AuxMOM (Algorithm 3), the same Lemmas still hold, except for Lemma C.8 which changes to the following Lemma.

Using LemmaC.4, we get :-

$$\mathbb{E}[f(\boldsymbol{y}_k^t) - f(\boldsymbol{y}_{k-1}^t)] \leq -\frac{\eta}{2}\mathbb{E}\big[\|\nabla f(\boldsymbol{y}_{k-1}^t)\|_2^2\big] + \frac{\eta}{2}\mathbb{E}\big[\|\boldsymbol{d}_k^t - \nabla f(\boldsymbol{y}_{k-1}^t)\|_2^2\big] + \frac{L\eta^2}{2}\sigma_h^2$$

$$\leq -\frac{\eta}{2}\mathbb{E}\big[\|\nabla f(\boldsymbol{y}_{k-1}^t)\|_2^2\big] + \frac{\eta}{2}(3\delta^2\Delta_{k-1}^t + 3E^{t-1} + 3a^2\sigma_{f-h}^2) + \frac{L\eta^2}{2}\sigma_h^2$$

$$= \frac{3}{2}\delta^2\eta\Delta_{k-1}^t + \frac{3}{2}\eta E^{t-1} + \frac{3}{2}\eta a^2\sigma_{f-h}^2 - \frac{\eta}{2}\mathbb{E}\big[\|\nabla f(\boldsymbol{y}_{k-1}^t)\|_2^2\big] + \frac{L\eta^2}{2}\sigma_h^2 .$$

Now we multiply LemmaC.5 by $\delta(1+\frac{2}{K})^{K-k}$. Note that $1 \leq (1+\frac{2}{K})^{K-k} \leq 8$.

$$\delta(1+\frac{2}{K})^{K-k}\Delta_k^t \leq \delta(1+\frac{2}{K})^{K-k}\big((1+\frac{1}{K})\Delta_{k-1}^t + 18K\eta^2 E^{t-1} + 18K\eta^2 a^2\sigma_{f-h}^2$$

$$+ 6K\eta^2\mathbb{E}[\|\nabla f(\boldsymbol{y}_{k-1}^t)\|_2^2] + \eta^2\sigma_h^2\big)$$

$$\leq \delta(1+\frac{2}{K})^{K-(k-1)}\Delta_{k-1}^t - \frac{\delta}{K}(1+\frac{2}{K})^{K-k}\Delta_{k-1}^t + 144K\delta\eta^2 E^{t-1} + 144K\delta\eta^2 a^2\sigma_{f-h}^2$$

$$+ 48K\delta\eta^2\mathbb{E}[\|\nabla f(\boldsymbol{y}_{k-1}^t)\|_2^2] + 8\delta\eta^2\sigma_h^2$$

$$\leq \delta(1+\frac{2}{K})^{K-(k-1)}\Delta_{k-1}^t - \frac{\delta}{K}\Delta_{k-1}^t + 144K\delta\eta^2 E^{t-1} + 144K\delta\eta^2 a^2\sigma_{f-h}^2$$

$$+ 48K\delta\eta^2\mathbb{E}[\|\nabla f(\boldsymbol{y}_{k-1}^t)\|_2^2] + 8\delta\eta^2\sigma_h^2$$

Adding the last two inequalities, we get :-

$$\mathbb{E}[f(\boldsymbol{y}_k^t)] + \delta(1+\frac{2}{K})^{K-k}\Delta_k^t \leq \mathbb{E}[f(\boldsymbol{y}_{k-1}^t)] + \delta(1+\frac{2}{K})^{K-(k-1)}\Delta_{k-1}^t$$

$$+ (\frac{3}{2}\delta^2\eta - \frac{\delta}{K})\Delta_{k-1}^t$$

$$+ (\frac{3}{2}\eta + 144K\delta\eta^2)E^{t-1}$$

$$+ (\frac{3}{2}\eta + 144K\delta\eta^2)a^2\sigma_{f-h}^2$$

$$+ (-\frac{\eta}{2} + 48K\delta\eta^2)\mathbb{E}[\|\nabla f(\boldsymbol{y}_{k-1}^t)\|_2^2]$$

$$+ (\frac{L}{2} + 8\delta)\eta^2\sigma_h^2$$

For $\eta \leq \frac{1}{288\delta K}$ we have $\frac{3}{2}\delta^2\eta - \frac{\delta}{K} \leq 0$, $\frac{3}{2}\eta + 144K\delta\eta^2 a^2 \leq 2\eta$ and $-\frac{\eta}{2} + 48K\delta\eta^2 \leq -\frac{\eta}{4}$ which gives the lemma.

**Distance moved in a cycle.**

For $\eta \leq \frac{1}{6K\delta}$ and under assumptions A3.1 and A3.3 with $G^t = \frac{1}{K}\sum_k \mathbb{E}\big[\|\nabla f(\boldsymbol{y}_k^t)\|_2^2\big]$, we have :-

$$\Delta^t\Big(:= \mathbb{E}\big[\|\boldsymbol{x}^t - \boldsymbol{x}^{t-1}\|_2^2\big]\Big) \leq 108K^2\eta^2\delta^2\Delta^{t-1} + 54K^2\eta^2 E^{t-1} + 54K^2\eta^2 a^2\sigma_{f-h}^2 + 18K^2\eta^2 G^t + 3K\eta^2\sigma_h^2$$

We follow the same strategy as in the proof of Lemma?? to prove that for $\eta \leq \frac{1}{12\delta K}$ we have :

$$\mathbb{E}[\|\boldsymbol{y}_k^t - \boldsymbol{x}^{t-1}\|_2^2] = \mathbb{E}[\|\boldsymbol{y}_{k-1}^t - \eta \boldsymbol{d}_k^t - \boldsymbol{x}^{t-1}\|_2^2]$$

$$\leq \mathbb{E}[\|\boldsymbol{y}_{k-1}^t - \eta \bar{\boldsymbol{d}}_k^t - \boldsymbol{x}^{t-1}\|_2^2] + \eta^2 \sigma_h^2$$

$$\leq (1 + \frac{1}{2K})\mathbb{E}[\|\boldsymbol{y}_{k-1}^t - \boldsymbol{x}^{t-1}\|_2^2] + (2K+1)\eta^2\mathbb{E}[\|\bar{\boldsymbol{d}}_k^t\|_2^2] + \eta^2\sigma_h^2$$

$$= (1 + \frac{1}{2K})\mathbb{E}[\|\boldsymbol{y}_{k-1}^t - \boldsymbol{x}^{t-1}\|_2^2] + 3K\eta^2\mathbb{E}[\|\bar{\boldsymbol{d}}_k^t \pm \nabla f(\boldsymbol{y}_{k-1}^t)\|_2^2] + \eta^2\sigma_h^2$$

$$\leq (1 + \frac{1}{2K})\mathbb{E}[\|\boldsymbol{y}_{k-1}^t - \boldsymbol{x}^{t-1}\|_2^2] + 6K\eta^2\mathbb{E}[\|\bar{\boldsymbol{d}}_k^t - \nabla f(\boldsymbol{y}_{k-1}^t)\|_2^2]$$

$$+ 6K\eta^2\mathbb{E}[\|\nabla f(\boldsymbol{y}_{k-1}^t)\|_2^2] + \eta^2\sigma_h^2$$

$$\leq (1 + \frac{1}{2K})\mathbb{E}[\|\boldsymbol{y}_{k-1}^t - \boldsymbol{x}^{t-1}\|_2^2] + 6K\eta^2(3\delta^2\Delta_{k-1}^t + 3E^{t-1} + 3a^2\sigma_{f-h}^2)$$

$$+ 6K\eta^2\mathbb{E}[\|\nabla f(\boldsymbol{y}_{k-1}^t)\|_2^2] + \eta^2\sigma_h^2$$

$$= (1 + \frac{1}{2K} + 36\delta^2 K\eta^2)\mathbb{E}[\|\boldsymbol{y}_{k-1}^t - \boldsymbol{x}^{t-2}\|_2^2] + 36K\eta^2\delta^2\Delta^{t-1} + 18K\eta^2 E^{t-1}$$

$$+ 18K\eta^2 a^2\sigma_{f-h}^2 + 6K\eta^2\mathbb{E}[\|\nabla f(\boldsymbol{y}_{k-1}^t)\|_2^2] + \eta^2\sigma_h^2$$

Where we used in the last inequality the fact :

$$\Delta_{k-1}^t = \mathbb{E}\left[\|\boldsymbol{y}_{k-1}^t - \boldsymbol{x}^{t-1}\|_2^2\right] \leq 2\mathbb{E}\left[\|\boldsymbol{y}_{k-1}^t - \boldsymbol{x}^{t-2}\|_2^2\right] + 2\Delta^{t-1} .$$

Using the condition $\eta \leq \frac{1}{12\delta K}$, we get :

$$\mathbb{E}\left[\|\boldsymbol{y}_k^t - \boldsymbol{x}^{t-1}\|_2^2\right] \leq (1 + \frac{1}{K})\mathbb{E}\left[\|\boldsymbol{y}_{k-1}^t - \boldsymbol{x}^{t-1}\|_2^2\right] + 36K\eta^2\delta^2\Delta^{t-1} + 18K\eta^2 E^{t-1} + 18K\eta^2 a^2\sigma_{f-h}^2$$

$$+ 6K\eta^2\mathbb{E}[\|\nabla f(\boldsymbol{y}_{k-1}^t)\|_2^2] + \eta^2\sigma_h^2$$

We use now the fact $\boldsymbol{x}^t = \boldsymbol{y}_K^t$.

$$\Delta^t = \mathbb{E}\left[\|\boldsymbol{y}_K^t - \boldsymbol{x}^{t-1}\|_2^2\right]$$

$$\leq \sum_k (1 + \frac{1}{K})^{K-k}\left(36K\eta^2\delta^2\Delta^{t-1} + 18K\eta^2 E^{t-1} + 18K\eta^2 a^2\sigma_{f-h}^2 + 6K\eta^2\mathbb{E}[\|\nabla f(\boldsymbol{y}_{k-1}^t)\|_2^2] + \eta^2\sigma_h^2\right)$$

$$\leq 108K^2\eta^2\delta^2\Delta^{t-1} + 54K^2\eta^2 E^{t-1} + 54K^2\eta^2 a^2\sigma_{f-h}^2 + 18K^2\eta^2\frac{1}{K}\sum_k \mathbb{E}[\|\nabla f(\boldsymbol{y}_k^t)\|_2^2] + 3K\eta^2\sigma_h^2$$

$$= 108K^2\eta^2\delta^2\Delta^{t-1} + 54K^2\eta^2 E^{t-1} + 54K^2\eta^2 a^2\sigma_{f-h}^2 + 18K^2\eta^2 G^t + 3K\eta^2\sigma_h^2 .$$

Where we used the fact that $(1 + \frac{1}{K})^{K-k} \leq 3$.

**Momentum variance of AUXMVR.** Here we will bound the quantity $E^t$.

**Lemma C.11.** *Under assumptions A4.5,A3.2 , we have :*

$$E^t \leq (1-a)E^{t-1} + 2\delta^2\Delta^{t-1} + 2a^2\sigma_{f-h}^2 .$$

Combining this inequality with Lemma C.7, we get for $\eta \leq \frac{1}{5K\delta}$ and $a \geq 144K^2\delta^2\eta^2$:

$$E^t \leq (1 - \frac{a}{2})E^{t-1} + 36K^2\delta^2\eta^2 G^{t-1} + 2a^2\sigma_{f-h}^2 + 6K\delta^2\eta^2\sigma_h^2 .$$

*Proof.* First, we notice that

$$
\begin{aligned}
e^t &= m^t - \nabla f(x^{t-1}) + \nabla h(x^{t-1}) \\
&= (1-a)m^{t-1} + a(g_{f-h}(x^{t-1}, \xi_f^{t-1}) - h^{(t-1, \xi_h^{t-1})}) \\
&\quad + (1-a)\Big(g_{f-h}(x^{t-1}, \xi^{t-1}_{f-h}) - g_{h-h}(x^{t-1}, \xi_h^{t-1}) - f^{(t-2, \xi^{t-1}_{f-h})} + h^{(t-2, \xi_h^{t-1})}\Big) - \nabla f(x^{t-1}) + \nabla h(x^{t-1}) \\
&= (1-a)e^{t-1} + a(g_{f-h}(x^{t-1}, \xi) - h^{(t-1, \xi_h^{t-1}_{f-h})} - \nabla f(x^{t-1}) + \nabla h(x^{t-1})) \\
&\quad + (1-a)\big(g_{f-h}(x^{t-1}, \xi^{t-1}_{f-h}) - \nabla f(x^{t-1}) - h^{(t-1, \xi_h^{t-1})} + \nabla h(x^{t-1}) - g_{f-h}(x^{t-2}, \xi^{t-1}_{f-h}) \\
&\quad + \nabla f(x^{t-2}) + h^{(t-2, \xi_h^{t-1})} - \nabla h(x^{t-2}))
\end{aligned}
$$

Notice that $e^{t-1}$ is independent of the rest of the formulae which is itself centered (has a mean equal to zero), so :

$$E^t \leq (1-a)^2 E^{t-1} + 2a^2\sigma_{f-h}^2 + 2\delta^2\Delta^{t-1} .$$

**Progress in one round.** Under the same assumptions as in Lemma??, we have :

$$\frac{\eta}{4}G^t \leq \frac{F^{t-1} - F^t}{K} + \frac{8\delta}{K}\Delta^{t-1} + 2\eta E^{t-1} + 2\eta a^2\sigma_{f-h}^2 + (\frac{L}{2} + 8\delta)\eta^2\sigma_h^2 .$$

We use the inequality established in Lemma??, which can be rearranged in the following way : Now for $\eta \leq \frac{1}{5K\delta}$, we can use Lemma C.7 and we get

$$\frac{\eta}{4}E^t \leq f^{(t}_{k-1}1-a)E^{t-1} + 2a^2\sigma_{f-h}^2 + 2\delta(1^2\big(36K^2\eta^2 E^t + \frac{2}{K}\big)^{K-(k-1)}\Delta 18K^2\eta^2 G^t_{k-1} - f^{(t}_{k)} + \delta 3K\eta^2\sigma_h^2\big)$$

$$\leq (11 - a + \frac{2}{K})^{K-k}\Delta_k^t + 274K^2\delta^2\eta^2)E^{t-1} + 2\eta a^2\sigma_{f-h}^2 + (\frac{L}{2}36K^2\delta^2\eta^2 G^{t-1} + 86K\delta)^2\eta^2\sigma_h^2 .$$

We sum this inequality from $k=1$ to $k=K$, this will give:

$$\frac{K\eta}{4}G^t \leq \mathbb{E}\big[f(y_0^t) + \delta(1 + \frac{2}{K})^K\Delta_0^t\big] - \Big(\mathbb{E}\big[f(y_K^t) + \delta\Delta_K^t\big]\Big) + 2\eta K E^{t-1} + 2\eta K a^2\sigma_{f-h}^2 + K(\frac{L}{2} + 8\delta)\eta^2\sigma_h^2 .$$

We note that $y_0^t = x^{t-1}$ and $y_0^t = x^{t-1}$, which means this time that $\Delta_0^t = \Delta^{t-1}$. So we have :

$$\frac{\eta}{4}G^t \leq \frac{F^{t-1} - F^t}{K} + \frac{8\delta}{K}\Delta^t + 2\eta E^{t-1} + 2\eta a^2\sigma_{f-h}^2 + (\frac{L}{2} + 8\delta)\eta^2\sigma_h^2 .$$

It is easy to verify that for $a \geq 144K^2\delta^2\eta^2$, we have $74K^2\delta^2\eta^2 \leq a/2$ which finishes the proof. □

Let's derive now now derive the convergence rate of AUXMVR.

We have :

$$
\begin{cases}
\frac{\eta}{4}G^t \leq \frac{F^{t-1} - F^t}{K} + 2\eta E^t + (\frac{L}{2} + 8\delta)\eta^2\sigma_h^2 , \\
E^t \leq (1 - \frac{a}{2})E^{t-1} + 36K^2\delta^2\eta^2 G^{t-1} + 2a^2\sigma_{f-h}^2 + 6K\delta^2\eta^2\sigma_h^2 .
\end{cases}
$$

We will add to both sides of the first inequality the quantity $\frac{\beta\eta}{a}E^t + \left(\frac{\gamma\eta}{a} + \frac{\alpha\delta}{K}\right)\Delta^t$ for $\alpha, \beta, \gamma$ positive numbers to be defined later. So :

$$\frac{\eta}{4}G^t + \frac{F^t}{K} + \frac{\beta\eta}{a}E^t + \left(\frac{\gamma\eta}{a} + \frac{\alpha\delta}{K}\right)\Delta^t$$

$$\leq \frac{F^{t-1}}{K} + \frac{\beta\eta}{a}E^{t-1} + (2 + 54\frac{\gamma K^2\eta^2}{a} + 54\alpha\delta K\eta - \beta)E^{t-1}$$

$$+ \frac{\gamma\eta}{a}\Delta^{t-1}\left(\frac{2\beta\tilde\delta^2}{\gamma} + 108K^2\eta^2\delta^2\right) + \frac{\alpha\delta}{K}\Delta^{t-1}\left(\frac{8}{\alpha} + 108K^2\eta^2\delta^2\right)$$

$$+ (2\beta a + 54\alpha K^2\eta^2 + 54\alpha\delta K\eta^2 a + 2)\eta a\sigma_{f-h}^2$$

$$+ \left(\frac{18\delta^2 K^2\eta^3}{a} + 18\alpha\delta K\eta^2\right)G^{t-1}$$

$$+ (L/2 + 8\delta + \frac{\gamma\eta}{a} + \frac{\alpha\delta}{K})\eta^2\sigma_h^2,$$

$\frac{4\eta}{a}E^t$.

So :

$$\frac{\eta}{4}G^t + \frac{4\eta}{a}E^t \leq \frac{F^{t-1} - F^t}{K} + 2\eta E^t + (\frac{L}{2} + 8\delta)\eta^2\sigma_h^2 + \frac{4\eta}{a}\left((1 - \frac{a}{2})E^{t-1} + 36K^2\delta^2\eta^2 G^{t-1} + 2a^2\sigma_{f-h}^2 + 6K\delta^2\eta^2\sigma_h^2\right),$$

We choose $\alpha, \beta, \gamma$ such that : Let's define the potential $\Phi^t = \frac{F^t}{K} + (\frac{4\eta}{a} - 2\eta)E^t \leq \frac{F^t}{K} + \frac{4\eta}{a}E^t$.

Then

$$\frac{\eta}{4}G^t \leq \Phi^{t-1} - \Phi^t + (\frac{L}{2} + 8\delta)\eta^2\sigma_h^2 + \frac{144K^2\delta^2\eta^3}{a}G^{t-1} + 8a\eta\sigma_{f-h}^2 + \frac{24K\delta^2\eta^3}{a}\sigma_h^2,$$

It is easy to show that for $\eta \leq \frac{1}{1926\delta K}$ and $a \geq 144\gamma K^2\eta^2$ we can take $\beta = 3$, $\gamma = 9\delta^2$ and $\alpha = 9$ to satisfy all the above inequalities. This means :

$$\frac{\eta}{4}G^t - \frac{\eta}{8}G^{t-1} \leq \Phi^{t-1} - \Phi^t + 3\eta a\sigma_{f-h}^2 + (L/2 + 8\delta + \frac{9\delta^2\eta}{a} + \frac{9\delta}{K})\eta^2\sigma_h^2,$$

for $a \geq 1152K^2\delta^2\eta^2$, we have $\frac{144K^2\delta^2\eta^3}{a} \leq \eta/8$, which means

$$\frac{\eta}{4}G^t - \frac{\eta}{8}G^{t-1} \leq \Phi^{t-1} - \Phi^t + (\frac{L}{2} + 8\delta)\eta^2\sigma_h^2 + 8a\eta\sigma_{f-h}^2 + \frac{24K\delta^2\eta^3}{a}\sigma_h^2,$$

For a potential $\Phi^t = \frac{F^t}{K} + \frac{3\eta}{a}E^t + 9\left(\frac{\tilde\delta^2\eta}{a} + \frac{\delta}{K}\right)\Delta^t$.

Summing the inequality ?? Summing the last inequality over $t$ , gives : gives :

$$\frac{1}{8T}\sum_{t=1}^{T}G^t \leq \frac{\Phi^0}{\eta T} + 38 a\sigma_{f-h}^2 + (L/2 \frac{L}{2} + 8\delta + \frac{9\delta^2\eta}{a} + \frac{9\delta}{K})\eta\sigma_h^2 + \frac{24K\delta^2\eta^2}{a}\sigma_h^2,$$

$$\leq \frac{F^0}{\eta K T} + \frac{3}{aT}\frac{4}{aT}E^0 + 9\frac{\delta^2}{aT} + \frac{\delta}{\eta K T}\Delta^0 + 38 a\sigma_{f-h}^2 + (L/2 \frac{L}{2} + 8\delta + \frac{9\delta^2\eta}{a} + \frac{9\delta}{K})\eta\sigma_h^2 + \frac{24K\delta^2\eta^2}{a}\sigma_h^2,$$

~~Note that $\Delta^0 = 0$. If we use a batch $T$ times larger at the beginning, we can ensure $E^0 \leq \frac{\sigma_f^2 + \sigma_h^2}{T}$, so:~~

If we use a batch $T$ times larger at the beginning, we can ensure $E^0 \leq \frac{\sigma_{f-h}^2}{T}$, so:

$$\frac{1}{8T}\sum_{t=1}^{T} G^t \leq \frac{F^0}{\eta KT} + \frac{3\sigma_{f-h}^2}{aT^2}\frac{4\sigma_{f-h}^2}{aT^2} + 38 a\sigma_{f-h}^2 + (L/2\frac{L}{2} + 8\delta + \frac{9\delta^2\eta}{a} + \frac{9\delta}{K})\eta\sigma_h^2 + \frac{24K\delta^2\eta^2}{a}\sigma_h^2 ,$$

~~Now taking $a = \max(\frac{1}{T}, 1296\delta^2 K^2\eta^2)$ if $\sigma_h = 0$ and $a = \max(\frac{1}{T}, 36\delta K\eta)$ otherwise, we get :~~

Now taking $a = \max(\frac{1}{T}, 1152\delta^2 K^2\eta^2)$, we get :

$$\frac{1}{8T}\sum_{t=1}^{T} G^t \leq \frac{F^0}{\eta KT} + 3888\frac{4\sigma_{f-h}^2}{T} + \frac{8\sigma_{f-h}^2}{T} + 9216\delta^2\eta^2 K^2\sigma_{f-h}^2 + (L/2\frac{L}{2} + 1268\delta K)\eta\sigma_h^2 + \frac{3\sigma_{f-h}^2}{T}\frac{\sigma_h^2}{48K} ,$$

Taking into account all the conditions on $\eta$ that were necessary, we can take :

$$\eta = \min\left(\left(\frac{1}{L}, \frac{1}{1926\delta K}\frac{1}{192\delta K}, \frac{1}{K}\left(\frac{F^0}{7776\delta^2 T\sigma_{f-h}^2}\frac{F^0}{18432\delta^2 T\sigma_{f-h}^2}\right)^{1/3}, \sqrt{\frac{F^0}{KT(L/2 + 126\delta K)}}\sqrt{\frac{F^0}{KT(L/2 + 8\delta K)}}\right)\right).$$

This choice gives us the rate :

$$\frac{1}{8T}\sum_{t=1}^{T} G^t \leq 2\sqrt{\frac{(L/2 + 126\delta K)F^0}{KT}} + 3040\left(\frac{\delta F^0\sigma_{f-h}}{T}\right)^{2/3} + \frac{(L + 1926\delta K)F^0}{KT}2\sqrt{\frac{(L/2 + 8\delta)F^0}{KT}} + \frac{3\sigma_{f-h}^2}{T}\frac{(L + 192\delta K)F^0}{KT} + \frac{8\sigma}{}$$

## C.4 Generalization to multiple decentralized helpers.

We consider now the case where we have $N$ helpers: $h_1, \ldots, h_N$. This case can be easily solved by merging all the helpers into one helper $h = \frac{1}{N}\sum_{i=1}^{N} h_i$ for example (it is easy to see that if each $h_i$ is $\delta_i$-BHD from $f$ that their average $h$ would be $\frac{1}{N}\sum_{i=1}^{N} \delta_i$-BHD from f ). However, this is not possible if the helpers are decentralized (are not in the same place and cannot be made to be for privacy reasons for example). For this reason, we consider a Federated version of our optimization problem in the presence of auxiliary decentralized information.

In this case, we consider that all functions $h_i$ are such that :

$$\forall \boldsymbol{x}, \xi \ : \ \|\nabla^2 f(\boldsymbol{x}) - \nabla^2 h_i(\boldsymbol{x})\|_2 \leq \delta .$$

We will also need an additional assumption on $f$ :

**Assumption C.12. (Weak convexity.)** $f$ is $\delta$- weakly convex i.e. $\boldsymbol{x} \mapsto f(\boldsymbol{x}) + \delta\|\boldsymbol{x}\|_2^2$ is convex.

**Lemma C.13.** *Under Assumption C.12 the following is true :*

$$\forall N\forall \boldsymbol{x}, \boldsymbol{x}_1, \ldots, \boldsymbol{x}_N\forall \alpha \geq \delta \ : \ f(\frac{1}{N}\sum_{i=1}^{N} \boldsymbol{x}_i) + \alpha\|\frac{1}{N}\sum_{i=1}^{N} \boldsymbol{x}_i - \boldsymbol{x}\|_2^2 \leq \frac{1}{N}\sum_{i=1}^{N}\left(f(\boldsymbol{x}_i) + \alpha\|\boldsymbol{x}_i - \boldsymbol{x}\|_2^2\right).$$

**About the need for Assumption C.12.** Assumption C.12 becomes strong for $\delta$ small which is not good, as this should be the easiest case. however, we would like to point out that we only need Assumption C.12 to deal with the averaging that we perform to construct the new state $\boldsymbol{x}^t$. In the case where we sample each time one and only one helper (i.e. $S = 1$), we don't need such an assumption. This assumption was made in the context of Federated Learning for example in (Karimireddy et al., 2020a).

### C.4.1 Decentralized momentum version

As in C.2.1 we start from $\boldsymbol{x}^{t-1}$, we sample ~~(randomly) a set $S^t$ of $S$ helpers, we sample $\xi_f^t$~~ , compute $\boldsymbol{g}_f(\boldsymbol{x}^{t-1}, \xi_f^t)$~~and then~~, share it with all of the helpers ~~$h_i$ which will construct $\boldsymbol{m}_i^

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

---