# OpenReview forum: "Optimization with Access to Auxiliary Information"
_TMLR — Accepted by TMLR_

### Review · Reviewer_UJ5V · 2023-10-04

**Summary Of Contributions:**

This paper studies minimizing f(x) aided with an auxiliary function h(x) that correlates with f(x) but can be cheaper to obtain. Building on a two-loop meta algorithm extended from SVRG, the proposed approaches smartly correct the bias induced with h(x) using either momentum of momentum based 'variance reduction'. Theoretical results suggest the possible benefits of proposed approaches, and numerical results further support the theories.

**Audience:**

Yes

**Broader Impact Concerns:**

NA.

**Claims And Evidence:**

Yes

**Requested Changes:**

Besides the weakness above, this paper can also benefit from an improved presentation. Some of typos are listed below.

- Grammar error at one line above sec. 4.2, 'Hence approaches that try to reduce it like [cited work]'.

- Missing '.' in the discussion of region II in AuxMom.

- Second sentence in Sec. 3, --> We will consider mainly two approaches. The first one ....

**Strengths And Weaknesses:**

=======**Strength:**=======

S1. This paper extends variance reduction to a more general form, enabling almost plug-and-play application for various settings in machine learning.

S2. Theoretical results illustrate the inefficiency of 'naive approach,' which motivates well the proposed algorithms. A detailed discussion on the choice of $K$ is also provided.

S3. The alignment between theoretical findings and experimental results support well effectiveness of the proposed research.

=======**Weakness:**=======

W1. While the paper builds the gradient estimate for $f(x)$ based on SVRG-type estimates, it does not provide a justification for choosing SVRG over SARAH-type gradient estimators, as other alternatives exist [1, 2, 3].

W2. This paper develops two approaches, AuxMom and AuxMVR, but it lacks clarification regarding when to opt for AuxMom over AuxMVR or vice versa within the given context.

W3. The numerical results presented in the paper appear to be somewhat simple in supporting of the claimed applicability across a broad range of machine learning tasks. This paper can benefit from a more diverse set of experiments.


======= References =======

[1] LM. Nguyen, J. Liu, K. Scheinberg, and M. Takáč. "SARAH: A novel method for machine learning problems using stochastic recursive gradient." In International conference on machine learning, pp. 2613-2621. PMLR, 2017.

[2] B. Li, M. Ma, and G. B. Giannakis. "On the convergence of SARAH and beyond." In International Conference on Artificial Intelligence and Statistics, pp. 223-233. PMLR, 2020.

[3] Z. L, H. Bao, X. Zhang, and P. Richtárik. "PAGE: A simple and optimal probabilistic gradient estimator for nonconvex optimization." In International conference on machine learning, pp. 6286-6295. PMLR, 2021.

---

> ### Author Response · Authors · 2023-10-05
>
> Dear Reviewer,
> We would like first to thank you for your time, expertise, and feedback.
>
> We will try to address the weaknesses that you raised about the paper. We hope that our answers address all raised issues, and if not, please don't hesitate to point it out or ask additional questions.
>
> - **W1**. Why SVRG and not SARAH or Spider or other variance reduction techniques?:
>    - The first reason is that from the decomposition $f= h + f - h$ we get something that looks like SVRG (but that is not it exactly).
>    - The second reason is that in the noiseless case (when we can access the whole gradient of $f$) we don't get any theoretical improvement when using the SARAH-like update compared to the SVRG-like update, in both cases, we get a rate $\mathcal{O}((L/K + \delta)\frac{f(x_0)-f^\star}{T})$.
>    - The third reason is that we don't know how to treat the case where we only have noisy estimates of the gradient of the main function $f$. We note that in this case, our paper proposes to use the momentum of $f-h$ (if we use just use $f$ we don't get a theoretical improvement anymore).
>    - We will try to explain why we don't expect to get any improvement from using the SARAH-like update. This update is (in the absence of noise) $g^k_t = \nabla h(y^t_k) - \nabla h(y^t_{k-1}) + g^t_{k-1}$,  because $h$ is potentially different from $f$, we have an additional bias on top of the biasedness of SARAH and because of this we lose what SARAH gains over SVRG. We can explain this more if needed.
>
> - **W2.** When to use AuxMom vs AuxMVR? we should have said this more clearly in the paper, AuxMVR needs a stronger similarity assumption to hold between $f$ and $h$ (we need the $\|\nabla^2 f(\cdot,\zeta_f) - \nabla^2 h(\cdot,\zeta_h)\|\leq 2\delta$ for all $\zeta_f,\zeta_h$) and so AuxMVR has a stronger rate but needs stronger assumptions. We mainly included it to show that we can compete with the lower bound for first-order methods for functions with individual smoothness (meaning $f(\cdot,\zeta)$ is smooth for any $\zeta$) (reached by STORM which we call here MVR).
>
> - **Using a helper we can beat the known lower bounds.** One theoretical result we have in the paper is that by using a helper function we can beat the known lower bounds, the one reached by SGD for smooth functions and the one reached by MVR for "individual smooth functions".
>
> - **Additional experiments.** Our work is mainly theoretical, but we are open to providing additional experiments if you think that it is absolutely needed. We note that for Federated Learning AuxMom is similar to FederatedSVRG (for $a=1$ a is the momentum parameter, they are the same), it is also very close to MiMeSGD (from the MiMe paper) and thus the experiments should give similar results.
>
> - **Presentation issues.** We are very open to improving the clarity of the paper, we would really appreciate it if you could tell us more about what parts were unclear and need rewriting and if you have any suggestions on how we should do this.

---

> > ### Author Response · Authors · 2023-10-09
> >
> > We forgot to mention that we can use the idea from PAGE of updating the parameters of the main function $f$ with a probability $p$ instead of once each $K$ iterations. In this case, we can simply think of $p$ as equal to $\frac{1}{K}$, if we do this, then our theory still holds without needing additional changes.

---

> > > ### Comment · Reviewer_UJ5V · 2023-10-18
> > > **Thanks for your responses.**
> > >
> > > The technical concerns have been well addressed.
> > >
> > > Regarding numerical results, it would be helpful for add more since at least 6 applications are claimed to fit in this framework, but less than 3 are tested.
> > >
> > > Moreover,  I still believe this paper can benefit from improved writing. Here are some examples with grammar errors or typos. The authors need to go over the paper again and minimize these easily fixed issues.
> > >
> > > - "We will consider mainly two approaches, the first one we call the Naive approach and the second one we
> > > refer to as Bias correction" --> missing conjunctions.
> > > - In the first contribution,  'iv) training with sparse models' --> **vi)** training ...
> > > - In the third contribution, 'Based on the above trick, We design ' --> ... trick, **we** design  ..

---

> > > > ### Author Response · Authors · 2024-01-01
> > > >
> > > > Dear reviewer,
> > > >
> > > > We are writing to inform you that we have uploaded a new version of our paper where we tried to address all your concerns.
> > > > We also added a new experiment on Coresets. The theory also was updated slightly.
> > > >
> > > > We would be very happy to hear your opinion about the new version.

---

### Review · Reviewer_FjHB · 2023-10-15

**Summary Of Contributions:**

The paper dives into optimizing a function $f$ via optimizing auxiliary side functions $h$ whose gradients are either more available or cheaper to compute. Such an idea is very interesting and has beneficial applications.

**Audience:**

Yes

**Broader Impact Concerns:**

I have no concerns regarding the ethical implications of this work.

**Claims And Evidence:**

Yes

**Requested Changes:**

First of all, please make sure to address the weaknesses raised in the previous sections. As for the additional experiment:
In the field of subset selection, the fact that a subset is being sampled with provable guarantees (for example see [1]), implies that the trained model is different from the same model if it was trained on the entire data. An open question in such a field is whether one can train a model on subsets in some fashion and yet retrieve a model very close (inference-related) to a model being trained on the entire data. For example, training a model on $5\%$ of the data is very beneficial from a computational point of view, however, all subset selection techniques result in a major drop in accuracy when using such a ratio. My question here is as follows: Can you make an experiment in such a setting showing that using your technique, subset selection technique will obtain less drop in accuracy, thus making your paper an essential tool in the field of subset selection?



    [1] Tukan, M., Zhou, S., Maalouf, A., Rus, D., Braverman, V. &amp; Feldman, D.. (2023). Provable Data Subset Selection For Efficient Neural Networks Training. <i>Proceedings of the 40th International Conference on Machine Learning</i>, in <i>Proceedings of Machine Learning Research</i> 202:34533-34555 Available from https://proceedings.mlr.press/v202/tukan23a.html.

**Strengths And Weaknesses:**

Below are the strengths and weaknesses I have found in the paper:

* Strengths:
  * The idea of optimizing a function $f$ via directly handling auxiliary functions $h$ is an essential problem in federated learning, data subset selection techniques (e.g., coresets), etc.
  * Careful analysis of the results and the idea of Hessian similarity is intriguing.
  * Variety of applications of this problem, considering this work to be the first of its line focusing on such optimization problem.


* Weaknesses:
   * While the paper has strong implications, the writing is done poorly. At the simplest:
      -  Use commas.
      -  After a dot, start with capital letters.
      - When using abbreviations, state them beforehand, what do they stand for?
      - There are redundant words such as "in in" (see page 7).
   * When using specific data, either state it is synthetic or simply cite it. For example, the "mushrooms" dataset -- where did you get this data? Was it from the UCI repository? If so, did you apply any preprocessing?
   * What does the shaded region in the graph stand for? Is it standard deviation while the lines represent the mean?
   * I believe the experimental section is lacking an experiment in the field of subset selection (see the section below).

---

> ### Author Response · Authors · 2023-10-27
>
> Dear Reviewer,
> We would like to thank you for your time, expertise, and feedback.
>
> - **Writing issues**: we will improve the quality of the writing, if you have any specific suggestions on how to do this or if you think some parts of the paper are not clear enough, please do not hesitate to mention it, it will be of great help to us.
>
> - **Mushrooms dataset**: we took the dataset from libsvmtools here [mushrooms dataset](https://www.csie.ntu.edu.tw/~cjlin/libsvmtools/datasets/binary/mushrooms). The dataset is already preprocessed (Basically, it is the one from the UCI repository  + replacing text features with binary features).
>
> - **Shaded region in our graphs**: Yes, it is exactly as you said. The shaded region is +/- one standard deviation and the lines represent the mean over 10 runs.
>
> -**Corsets experiment**: We had an experiment on this that we did not include. Our experiment is basic, we select a small subset at random and use it as the helper $h$, in our case we did not observe any degradation in performance, but we would like to note that do not use the gradients of the small subset only but we couple them (using AuxMom) with gradient estimates of the main objective $f$ (in this case, the whole dataset), this means that we keep using $f$ to guide the training (make sure the bias introduced from using $h$ is limited).
>
> We will add this experiment.

---

> ### Comment · Reviewer_FjHB · 2023-11-14
> **Thanks**
>
> Dear authors,
>
> Thanks for replying.
>
> **Concerning the writing**: please see the four bullets above regarding the writing.
>
> **As for the experiment**: It will greatly highlight the use of your approach in the realm of subset selection, and minimize the gap between training on full data and training on subsets of the data.

---

> > ### Author Response · Authors · 2024-01-01
> > **Coreset experiment**
> >
> > Dear reviewer,
> > We are writing to inform you of the new version of our paper.
> > We added an experiment on Coresets. It is a basic experiment where we randomly sampled one-fifth of the training data and used it as a Coreset; then, we compared using the corset alone or correcting for the potential bias using AuxMOM (Figure 7).
> > In the experiment, using the corset alone loses approximately $2$ percent accuracy, but with AuxMOM, this is not the case.
> > We also had an experiment (which is not included in the paper for now) where we resample the Coreset instead of using AuxMOM; this performed similarly to AuxMOM.
> >
> > If you want us to try something else, do not hesitate to ask.

---

> ### Comment · Reviewer_FjHB · 2024-01-05
> **Thanks**
>
> Dear authors,
>
> Thanks for adding the experiments I had previously asked for. While the results are favorable, the basis is weak because the result is limited only to MNIST. It would be great if it is possible to obtain results concerning CIFAR100 where the coreset training attains a higher gap from the full training (a higher drop in accuracy), for example in the regime of coreset ratio of 5% of the data or even 10%.
>
> Note that MNIST is quite a simple dataset that coresets are known to handle well, while CIFAR10/100, ImageNet, etc. are much harder to deal with.
>
>  With that being said, concerning CIFAR100, it would be interesting to witness that with AuxMOM, even with small coreset sizes (e.g., 5%), it is possible to minimize the accuracy drop to be very small, attaining higher accuracy than without using AuxMOM or even resampling the coreset.
>
>
> As for the other experiments you have mentioned, please make sure to add them to the final version of the paper.
> Finally, would resampling behave the same as AuxMOM even on CIFAR100?

---

> ### Author Response · Authors · 2024-01-08
>
> We added two experiments using the CIFAR10 and CIFAR100 datasets (see Figure 8). In our experiments, we considered very simple convolutional networks and a constant step-size strategy, which means we shouldn't reach state-of-the-art results; we also considered a random coreset of 25% size. In this setting, we see the same results that were observed for the MNIST experiment, although this time the performance obtained by Coreset is much weaker compared to when the whole dataset is used.
>
> We will try a smaller coreset size.

---

> > ### Author Response · Authors · 2024-01-09
> >
> > Dear Reviewer,
> >
> > We tried different coreset sizes and compared them in Figure 9.
> > While the corset size has a big effect on the accuracy reached when training on the coreset alone, for AuxMOM, it seems it only affects the number of steps needed to reach training on the whole dataset.
> >
> > The last experiments show that our method is promising for replacing training on coresets alone.

---

> > > ### Comment · Reviewer_FjHB · 2024-01-09
> > > **Great progress**
> > >
> > > Dear authors,
> > >
> > > Thanks for adding the requested graphs. The results show that your technique is indeed improving the coreset-associated classification qualities. With this being said, how much time in practice does the AuxMOM add to the training with coresets? Is it negligible? Please add a small paragraph in the paper discussing this.

---

> > > > ### Author Response · Authors · 2024-01-10
> > > >
> > > > Dear Reviewer,
> > > >
> > > > The only additional time AuxMOM needs is the time necessary for sampling a new batch from the original dataset, computing its gradient, and updating the momentum used by AuxMOM; it is negligible in practice.
> > > > One disadvantage to using AuxMOM for training with coresets is that we need to keep some access to the original dataset to sample gradients from it used to update the momentum.
> > > >
> > > > We will add a paragraph to discuss this.

---

### Review · Reviewer_ZsY4 · 2023-12-21

**Summary Of Contributions:**

Paper studies minimization of a function f whose stochastic gradients are expensive or unreliable to compute, while at the same time we have access to an auxiliary function h whose gradients are easier or more reliable to compute. Crucially, authors assume that f and h satisfy a Hessian similarity property, which is known in Federated Learnig(FL)/Decentralized Optimization literature.

Authors want to study algorithms which alternate between 1 step and K-1 steps of gradient descent steps using f and h respectively. If h is easier to compute average iteration speed could be faster. However, they show that this naive method fails under their assumption. To fix this they employ a bias correction term in the updates using the h. Further since they assume access to only stochastic gradients, bias correction is done by estimating the bias using momentum method (AuxMom). They also study a momentum-based variance reduction (known technique) like algorithm (AuxMVR) which requires additional assumption of per sample Hessian similarity. These are algorithms are shown to converge, and their converge rate is better than purely optimizing over f in some regimes. Finally, authors also provide an algorithm and results for the case with decentralized auxiliary loss for handling problems like FL under additional very strict assumption.

Then authors provide some motivating problems from machine learning as transfer learning, learning from coreset, pre-training, and semi-supervised learning. Some of these example seemed a bit far fetched. Authors also note when solving a linear/logistical regression with noisy labels, their setting arises with perfectly matching Hessians. Finally, the paper provides some experimental results to verify the the practical gains of their algorithm on synthetic problems, learning from noisy data, and transfer learning. Authors note that they leave any in-depth study characterization of useful objective similarity in real applications to future work. Discussion section is fair and it raises some good points. However, when reading paper initially it wasn’t clear it was tackling problems with auxiliary losses with similar enough Hessians.

**Audience:**

Yes

**Broader Impact Concerns:**

None as this is mostly a theoretical work

**Claims And Evidence:**

Yes

**Requested Changes:**

I am hoping that the authors can address the following issues. Please see the weaknesses above for details and note that some sub-issues are marked as minor.
1. Address the issues with the theoretical results and algorithm
2. Address any potential over-claiming and cite relevant work at appropriate places
3. Address questions about experiments
4. Proof reading the mathematical portions and fixing imprecise/contradictory statements

Based on 1 and 3 I am happy to change my answer to “claims and evidence” question.

**Strengths And Weaknesses:**

# Strengths

This is a nice paper combining known techniques for a novel special case of optimization problem with an auxiliary objective with similar Hessian. They also motivate the problem well with many instances from ML. Authors provide intuitive algorithms and theoretical results seem plausible (see weaknesses and questions for issues). Experiments are mostly convincing (see weaknesses and questions for concerns). Application wise, the algorithms studied in this paper can potentially improve the efficiency of solving some ML problems. Authors also seem to have put some effort in providing the intuition behind their algorithms and making the proofs parsable (both scan be improved). AFAIK, this optimization angle of general auxiliary objectives was not considered in any prior work.

# Weaknesses

1. Algorithmic of analysis innovation is very limited as most of building blocks are known in very similar setting (special cases). However, many times authors fail to refer prior work when first introducing these concepts. For example, it not clear if the bias reduction trick can be called a contribution since it has appeared in many prior work especially in incremental gradient methods and FL. It would be better if this trick is attributed to prior work, otherwise there should be proper justification for its novelty. In another example, history of hessian similarity was never discussed. It is a widely studied assumption in FL and distributed optimization. These could be misleading to the readers.

2. First few sections of the paper are very confusing and is written to imply that the problem that they study is very general. Only after reading the 5th page (!) we get to the actual problem under consideration which requires smoothness and Hessian similarity. If the objective of the paper was to study auxiliary information in general there I would have expected more justification given for their assumptions and verifying if they are satisfied in real world problem. Yes, at the last sections authors note that their assumptions may not be satisfied in real-world problems, but similar candor could be used in initial sections too.

3. Various assumptions are rather strong, especially the $\delta$-Hessian similarity and weak convexity with precisely the $\delta$ factor for decentralized setting.

4. There are a few contradictory or imprecise statements in the paper.
  a. For example
    i. ”we don’t explicitly make assumptions on how the target function f is related to the auxiliary side information h (potentially a set of functions) like in Distributed optimization or Federated learning where we assume f is the average of the side-information h.”
    ii.  “Hence we need to assume some similarity between f and h. In our case, we propose to use the hessian similarity (defined in assumption 3.3)”
  It might be better to just say assumptions are weaker than some FL assumptions.
  c. In another example, it is not clear how alternating between K-1 steps of h update and 1 step of f update is FedAvg. Usually FedAvg, doesn’t do any f update.
  d. “this shows that hessian similarity (Assumption3.3) and gradient dissimilarity (Assumption4.1) are orthogonal.“ This statement is hand-wavy.
  e. Uses the term “round” without defining it. I understand it is probably coming from FL literature.

5. I can see that authors have made some effort in making the proofs parsable. However there are many typos/confusion in the algorithms and proofs. Due to this it is not easy to check the veracity of the statements. Please proof read and revise them. Considering this is a journal venue with rolling submissions, it would have been better if author had proof read the paper more carefully. I will list my concerns below

## Theorem 4.2 and naive algo:

After reading the theorem further it is clear that lower bound holds even if hessian similarity is perfect. A better way to write the theorem 4.2 would be to say that under perfect hessian similarity assumption the lower bound could be arbitrarily bad depending on the gradient bias. Then say the the lower bound depends on the gradient bias. Otherwise, it is confusing to have different assumptions for naive and proposed algorithms.

## Theorem 4.3 and AuxMom:

1. Proof of Lemma C.8: seem to miss the term $(6 \delta^2/a) K \eta^2 \sigma_h^2$ (corresponding to last term in Lemma C.7). This seems potentially fixable, but this means rest of the proof needs reworking before verification.
2. If E0 can be removed why isn’t main theorem written with E0 removed?
3. Final step of the theorem proof (where we eliminate E^0) is not obvious. Please elaborate the steps. It is not clear how $a=max(1/T, 32 K \eta \delta)$ is helpful.
4. Even after taking care of E0, why does the upper bound have a dependence on $1/\delta$? Shouldn’t the problem get easier as $\delta$ is small? This term seems to be ignored in most of analysis and follow-up interpretation.

### Minor errors
1. Algorithm 3: $\xi^{t-1}$ instead of $\xi^{t}$
2. Proof of  Lemma C.7: typo $K(K+1)$ instead of $K^2$
3. Proof of Lemma C.8: typo it should be 36 instead of 32
4. Proof of Lemma C.9:  typo missing square in the LHS

## Theorem 4.5 and AuxMVR:

AuxMVT seems to make some non-intuitive choices to differ from AuxMom and STORM (Originally introduced MVR).

1. Why is momentum update done at end of the round whereas in AuxMom it is done at the beginning? It would be good to provide justification for this choice. Better is it possible to unify the algorithm to make it consistent?

2. Why is $m^{t-1}$ directly not used like in STORM instead of $(1-a)m^{t-1} + a g_prev^{f-h}$? Can we simplify Lemma C.11 (eliminate the last variance term) using the simpler update? If yes, I highly recommend authors doing so.

3. AuxMVR also departs from Algorithm 1 pseudo code, since $m^{t-1}$ is used instead of $m^t$.

### Minor errors
1. There seems to be some confusion on whether $\xi^{t-1}$, $\xi^{t-2}$, or $\xi^t$ is being used at various places. Algorithm 3 uses $x^{t-1}$ and $\xi_{f-g}$ but analysis uses $x^{t-2}$, $\xi_{f}$ and $\xi_{g}$.
2. Proof of C.11: Missing overhead bar over $d^t_k$ in the final inequality LHS
3. Proof of C.12: $\pm$ is not a standard notation.
4. Proof of C.13: References C.5 instead of C.12.
5. Proof of C.16: Typo in last inequality $\Delta^{t}$
6. Proof of Theorem 4.5: What is $\tilde{\delta}$? Upper bound on $E^0$ is using wrong variance.

## Experiments:
1. It would be great if the authors can provide the code for the experiments.
2. Fig 2: Why does speed of fine tuning stage of FT depend on the delta when $f(x) = x^2/2$ always? This is counter intuitive.
3. MNIST: From the description it is not clear how many samples are used in f and h. Number of samples in f and h is important in the trade-off between using f and h here.
4. Real datasets:
  a. It would be nicer if the authors compared AuxMVR with AuxMom in the experiments.
  b. Crucially authors do not compare against a case when the models are purely trained on f. This will characterize the gain and speed or generalization gain of using h.
  b. What is the batchsize used for these experiments? It would be good to do run some ablation on batchsize.

---

> ### Author Response · Authors · 2024-01-01
> **Responding to your concerns and new version of the paper**
>
> Dear Reviewer, We would like to thank you for the very good review and all the time you spent on it; it was very helpful to us.
> We will try to address your concerns in this reply.
>
> - **Theorem 4.2**. We changed it to what you suggested. It makes more sense.
>
> - **Theorem 4.3**. 1) indeed, we missed that term in Lemma C.8; we were clumsy because, in a previous version, we did not consider the case where gradients of $h$ are noisy, but then realized that we could do it and we did it on top of the, then, already existing proof and missed this. We corrected it now; it led at the end to an additional term of $\mathcal{O}(\sigma_h^2/K)$ which corresponds to the error of solving the inner problem defined by $h$ (Eq. 4 in the paper). 2) **Eliminating $E^0$**, indeed it was not very clear; we had a term like this $E^0/(aT)$ and the dependence on $\delta$ which is indeed a problem that we did not fully appreciate only comes from replacing $a$ by its value $\sim K\delta\eta$, but if we choose $a=max(1/T,K\delta\eta)$ (ignoring multiplicative constants), then $a\geq 1/T$ and thus $E^0/(aT) \leq E^0\leq \sigma_{f-h}^2/T$ and we don't have the problem anymore. we avoided this version of our Theorem because in practice we did not need to enforce this condition on $E^0$, we now use this version in the main text.
>
> - **Theorem 4.5 and AuxMVR.** The version of AuxMVR you suggest also works, its proof is even simpler than the version we had in the paper as we only need to change Lemma C.8 (we kept the complicated version because we had it written, but we changed it now).
>
> - **Experiments.** 1) we will provide the code, we just realized that the link in the paper leads to an outdated repository (we will force it to update or provide a new link). 2) FT for the simple function, the problems come from our old choice of the step size $\eta = min(0.5,1/K\delta)$ which depends on $\delta$, we updated these experiments. 3) we used a 256 batch size for $f$ and a $64$ batch size for $h$, we rerun the experiment with an $h$ batch size of 128,256 and 512 but the results were very similar (there were fewer oscillations but overall the changes are negligible) 4) the case K=1 is the same as running SGDm, from the plots we see that increasing K improves the results, thus using $h$ improves the performance. 5) AuxMOM vs AuxMVR, AuxMVR leads to similar results as AuxMOM, we couldn't observe a huge improvement from using it, we suspect this is because we did not tune the momentum parameter $a$ as in our theory it is suggested it should be smaller for AuxMVR, we also tried to compare SGDm to STORM (MVR), but they led to similar results too (We did not use the adaptive version of STORM which works better as our theory does not take this into account):
>
> We also added an experiment on Coresets which suggests that we can minimize the loss in performance by combining Coresets with AuxMOM.
>
> We wrote the paper, if you have the time to check the new version, it would be very helpful for us.

---

> > ### Comment · Reviewer_ZsY4 · 2024-01-09
> > **Minor typos and diff**
> >
> > Dear authors,
> >
> > Thank you very much for updating the paper and fixing the algorithms & theorems. I am still reviewing the changes. However, I noticed that paper still has some typos, e.g. eqn (6). Would you be able to proof read and fix the typos? Additionally, would it be possible for you to color code or provide a diff of the changes between the submission and the latest version.
> >
> > Thanks and regards,
> >
> > Reviewer

---

> > > ### Author Response · Authors · 2024-01-09
> > >
> > > Dear Reviewer,
> > >
> > > Sorry, we missed this typo. We went through the whole paper, corrected typos and punctuation, and downplayed some statements judged as over-claiming.
> > >
> > > We hope equation (6) was the last typo; we will go through the paper again to ensure nothing was missed.
> > >
> > > We will provide a diff pdf between the current version and the first submission as supplementary material.
> > >
> > > Thank you very much again for your time.

---

> > > > ### Comment · Reviewer_ZsY4 · 2024-01-12
> > > > **Thanks for updates**
> > > >
> > > > Thanks for making the suggested updates and providing the diff. I think authors have addressed most of my previous concerns including fixing the theorems, simplifying algorithm, and attributing prior work. Also the latest coreset experiments are insightful. I would also like to suggest the authors add the reason for not using SARAH like updates in the main text as these negative results will be useful for future work.
> > > >
> > > > I have only two further concerns
> > > > 1.
> > > > > "Remarkably, although we are using SGD steps of the helper h, we get a $1/K$ error instead of $1/\sqrt{K}$; this can be explained by the fact that we initialize using the (approximate) solution of the last inner problem which accelerates convergence; we can potentially get faster rates by using variance reduction methods on h"
> > > >
> > > > Is it possible to elaborate more and make this more concrete? Will it help to analyze the case where $f=h$ and $\sigma_f = 0$, but $\sigma_{h} = \sigma_{f-h} \neq 0$? I also note that once you factor in the value of $\beta$ we do get a term $O(\sqrt{L F^0 \sigma^2_h/KT})$.
> > > >
> > > > 2. "Algorithm 4....works well in practice and is more robust...": Do you provide any experimental evidence for this claim? May be I missed something. If not, it would be great for readers to see these results.

---

> ### Author Response · Authors · 2024-01-12
>
> Dear Reviewer,
> Thanks again for the feedback,
>
> - We will add a paragraph on the SARAH-like update rule and updating with a probability $p$ instead of after $K$ steps.
>
> - Yes, there is "implicitly" a term $\mathcal{O}(\sqrt{LF^0\sigma_h^2/(KT)})$, which should be much smaller than the dominant term for large values of $K$, note that this is what leads to the term $1_{\sigma_h\neq 0}\frac{LF^0}{\varepsilon}$ in the iteration complexity.
>
> - The case that you are suggesting $\sigma_f=0$ and $\sigma_h=\sigma_{f-h}\neq 0$ is exceptional as we thought that when there is no noise in $f$, we should ask the same from $h$, but still we can get an improvement even in this case, although it is not covered in Corollary 4.4. In this case, we have the rate $\mathcal{O}\Big(\sqrt{LF^0\sigma_h^2/(KT)} + \delta F^0/T + LF^0/(KT) + \sigma_{f-h}^2/T + \sigma_h^2/K\Big)$ and we should compare this to $\mathcal{O}\Big(LF^0/T\Big)$, all the terms that depend on $K$ are not a problem as we can take $K$ big enough to make them smaller than $LF^0/T$, the only potential problems are the terms $\delta F^0/T$ and $\sigma_{f-h}^2/T$, which will be smaller than $LF^0/T$ only if $\delta\ll L$ and $\sigma_{f-h}^2\ll LF^0$.
>
> - If we also assume $f=h$, then we have $\delta=0$ and we still need to have $\sigma_{f-h}^2\ll LF^0$. We note that in this case, it does not make sense to assume $\sigma_f=0$ and $\sigma_h=\sigma_{f-h}\neq 0$, as we can we the same effort sample true gradients of $h$.
>
> - To go back to our statement about getting a $\sigma_h^2/K$ term instead of $\sigma_h^2/\sqrt{K}$, this has to do with the form of our update which $g_h + m_{f-h}$, where $m_{f-h}$ is fixed inside the inner iterations if we ignore this $m_{f-h}$ term, then we are using SGD on $h$ and should expect a $1/\sqrt{K}$ term in our rate, but we get a faster $1/K$ rate; a potential explanation for this is that it has to do with how we initialize the inner problem.
>
> - Algorithm 4, we did not provide any experimental evidence, but we considered this algorithm in the beginning but could not prove any theoretical improvement for it; we just thought we could mention it in the paper. We will rewrite the statement.

---

> > ### Comment · Reviewer_ZsY4 · 2024-01-18
> > **Thanks for the clarification**
> >
> > Thanks for the clarification

---

### Decision · Action_Editor_xUYQ · 2024-02-14

**Recommendation:** Accept as is

**Comment:**

This is a nice conceptual contribution, underpinned by sound theoretical analysis, albeit not particularly novel. The experiments do a good of providing some partial evidence in favor of the proposed methods, but the full practical implications and impact are still not clear.

**Audience:**

Yes. Individuals interested in a different perspective on a set of common techniques in distributed optimization/federated learning/stochastic optimization may benefit from this paper.

**Claims And Evidence:**

Yes. Mathematical theorems are proved in full and sufficient empirical evidence is presented.